# Lunar eclipses illuminate timing and climate impact of medieval volcanism

Sébastien Guillet[1✉], Christophe Corona[1,2], Clive Oppenheimer[3], Franck Lavigne[4], Myriam Khodri[5], Francis Ludlow[6], Michael Sigl[7,8], Matthew Toohey[9], Paul S. Atkins[10], Zhen Yang[6], Tomoko Muranaka[1], Nobuko Horikawa[10] & Markus Stoffel[1,11,12]

Explosive volcanism is a key contributor to climate variability on interannual to centennial timescales[1]. Understanding the far-field societal impacts of eruption-forced climatic changes requires firm event chronologies and reliable estimates of both the burden and altitude (that is, tropospheric versus stratospheric) of volcanic sulfate aerosol[2,3]. However, despite progress in ice-core dating, uncertainties remain in these key factors[4]. This particularly hinders investigation of the role of large, temporally clustered eruptions during the High Medieval Period (HMP, 1100–1300 CE), which have been implicated in the transition from the warm Medieval Climate Anomaly to the Little Ice Age[5]. Here we shed new light on explosive volcanism during the HMP, drawing on analysis of contemporary reports of total lunar eclipses, from which we derive a time series of stratospheric turbidity. By combining this new record with aerosol model simulations and tree-ring-based climate proxies, we refine the estimated dates of five notable eruptions and associate each with stratospheric aerosol veils. Five further eruptions, including one responsible for high sulfur deposition over Greenland circa 1182 CE, affected only the troposphere and had muted climatic consequences. Our findings offer support for further investigation of the decadal-scale to centennial-scale climate response to volcanic eruptions.

Large explosive volcanic eruptions can inject enormous quantities of sulfur-bearing gases into the stratosphere, where they generate sulfate aerosols[1]. The resulting aerosol veils perturb the energy budget of the Earth, inducing seasonal and regional surface temperature and precipitation anomalies, whose severity in combination with societal vulnerabilities has been linked to historical cases of agronomic and pasturage deficits, civil and political unrest, pestilence and migration[6]. Although the geologic record constitutes the primary evidence of past volcanism, with chronologies based on radiocarbon and other radiometric methods, polar ice cores arguably furnish the most comprehensive and accessible picture of climatically notable volcanism through the compilation of sulfur deposition time series[2,4]. Of particular note in such records is a proliferation of sulfur-rich eruptions during the HMP (circa twelfth and thirteenth centuries), beginning with a cluster of events around 1108–1110 CE (ref. [7]) and including the colossal Samalas eruption around 1257 CE (refs. [8,9]). These events have been linked with substantial cooling and subsistence crises[7,9] and the combined effect of their forcing has been posited as a contributor to the onset of the Little Ice Age[5].

The dating of past volcanic events from ice cores presents several challenges owing to the complexity of atmospheric transport leading to temporally and spatially variable sulfur deposition[10], poorly constrained age models[11–13] and uncertainties in layer counting related to accumulation rates and post-depositional processes[3]. A further challenge is the discrimination between tropospheric and stratospheric transport of volcanic aerosol, the latter being more indicative of a climate-forcing explosive eruption[4]. Sulfur isotopic ratios measured in ice cores can help to make this distinction, but the approach has not been extensively applied and does not necessarily distinguish between tropospheric and lower stratospheric (below ozone layer) aerosol transport[3,14].

The rare and often visually spectacular atmospheric optical phenomena that can arise from the presence of volcanic dust veils in the stratosphere, such as solar dimming, coronae or Bishop's rings, peculiar twilight coloration and dark total lunar eclipses, have long been regarded as portents worth recording. References to such phenomena have provided independent evidence to evaluate the timing and impact of volcanism for the periods 1500 BCE to 1000 CE (refs. [2,15]), 1500–1880 CE (refs. [16,17]) and 1880–2000 CE (refs. [18,19]). Here we focus on the notable lacuna in past studies, that is, the HMP, and on references in Eurasian sources to the coloration of total lunar eclipses, as they are relatively frequent and their occurrences are known precisely

[1]Climate Change Impacts and Risks in the Anthropocene (C-CIA), Institute for Environmental Sciences, University of Geneva, Geneva, Switzerland. [2]GEOLAB, Université Clermont Auvergne, CNRS, Clermont-Ferrand, France. [3]Department of Geography, University of Cambridge, Cambridge, UK. [4]Laboratoire de Géographie Physique, Université Paris 1 Panthéon-Sorbonne, Thiais, France. [5]Laboratoire d'Océanographie et du Climat: Expérimentations et Approches Numériques, IPSL, Sorbonne Université/IRD/CNRS/MNHN, Paris, France. [6]Trinity Centre for Environmental Humanities, Department of History, School of Histories & Humanities, Trinity College Dublin, Dublin, Ireland. [7]Climate and Environmental Physics, University of Bern, Bern, Switzerland. [8]Oeschger Centre for Climate Change Research, University of Bern, Bern, Switzerland. [9]Department of Physics and Engineering Physics, University of Saskatchewan, Saskatoon, Saskatchewan, Canada. [10]Department of Asian Languages & Literature, University of Washington, Seattle, WA, USA. [11]Department of Earth Sciences, University of Geneva, Geneva, Switzerland. [12]Department F.-A. Forel for Environmental and Aquatic Sciences, University of Geneva, Geneva, Switzerland. ✉e-mail: sebastien.guillet@unige.ch

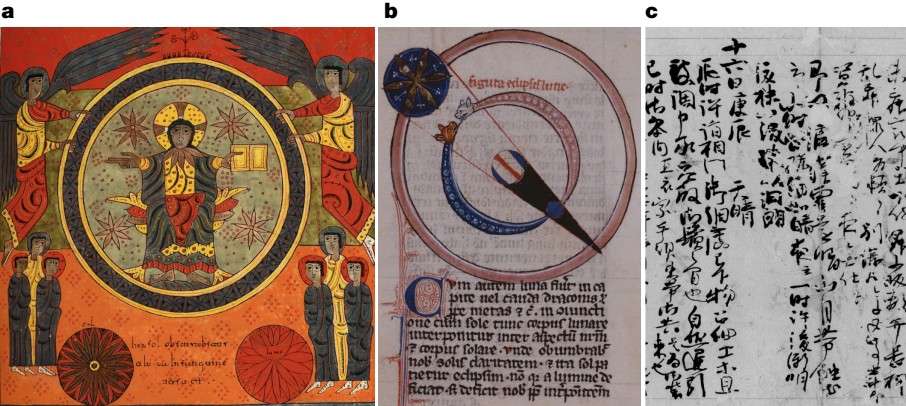

**a**

**b**

**c**

**Fig. 1 | Representations of lunar eclipses in medieval manuscripts.**
**a**, *Commentary on the Apocalypse* by Beatus of Liébana, from the monastery of Santo Domingo de Silos, near Burgos, Spain, 1090–1109 CE. Credit: British Library Board (Add. MS 11695, f108r). The text at the bottom of the miniature, between the dark circle on the left representing a total solar eclipse and the red circle on the right representing a total lunar eclipse, reads: "hic sol obscurabitur et luna in sanguine versa est" ("and the Sun was obscured and the Moon turned into blood"). The blood-red eclipsed Moon was seen as one possible sign of the Apocalypse. Lunar occultation descriptions from the Middle Ages often follow the Book of Revelation, suggesting that the Bible provided justification and inspiration for recording lunar eclipses and their colour. **b**, Thirteenth-century depiction of a lunar eclipse by Johannes de Sacrobosco. Credit: The New York Public Library (*De Sphaera*, MssCol 2557, f112v). **c**, Facsimile of the *Meigetsuki* (明月記) diary by Fujiwara no Teika (藤原定家) describing the total lunar eclipse of 2 December 1229 CE. Credit: *Meigetsuki*, vol. 4, pp. 517, 2000. Reizei-ke Shiguretei Bunko. Tokyo: Asahi Shinbunsha. Teika mentions this event twice. The figure shows the first entry: "[…] the sky was free of clouds into the distance and the Moon over the hills emerged in eclipse, total for a little while, [its light] meagre as on a dark night. About an hour later it brightened gradually, and after it was extinguished [during the eclipse] it was especially luminous". The second entry, written four days later, details the unusual coloration of the Moon. Over the centuries, several portions of the Meigetsuki were cut apart, and the entry for 6 December 1229 CE is held in a private collection[49,50].

from astronomical retro-calculation. We derive an independent proxy for volcanic dust veil from medieval records of lunar eclipses and use the resulting time series, in conjunction with climate model outputs and summer temperature reconstructions from tree rings, to refine the NS1–2011 (Greenland) and WD2014 (Antarctica) ice-core chronologies, which hitherto have provided the primary constraints on timing of HMP eruptions[2,4]. The chronologies identify seven HMP eruptions that generated estimated volcanic stratospheric sulfur injections (VSSI) exceeding 10 Tg. Each of them ranks among the top 16 VSSI events of the past 2,500 years (refs. [2,4]). Their estimated eruption years are 1108 CE (UE1; in which UE stands for unidentified eruption; see Methods), 1171 CE (UE2), 1182 CE (UE3), 1230 CE (UE4), 1257 CE (Samalas), 1276 CE (UE5) and 1286 CE (UE6). We consider these events along with 13 lesser HMP eruptions and seek to confirm or refine the existing estimates of eruption year and season and to discriminate between tropospheric and stratospheric aerosol veils.

## Eclipses unveil stratospheric turbidity

The brightness of the Moon during eclipse is highly sensitive to aerosol abundance in the stratosphere. Dark total lunar eclipses indicate high turbidity, whereas a ruddy disk signifies a clear stratosphere[18,19]. To reconstruct past stratospheric turbidity across the Medieval Climate Anomaly to Little Ice Age transition, we comprehensively reexamined a large corpus of historical sources (Supplementary Dataset S1) written in the twelfth and thirteenth centuries, in search of credible lunar eclipse observations (Fig. 1). In China and Korea, lunar eclipses were recorded by official astronomers and preserved in sources such as the astronomical treatises and five-elements treatises of official dynastic histories, whereas Japanese eclipse observations are found in more diverse writings such as the diaries of courtiers, chronicles or temple records. In Europe, annals and chronicles from monasteries and towns represent the main sources. In Arabic sources, lunar eclipse observations are frequently found in universal chronicles[20].

According to the latest catalogues of lunar eclipses[21,22], 64 (Europe), 59 (Middle East) and 64 (East Asia) total lunar eclipses occurred and

would have been visible, weather permitting, between 1100 and 1300 CE. A total of 180 European, 10 Middle Eastern and 199 East Asian accounts describe 51, 7 and 61 individual total lunar eclipses, respectively. In Europe, although 12 individual eclipses are described in only one surviving source, many are corroborated in several accounts, up to 16 in the case of the eclipse of 11 February 1161 CE (Supplementary Dataset S1). This success in finding observations of retro-calculated eclipses for Europe (80%; Extended Data Table 1) is notable and comparable with that of later periods for which documentation is more abundant[16,17] (for example, 82%, 1665–1881 CE). It reflects the contemporary proliferation and geographic span of monastic communities across Europe[20], which improved the overall chances of clear-sky observations, and the attention that some chroniclers paid to celestial phenomena[23]. Observations of the Sun, the Moon or the stars were required to calculate the hours for prayer, as not all monasteries possessed water clocks or astrolabes for timekeeping[24–26]. Accurate observations of the age of the Moon were also important for the correct identification of the Easter full Moon, which served as the key point of reference for Easter Sunday and all other moveable feast days of the liturgical year[25,27]. East Asian records sometimes contain predictions rather than observations[28] and, for this reason, the proportion of eclipses documented by observers in China, Korea and Japan was not computed.

## Dark eclipses follow substantial HMP eruptions

Western and Eastern Christian sources together provide information on the colour and brightness of the Moon for 36 eclipses (Fig. 2a). Such attention to brightness is largely absent for Asian records[20,29], in which only one account describes coloration. References to a 'blood-red Moon' in Western and Eastern Christian sources are probably informed by texts such as the Book of Revelation of John, in which the blood Moon, along with earthquakes and solar eclipses, portended the End Times (Revelation 6:12–17; Fig. 1a). Lunar eclipse coloration was imbued with particular significance for Christian observers and often regarded as an ill omen, presaging disasters[26,30,31], emphasizing the influence of the Bible on the perception of natural phenomena during the Middle

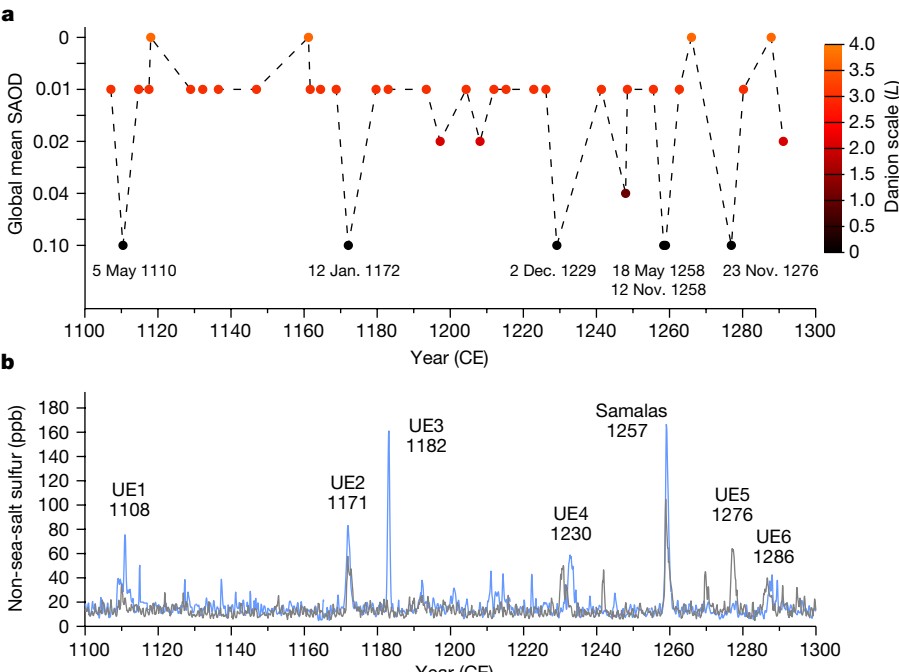

**Fig. 2 | Stratospheric turbidity derived from total lunar eclipse coloration and non-sea-salt sulfur records from polar ice cores. a**, Total lunar eclipse descriptions retrieved from European, Middle Eastern and East Asian historical sources from 1100 to 1300 CE (Supplementary Dataset S1), rated on the Danjon scale (right $y$ axis), and converted to equivalent global mean $SAOD_{550}$ (SAOD at 550 nm; left $y$ axis). **b**, Monthly resolved non-sea-salt sulfur concentrations from the Greenland NEEM-2011-S1 (blue line) and Antarctica WDC06A (grey line) ice cores[2].

Ages[32]. This does not mean that the physical causes of lunar eclipses were unknown to all medieval chroniclers[31]. Ancient Babylonian, Greek and later Muslim astronomers not only understood eclipse mechanisms but were able to predict lunar occultations[20], and this knowledge was ultimately transmitted to medieval Europe, as evident in contemporary astronomical treatises (for example, *De Lunationibus*, 1112 CE, and *De Dracone*, 1120–1121 CE, by Walcher of Malvern[33]; *De Sphaera*, circa 1230 CE, by Johannes de Sacrobosco[34]; Fig. 1b). Both natural and supernatural interpretations of lunar eclipses coexisted in the Middle Ages[31], underpinning the recovery of a near-complete series of lunar eclipse coloration spanning two centuries (Fig. 2a).

We rated the colour and luminosity of each observed eclipse on the Danjon scale[35], which quantifies naked-eye lunar brightness. It ranges from $L = 0$ (very dark) to $L = 4$ (very bright copper-red or orange eclipse). Of the 37 total lunar eclipses with brightness recorded in Eurasian sources, only six were rated $L = 0$, emphasizing the rarity and significance of such observations (Fig. 2a and Supplementary Dataset S1). These events occurred on the night of 5–6 May 1110 CE, 12–13 January 1172 CE, 2–3 December 1229 CE, 18–19 May 1258 CE, 12–13 November 1258 CE and 22–23 November 1276 CE. All testimonies are reported in Supplementary Dataset S1, with each description stressing a near-complete and prolonged disappearance of the lunar disk. One of the most outstanding accounts was retrieved from Japanese sources and pertains to the total lunar eclipse of 2 December 1229 CE. Although Asian sources rarely detail coloration[20,29], the *Meigetsuki* (明月記, *The Record of the Bright Moon*) written by Fujiwara no Teika (藤原定家, 1162–1241 CE) reports an extremely dark lunar eclipse despite clear weather. The *Meigetsuki* notes that the coloration of the Moon was deemed so unusual that the astronomers expressed fear over its appearance: "Regarding the recent total lunar eclipse, although on previous occasions there has been totality, the old folk had never seen it like this time, with the location of the disk of the Moon not visible, just as if it had disappeared during the eclipse. Moreover, the duration was very long, and the change was extreme. It was truly something to

fear. Indeed, in my seventy years I have never heard of or seen [such a thing]; the official astronomers spoke of it fearfully […]" (Fig. 1c and Supplementary Dataset S1).

All the dark ($L = 0$) lunar eclipses—in May 1110 CE, January 1172 CE, December 1229 CE, May 1258 CE, November 1258 CE and November 1276 CE—are contemporary with five of the seven largest HMP volcanic sulfate signals recorded in polar ice cores (UE1, UE2, UE4, Samalas and UE5; Fig. 2b), strongly suggesting that the darkening of the eclipsed Moon was related to the presence of volcanic aerosols in the stratosphere. This finding mirrors previous work that found all very dark total lunar eclipses since 1600 CE followed substantial volcanic eruptions[16–19,36] (Extended Data Table 2). For the remaining two of the top seven HMP eruptions, circa 1182 (UE3) and 1286 CE (UE6), descriptions of reddish ($L = 3–4$) total lunar eclipses point to low stratospheric turbidity in August 1179, December 1182 and October 1287 CE.

## Timing of HMP eruptions

We constrained the timing of HMP eruptions by developing a four-step procedure that integrates evidence from our eclipse record, global aerosol simulations, modern satellite observations and tree-ring reconstructions (see Methods and Extended Data Fig. 1). First, the appearance of lunar eclipses reported in historical archives was rated on the Danjon scale and converted to stratospheric aerosol optical depth (SAOD) following refs. [16–19], which showed that, for 46 well-observed lunar eclipses between 1880–1888 CE and 1960–2001 CE, dark total lunar eclipses ($L = 0$) only occurred when SAOD exceeded about 0.1. Next, drawing on observations for the 1883 Krakatau and 1991 Pinatubo eruptions (from the Sato/GISS and GloSSAC v2 datasets)[37,38], SAOD simulations[39] (from the eVolv2k datasets for UE1 to UE6 and both the circa 1257 CE Samalas and 1815 CE Tambora eruptions) and IPSL-CM5A-LR[40] climate model outputs (for Samalas and Tambora), we evaluated the post-eruptive duration of elevated stratospheric turbidity, that is, SAOD ≥ 0.1. This suggested that a total lunar eclipse is most probably observed as dark

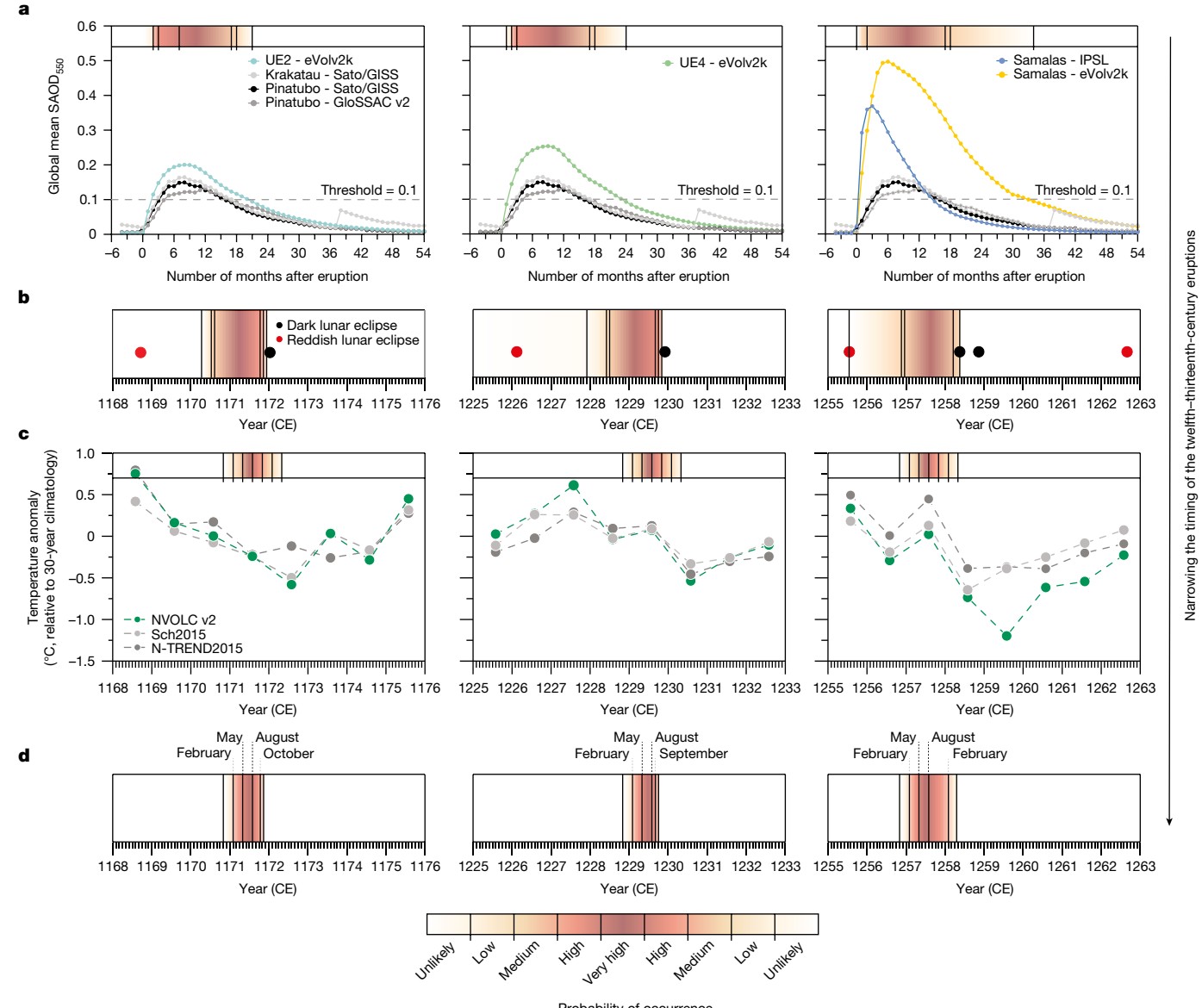

**Fig. 3 | Constraining the timing of HMP (1100–1300 CE) volcanic eruptions.**
**a**, Residence time of volcanic stratospheric aerosols and time windows with SAOD exceeding about 0.1. The residence time of aerosols is based on global mean SAOD₅₅₀ time series from the Sato/GISS[37] and GloSSAC v2 (ref. [38]) datasets (for the 1883 CE Krakatau and 1991 CE Pinatubo eruptions) and simulated by the EVA forcing generator[4,39] (for UE2, UE4 and the 1257 CE Samalas eruption) and the IPSL-CM5A-LR model[40] (for the Samalas eruption). Probability of occurrence of HMP eruptions based on the timing of dark lunar eclipse dates (**b**) and tree-ring records[7,41,42] (**c**). **d**, Integration of **b** and **c** to estimate the most probable time windows for UE2, UE4 and the Samalas eruption.

between 3 and 20 months following an eruption. Thus, we assume that, given an observation of a dark lunar eclipse, the eruption responsible occurred between 20 and 3 months beforehand. This assumption is corroborated if we consider the seven largest VSSI events since 1600, which were all followed by dark lunar eclipses between 9 (1912 CE Katmai, 1982 CE El Chichón), 14 (1815 CE Tambora, 1883 CE Krakatau), 18 (1991 CE Pinatubo) and 20 (1600 CE Huaynaputina) months later (Extended Data Table 2). The case of the 1963 CE Agung eruption provides further corroboration of our approach because, of the three lunar eclipses that occurred 10, 15 and 21 months after this eruption, only the last (falling outside our 3–20-month window) was not rated *L* = 0 on the Danjon scale (Extended Data Table 2). Then, to provide further constraint on the timing of each HMP eruption, we evaluated independently dated Northern Hemisphere tree-ring-based summer temperature reconstructions for post-volcanic climatic response (Sch2015 (ref. [41]), N-TREND2015 (ref. [42]), NVOLC v2 (ref. [7])).

Taking UE2 (Fig. 3a) as an example, combining the dark lunar eclipse date of January 1172 CE with the eVolv2k, Sato/GISS and GloSSAC v2 datasets, we find a high to very high probability that the event occurred between July 1170 and October 1171 CE (Fig. 3b). The peak cooling observed in Sch2015 and NVOLC v2 in summer 1172 CE reduces the likelihood of an eruption occurring between summer 1170 and autumn 1171 CE (Fig. 3c); we thus bracket the eruption date between May and August 1171 CE (Fig. 3d). We similarly constrain the probable time windows of other large HMP events to Northern Hemisphere winter 1108/1109 CE (UE1; Extended Data Fig. 2), Northern Hemisphere spring/summer 1229 CE (UE4) and Northern Hemisphere spring/summer 1257 CE for Samalas (Fig. 3d). This refined timing for the Samalas eruption is consistent with the isopach pattern of tephra fall in Indonesia[8], pointing to a dry-season eruption (between May and October) and counters an argument for a 1256 CE eruption date[43]. For UE5, we find a time window between September 1275 and July 1276 CE (Extended Data

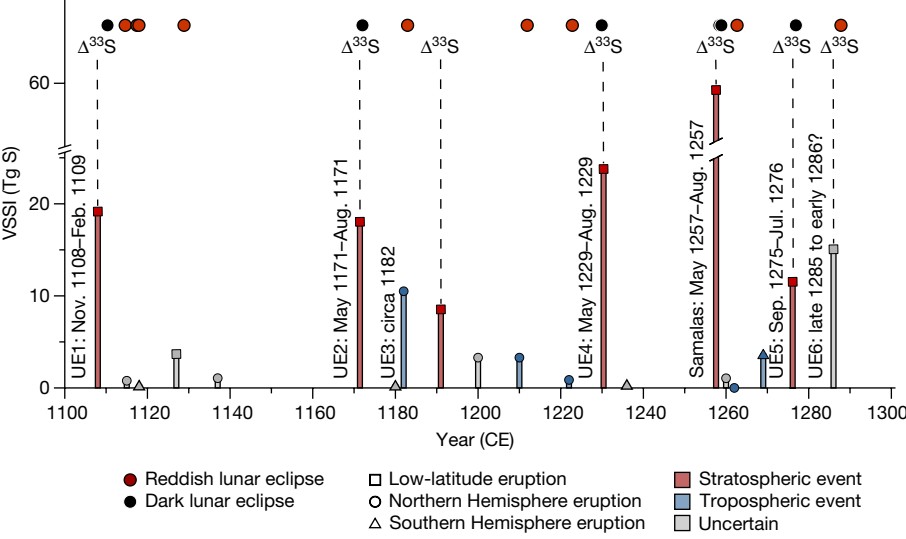

**Fig. 4 | Revised chronology of explosive volcanism in the twelfth and thirteenth centuries.** Vertical bars are based on the eVolv2k volcanic forcing reconstruction[4] and indicate the magnitude of VSSI. Using total lunar eclipse coloration (red and black dots) and Δ³³S isotope records[3], we discriminate between stratospheric (red bars) and tropospheric (blue bars) dust veils. Grey bars show uncertain events. Squares, circles and triangles refer to low-latitude, Northern Hemisphere extratropical and Southern Hemisphere extratropical eruptions, respectively[2].

Fig. 2). Further constraint is not possible in this case as pronounced summer cooling is not evident in the tree-ring-based temperature reconstructions (Extended Data Fig. 2).

## Stratospheric or tropospheric aerosols?

Stratospheric aerosols profoundly affect the brightness and coloration of the eclipsed Moon by reducing the transmission of sunlight into the Earth's umbra[18,19]. Here we use lunar eclipse coloration to distinguish between tropospheric and stratospheric aerosol veils (see Methods). The dark lunar eclipses observed after UE1, UE2, UE4, Samalas and UE5 indicate stratospheric aerosol veils (Fig. 4), consistent with Δ³³S isotope records from Dome C, Antarctica[3]. Although UE3 (circa 1182 CE) registers as the second greatest Northern Hemispheric extratropical eruption of the last millennium in terms of ice-core sulfate deposition[4] (second only to Laki 1783–1784 CE), the reddish lunar eclipse of 11 December 1182 CE (Fig. 2a) points to a comparatively low stratospheric aerosol burden, while tree-ring reconstructions show limited cooling (Supplementary Dataset S2). Further consideration of eclipse coloration, in conjunction with tree-ring-based summer temperature reconstructions, suggests that for four HMP eruptions associated with lesser VSSI around 1127, 1210, 1222 and 1262 CE, sulfate aerosols were mostly confined to the troposphere and any climatic impacts were limited (Fig. 4 and Supplementary Dataset S2).

The nature and timing of UE6 remain enigmatic (Fig. 4). A bipolar sulfate deposition distribution is observed around 1286 CE, implying a tropical eruption, whereas sulfur isotopic analysis suggests that the associated Antarctic deposition was stratospheric in origin[20]. However, the reddish lunar eclipse of 22 October 1287 CE (Fig. 2a), described in three independent records (Supplementary Dataset S1) from England and Italy, precludes substantial Northern Hemisphere stratospheric aerosol presence at this time. This apparent discrepancy may reflect a late 1285 or early 1286 CE eruption, thus being too early to darken the October 1287 CE lunar eclipse. Regardless, the tree-ring proxies do not show substantial Northern Hemisphere cooling during the period 1280–1290 CE, suggesting limited climatic impacts of UE6 (Supplementary Dataset S2).

## Implications

Our identification of marked stratospheric dust veils in 1110, 1172, 1229, 1258 and 1276 CE using contemporary observations of dark total lunar eclipses corroborates the general accuracy of the revised ice-core chronologies for Greenland (NS1–2011) and Antarctica (WD2014), while adding precision to the chronological framework of HMP eruptions. Given the inherent uncertainties in ice-core chronologies, our dataset of precisely dated dark total lunar eclipses offers a new, reliable and independent suite of chronological tie points that can complement established age markers in 536, 774/5, 939, 993/4, 1258, 1601 and 1816 CE to aid future chronology development. Our findings also suggest that five other events, likely associated with tropospheric-only aerosol veils, had a limited impact on climate.

No single source or method can, however, offer complete chronological control, and our eclipse data also have limitations (see Methods). The visibility of lunar eclipses is geographically and meteorologically constrained; accounts of partial and penumbral eclipses cannot be used, reducing the number of available records; and only comments on the colour of the Moon are relevant. We have thus developed here a multiproxy approach using diverse sources and methods that reflect or model different aspects of the volcano–climate system, each offering complementary constraints on eruption timing. These include ice-core sulfate deposition profiles, aerosol model simulations and palaeoclimatic proxies, as well as our eclipse observations. Further developing such integrative approaches will pave the way to even finer temporal resolutions, in particular as the representation of stratospheric aerosol formation, evolution and duration in climate models and the resolution of palaeoclimate records continue to improve (ref. [44]; see Methods).

Better constraints (ideally sub-annual) on the timing of historical explosive volcanic eruptions are critical because climatically important factors including aerosol distribution, altitude, size and radiative forcing are all influenced by the seasonally changing stratospheric circulation[45,46]. For unidentified eruptions, climate modelling typically uses notional eruption dates (for example, 1 April in the Community Earth System Model, 1 January in eVolv2k)[39,47] and assumes stratospheric aerosol presence, biasing the distribution, magnitude and persistence of modelled post-eruption thermal and hydroclimatic anomalies and

potentially contributing to persistent model–proxy discrepancies[45,48]. Our findings thus offer improved parameterizations for the next generation of Community Earth System Models in investigations of the impacts of HMP eruptions. We hope that our new dataset will help to inform the extent of their role in the onset of the Little Ice Age.

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

## Methods

### Description of the twelfth–thirteenth-century eruptions investigated

Ice-core records from Greenland and Antarctica suggest the occurrence of nine low-latitude eruptions dated, according to the NS1–2011 and WD2014 timescales[2,4], to 1108, 1127, 1171, 1191, 1230, 1257, 1260, 1276 and 1286 CE, seven Northern Hemisphere extratropical events (1115, 1137, 1182, 1200, 1210, 1222 and 1262 CE, identified by deposition signals in Greenland only) and four Southern Hemisphere extratropical events (1118, 1180, 1236 and 1269 CE, identified by deposition signals in Antarctica only) in the twelfth and thirteenth centuries. This period therefore represents one of the most volcanically perturbed periods of the past 2,500 years. The largest volcanic eruptions—with VSSI > 10 Tg S in ref. [4] —are UE1 (1108 CE), UE2 (1171 CE), UE3 (1182 CE), UE4 (1230 CE), the Samalas (circa 1257 CE) eruption[8], UE5 (1276 CE), UE6 (1286 CE) and rank as the 7th (VSSI, 19.2 Tg S), 10th (18.1 Tg S), 16th (10.1 Tg S), 4th (23.8 Tg S), 1st (59.4 Tg S), 15th (11.5 Tg S) and 13th (15.1 Tg S) largest volcanic events of the last millennium by sulfate deposition. With the exception of the circa 1257 CE event, attributed to Samalas in Indonesia[8], the sources of these eruptions remain unidentified. Although the 1108 CE sulfate spike was originally attributed to an eruption of a tropical volcano[2], a recent reassessment of ice-core records coupled with historical sources suggests that at least two eruptions occurring between 1108 and 1110 CE are registered in the observed polar sulfate deposition, one in the Northern Hemisphere extratropics and one in the tropics[7].

### Investigation of historical sources and development of a lunar eclipse database

**European and Middle Eastern lunar eclipse observations.** We extensively reexamined hundreds of annals and chronicles, written in the twelfth and thirteenth centuries CE, searching for references to lunar eclipses. For Europe, contemporary sources are mostly accessible in compilations of medieval texts edited in the series of the (1) *Monumenta Germaniae Historica*, (2) *Rerum Britannicarum Medii Ævi*, (3) *Recueil des historiens des Gaules et de la France* and (4) *Rerum Italicarum Scriptores*. Descriptions or observations of lunar eclipses originate from across Europe, namely, Austria, the Czech Republic, England, France, Germany, Iceland, Italy, Sweden and Switzerland. In a few instances, lunar obscurations were observed in the Middle East by Christians engaged in the Crusades. These chronicles were found in the *Recueil des historiens des croisades*, edited and published in the nineteenth century. Most sources consulted were composed by monks or clerics and, to a lesser extent, by urban laymen. Nearly all were composed in Latin, although the vernacular was occasionally used. Furthermore, we included observations retrieved from Ukrainian and Russian chronicles in our database[51]. We also examined observations of lunar eclipses recorded in Arabic chronicles based on an existing compilation[52]. However, because none of these sources contains information on the colour of the eclipsed Moon, we did not analyse them further.

**East Asian lunar eclipse observations.** In China and Korea, eclipse records are mainly found in the *Treatise on Astronomy*, *Treatise on the Calendar* and *Treatise on the Five Elements* of the official dynastic histories[20]. For China, we reexamined the *Song Shi* (宋史), the official history of the Song dynasty (960 to 1279 CE); the *Jin Shi* (金史), the official history of the Jin dynasty (1115–1234 CE); and the *Yuan Shi* (元史) the official history of the Yuan dynasty (1261 to 1367 CE). For Korea, we investigated the *Goryeosa* (高麗史), the history of the Goryeo dynasty (936 to 1392 CE). In Japan, the astronomical records are found in a variety of works ranging from privately and officially compiled histories to diaries of courtiers and temple records[20]. We thus focused on the lunar eclipse observations compiled in the benchmark work, the *Nihon Tenmon Shiryô* (日本天文史料)[53], by Japanese astronomer S. Kanda. This work lists solar and lunar eclipses, comets and aurorae and provides original texts. The most pertinent descriptions of lunar eclipses derive from the *Meigetsuki* (明月記, 1180–1235 CE; the diary of Fujiwara no Teika) and the *Azuma Kagami* (吾妻鏡, 1180–1266 CE; *Mirror of the East*, a chronology of the Kamakura shogunate). Chinese sources are written in classical Chinese, whereas Japanese and Korean sources are written in Sino-Japanese (Kanbun) and Sino-Korean (Hanmun), respectively. However, East Asian sources rarely report information on the colour of total lunar eclipses; only the eclipse of 2 December 1229 CE could be used to estimate stratospheric turbidity.

**Assessing the reliability of historical sources.** The reliability of each eclipse observation was assessed by historical source criticism and by reference to the five-millennium catalogue of lunar eclipses (1999 BCE to 3000 CE)[21] and the eight-millennium catalogue of lunar eclipses (4000 BCE to 4000 CE)[22], which uses the most up-to-date historical values of Delta ($\Delta T$)[54,55]. We also ensured that all the eclipses reported in historical sources were visible (where relevant) in Europe, the Middle East and East Asia using the visibility maps and the local circumstances tables provided by both catalogues. Care was taken to identify second-hand reports, that is, those that a given author did not witness but instead drew information on from another source. Frequent duplication occurred in Western and Eastern Christian sources owing to common underlying source materials and the scribal practices of copying, synthesizing and piecemeal updating of annals and chronicles. A table listing all total lunar eclipses records for which information about the colour of the Moon could be retrieved, and providing summary context for the historical sources investigated, is presented in Supplementary Dataset S1. An extended version of Supplementary Dataset S1 containing all descriptions of total lunar eclipses from Europe, the Middle East and East Asia considered in this study (with or without reference to colour), along with the eclipse visibility maps, can be accessed from the Zenodo repository: https://doi.org/10.5281/zenodo.6907654.

### Four-step procedure to refine the dating of HMP eruptions

To contrain the timing of HMP eruptions, we developed a four-step methodology based on analysis of historical sources (step 1); global aerosol simulations and observations (step 2); tree-ring-based temperature reconstructions (step 3); and integration of results of steps 1–3 (step 4). A more detailed breakdown of each step and a specific illustration of the procedure using the well-known example of the 1815 CE Tambora eruption are provided below.

**Deriving SAOD from accounts of total lunar eclipse coloration and brightness.** During a total lunar eclipse, as the Moon passes through the shadow of the Earth, it is partially illuminated by light refracted by the atmospheric limb. The spectrum of the refracted sunlight is influenced by scattering and absorption. Rayleigh (molecular) scattering is stronger at shorter wavelengths, least affecting orange or red-coloured light. When the stratosphere is little perturbed, the eclipsed Moon thus tends to appear copper to deep red. With a turbid stratosphere, scattering of visible light is strongly enhanced at all wavelengths, diminishing transmission through the atmospheric limb such that the Moon in eclipse appears dark(er). In extreme cases, it can appear to vanish almost completely[16–18,36,56,57]. The colour and luminosity ($L$) of the eclipsed Moon were rated according to the Danjon scale[35], which runs from $L = 0$ to $L = 4$:

- $L = 0$: very dark eclipse. Moon almost invisible, especially at mid-totality.
- $L = 1$: dark eclipse, grey or brownish in coloration. Surface detail is difficult to distinguish.
- $L = 2$: deep-red or rust-coloured eclipse. Very dark central shadow, whereas outer umbra is relatively bright.
- $L = 3$: brick-red eclipse. Umbral shadow usually has a bright or yellow rim.
- $L = 4$: very bright copper-red or orange eclipse. Umbral shadow has a bluish, very bright rim.

This scale was designed specifically to estimate the brightness of the Moon with the naked eye, which is well suited for our purpose because no high-resolution, technical aids existed in the twelfth and thirteenth centuries. All descriptions of lunar eclipses retrieved and assessed as credible (step 1.1) were rated using the Danjon scale (step 1.2). Note that accounts referring to penumbral and partial eclipses were excluded from analysis as only total lunar eclipse observations are suited to this method[16,17]. The most common adjectives describing lunar eclipses in medieval texts are 'rubeus-a-um' and 'sanguineus-a-um', meaning 'red' and 'blood-coloured', respectively; lunar eclipses so described were rated $L = 3$. A Danjon scale value $L = 4$ was attributed only if the eclipsed Moon was described as exhibiting intense and various colours, such as in this example by the English monk Bartholomew de Cotton of an eclipse on 22 October 1287 CE: "Eodem anno luno in plenilunio visa est crocei, rubei ac varii colori" ("The same year, during the full Moon, the Moon exhibited yellow, red and many other colours"). A Danjon value $L = 0$ was only attributed if the author specifically noted that the Moon had become invisible or extremely dark during the eclipse.

**Converting Danjon-scale luminosity estimates to SAOD.** To estimate the quantity of volcanic aerosols in the stratosphere and the related attenuation of incoming solar radiation, we converted Danjon's $L$ values derived for each lunar eclipse into SAOD values following existing conversion approaches by refs. [16–19] (step 1.3). The analysis of 46 lunar eclipses that occurred between 1880–1888 and 1960–2001 CE showed that Danjon values of $L = 4, 3, 2$ and $1$ can be closely associated to SAOD values of 0, 0.01, 0.02 and 0.04, respectively, and that an SAOD value of 0.1 or greater is needed for a dark total lunar eclipse ($L = 0$) to occur.

**Constraining eruption dates using global aerosol simulations and satellite observations.** To determine the period ($T_{dark}$) when SAOD exceeded 0.1, that is, conditions for a dark total lunar eclipse (step 2.1), we used four (five for Samalas) SAOD time series. For UE1–UE6, we extracted SAOD time series around the 1883 CE Krakatau and 1991 CE Pinatubo eruptions from the Sato/GISS dataset[37]. This dataset, based on satellite observations, ground-based optical measurements and volcanological evidence, reports SAOD at 550 nm since 1850 CE. We also extracted SAOD time series for the Pinatubo eruption from the Global Space-based Stratospheric Aerosol Climatology (GloSSAC v2) dataset[38], which spans the period 1979–2018 CE. As observational data are unavailable before the mid-nineteenth century, we estimated the residence time of volcanic stratospheric aerosols for each eruption (UE1–UE6) from the eVolv2k database[4]. In the case of the 1257 CE Samalas eruptions, we also relied on results of the IPSL climate model (IPSL-CM5A-LR)[40], as it treats aerosol microphysics and has been validated for the well-observed case of the 1991 CE Pinatubo eruption[58].

Each SAOD time series covers a 56-month time window (4 months before the eruption and 52 months after). For eruptions releasing more sulfur than Pinatubo, substantial uncertainties remain about the residence time of aerosols in the stratosphere[44], and the time window for SAOD ≥ 0.1 differs depending on the aerosol model selected (see Methods). Accordingly, for each month within the 56-month time window, we determined the probability for the SAOD = 0.1 threshold to be exceeded on the basis of agreement between time series. Probability was defined as 'very high' if all the time series (four for UE1–UE6, five for Samalas) indicated exceedance of the threshold in a given month. Likewise, probability was considered 'high' or 'medium' if at least three or two, respectively, datasets agreed and as 'low' if only one dataset indicated threshold exceedance (step 2.2). SAOD time series provide, for each eruption, the most probable time interval [Month min: Month max] during which a dark lunar eclipse can be observed after an eruption. Reciprocally, we can assume that the most probable eruption date falls within the time interval [Month max: Month min] before the date of the dark lunar eclipse (step 2.3).

**Constraining eruption dates using large-scale tree-ring reconstructions.** Abrupt cooling detected in large-scale tree-ring temperature reconstructions has provided independent corroboration of the dating of climatically important volcanic eruptions[59,60]. Here we used the NVOLC v2 (ref. [7]), Sch2015 (ref. [41]) and N-TREND2015 (ref. [42]) reconstructions to refine the dating of HMP eruptions (step 3.1). Following the largest eruptions for which event dates are known (that is, Huaynaputina in February 1600 CE, Parker in January 1641 CE, Tambora in April 1815 CE, Krakatau in August 1883 CE and Pinatubo in June 1991 CE), trees recorded a cooling in Northern Hemisphere summer (June–August, or JJA) temperatures in 1601, 1641, 1816, 1884 and 1992 CE, that is, starting at 17, 6, 14, 10 and 12 months after each eruption, respectively. Accordingly, we consider that cooling observed less than 3 and more than 24 months after an eruption cannot be confidently attributed to volcanic forcing. In agreement with the existing literature[60,61], we assume that peak cooling associated with a volcanic eruption occurs between 6 and 18 months after the eruption, with a highest probability between 9 and 15 months (step 3.2).

**Combining evidence for best estimates of eruption timing.** In the last steps of our procedure, we estimated the most probable eruption time windows by combining the results from observed and simulated SAOD time series, lunar eclipse (step 2.3) and tree-ring records (step 3.2) using a decision matrix developed as step 4.1 and presented in Extended Data Fig. 1. On the basis of this matrix, we thus considered that an eruption most probably occurred during time windows for which SAOD time series, lunar eclipse and tree-ring records indicate consistently high probabilities (step 4.2).

### Testing the four-step procedure using the emblematic 1815 CE Tambora eruption

To test the robustness of our dating approach, we use the well-dated Tambora eruption that occurred on 5 April 1815 CE (Extended Data Fig. 1). Several contemporary sources reported the occurrence of a total lunar eclipse on 9–10 June 1816 CE, among which are the observations made by Capel Lofft (1751–1824 CE) in Ipswich (England) and published in the *Monthly Magazine* in 1816 CE (step 1.1). The darkness of this eclipse impressed contemporary astronomers and was rated $L = 0$ on the Danjon scale[17,62] (step 1.2). Using the conversion scale proposed by refs. [16–19] (step 1.3), we assume that SAOD had exceeded 0.1 on 9–10 June 1816 CE, that is, 14 months after the Tambora eruption. SAOD observations[37,38] and simulations[39,40] show that the highest probability of SAOD to exceed 0.1 is between 3 and 20 months after an eruption. Therefore, we assume that the most probable eruption time window falls within 3–20 months before the dark lunar eclipse of 9–10 June 1816, that is, between December 1814 and March 1816 CE (step 2.3). Then, we use the abrupt cooling recorded in Northern Hemisphere tree-ring-based temperature reconstructions to refine the time window determined in step 2.3. Peak cooling is generally detected in tree-ring records between 9 to 15 months after a large volcanic eruption. The strong cooling observed in summer 1816 CE in Northern Hemisphere tree-ring reconstructions[7,41,42] therefore points to an eruption occurring between April and October 1815 CE (step 3.2). Finally, we combine probabilities of occurrence estimated from steps 2.3 and 3.2 using a decision matrix (step 4.1) and estimate that the Tambora eruption most probably occurred between May 1815 CE and August 1815 CE (step 4.2). Our estimate thus aligns closely with the actual date of the Tambora eruption (April 1815 CE) and confirms the robustness of our approach, as well as its applicability in the case of HMP eruptions.

### Four-step-procedure: uncertainties, caveats and scope for improvement

As with any other methods, the procedure presented in this study to constrain the timing of HMP eruptions comes with several limitations.

We address these caveats in the following sections but also present several avenues of research to further refine our estimates.

**Challenges in dating volcanic eruptions using ancient lunar eclipse records.** Historical observations of lunar eclipse brightness are recognized as a valuable proxy for SAOD following large volcanic eruptions[7,16–19]. However, precautions are necessary to use this proxy appropriately:

i. Only total lunar eclipses are suitable. Partial and penumbral eclipses cannot be used for reliable estimates of stratospheric turbidity[16].

ii. Totality should preferably have been observed in good weather conditions (that is, clear, dark sky), not too near the horizon and not too close to dawn or dusk[16].

iii. The physical appearance of the Moon during totality must be explicitly described and the colour of the eclipsed disk indicated.

iv. Reports should be contemporary with the event and preferably by an eyewitness. These conditions are not always met for the medieval sources available (see Supplementary Dataset S1 for more information).

v. Our study suggests that only lunar eclipses occurring within about 20 months of an eruption are useful for dating purposes and for discriminating the tropospheric versus stratospheric transport of volcanic aerosols. At a given location, the interval between two successive total lunar eclipses ranges between 6 months and 3–4 years (refs. [17,63]). The irregular occurrence of total lunar eclipses can therefore prevent the dating of a volcanic eruption if the eclipse occurs outside this 20-month period.

vi. Careful treatment and interpretation is required when studying historical reports of lunar eclipses[64,65], as some descriptions may be too brief or cryptic to provide useful information on stratospheric turbidity and potentially lead to erroneous interpretations. One such example is the lunar eclipse of November 1258 CE recorded in the *Azuma Kagami* (吾妻鏡, vol. 5, pp. 625)[66]:

Shōka 2.10.16

"Clear in the morning. After the hour of the Snake [9 am–11 am], heavy rain and flooding. Houses were swept away and people drowned. At the hour of the Horse [11 am–1 pm] the weather began to clear. During the hour of the Rat [11 pm–1 am] the Moon was eclipsed; it was not properly visible" (see Supplementary Dataset S1).

The description of this event is brief and ambiguous, making it difficult to confidently classify it as a dark lunar eclipse. This account was written several decades after the event and is based on an earlier source that is now lost. As a consequence, we did not attribute any luminosity value to this account.

Despite these challenges, lunar eclipses so far represent the only proxy providing a direct and precise estimate of past atmospheric perturbation by volcanic aerosols. By contrast, sun dimming—references to which have been repeatedly used to identify volcanic dust veils[2,67–74]—can only rarely be dated with comparable accuracy and can also be mistakenly identified (when originating from solar eclipses or solar haloes[75,76]).

**Uncertainties in simulating the time evolution of stratospheric aerosols in climate models.** An important step in our study is the estimation of a timespan after an eruption in which we expect the stratospheric aerosol to be sufficiently optically thick to cause dark lunar eclipses. This interval, $T_{dark}$, is computed as the interval when the SAOD exceeds 0.1. We produce a probabilistic estimate of $T_{dark}$ from a combination of observed and simulated global mean SAOD time series. The use of models is necessary because some eruptions included in the study generated much greater SAOD than the largest eruptions of the modern period for which good observations are available. However, this introduces large uncertainties in the time evolution of stratospheric aerosol for the largest sulfur yields. After an initial growth period, SAOD from recent eruptions decays approximately exponentially with time[1]. If this behaviour holds for larger eruptions, then the period during which dark eclipses may occur would lengthen for larger eruptions. On the other

hand, models including microphysical processes suggest that larger eruptions produce larger sulfate aerosol particles with consequently shorter stratospheric residence[77–79]. If so, it would imply shorter $T_{dark}$ periods. This complexity is reflected in the wide spread in simulated SAOD by an ensemble of state-of-the-art aerosol models in coordinated simulations of the 1815 Tambora eruption[80].

Our analysis considers this uncertainty in SAOD evolution for large sulfur yields. The eVolv2k SAOD time series is produced with the EVA model, which is based on observations of the 1991 Pinatubo eruption and uses only a simple variation of the SAOD decay timescale with eruption magnitude. Comparison of eVolv2k SAOD with the comprehensive aerosol models of the Tambora simulations (Fig. 3 in ref. [80]) shows that the $T_{dark}$ timespan from eVolv2k is comparable with that obtained from models producing the most enduring aerosol perturbations. The large SAOD and aerosol lifetime model differences reflect uncertainties at present about processes for aerosol formation and transport among current state-of-the-art models[44,81]. The eVolv2k SAOD time series depends on the estimated VSSI of each eruption; accordingly, the eVolv2k SAOD time series represents an upper limit for the $T_{dark}$ interval. By contrast, the IPSL model, based on free-running aerosol microphysical processes[82], is seen in the Tambora ensemble to produce one of the fastest SAOD decays and therefore shortest $T_{dark}$ intervals. This behaviour reflects rapid growth of stratospheric sulfate aerosols and greatly enhanced gravitational settling. The IPSL results therefore provide an estimate of the lower limit on $T_{dark}$. By including these approximate upper and lower limits on $T_{dark}$ in our analysis, we incorporate the uncertainty in stratospheric aerosol evolution and propagate it into our final constraints on eruption timing.

**Caveats on the use of tree-ring-based proxies.** Tree rings have been used for nearly 40 years to evaluate the timing and assess the environmental consequences of volcanic eruptions[48,83–92]. Yet, detecting volcanic signals in tree-ring records comes with several challenges. Several tree-ring parameters have been used to study past volcanic events. One of these is ring width (RW), the annual increments of the growth rings in wood. Trees typically respond to volcanically induced cooling by producing narrow RW. However, the use of this parameter to date and quantify cooling induced by large volcanic eruptions is debated. RW is known to be strongly influenced by biological persistence, which can lead RW temperature-based reconstructions to underestimate, lag and exaggerate the duration of post-eruption cooling[83,90,93,94]. The tree-ring community has therefore advocated for the use of another parameter called 'maximum latewood density' (MXD), regarded as the "gold standard of high-resolution paleoclimatology for temperature reconstructions"[95]. MXD, obtained from high-resolution density profiles measured by X-ray radiodensitometry, is indeed less prone to biological memory and responds more rapidly to climate extremes. However, unfortunately, rather few MXD chronologies extend to before 1300 CE[83,94]. Moreover, volcanic eruptions do not result in globally uniform summer cooling[9,83,90]. Depending on the sulfur yield of the eruption, its latitude and season, prevailing climate conditions and internal variability, some regions will cool, whereas others experience little change[83]. When tree-ring chronologies from various regions are averaged to produce hemispheric temperature reconstructions, the volcanic signal can thus become muted and harder to detect[83].

We used the NVOLC v2 (ref. [7]), Sch2015 (ref. [41]) and N-TREND2015 (ref. [42]) reconstructions to refine the dating of HMP eruptions. The NVOLC v2 reconstruction is composed of 25 tree-ring chronologies (12 MXD and 13 RW chronologies), Sch2015 relies on 15 MXD chronologies distributed across the Northern Hemisphere extratropics, whereas N-TREND2015 is based on a network of 54 records (11 RW, 18 MXD and 25 mixed series composed of RW, MXD and blue intensity records). These reconstructions were selected because they integrate a substantial number of MXD records. We did not use recently published reconstructions that rely exclusively on RW records[96].

Overall comparison of the three Northern Hemisphere reconstructions shows good agreement for UE1–UE6 (Fig. 3 and Extended Data Fig. 2). For UE2, we note that the maximum peak cooling is observed in 1171 CE in N-TREND2015 and one year later in Sch2015 and NVOLC v2. For the 1257 CE Samalas eruption, Sch2015 and N-TREND2015 show less pronounced cooling, which—however—remains in the range of uncertainties of the NVOLC v2 reconstruction. We recognize three sources for these discrepancies. (1) Differences in the tree-ring networks used in the different studies. NVOLC v2 includes only chronologies that encompass the full period between today and the twelfth century, whereas Sch2015 and N-TREND2015 include shorter series as well. (2) Differences in the transfer function used. NVOLC v2 is based on a nested principal component regression—to gradually adjust to a changing number of available proxy records[48,97]—combined with a 1,000-iteration bootstrap approach enabling calculation of uncertainties associated with the reconstruction. By contrast, Sch2015 and N-TREND2015 are based on a scaling approach. (3) The climatological datasets used for calibration. NVOLC v2 uses monthly mean (1805–1972 CE) JJA temperature anomalies (40–90° N) from the recently released Berkeley Earth Surface Temperature (BEST) dataset[98]. Schneider et al.[41] calibrated their proxy records against monthly mean JJA temperature anomalies (1901–1976 CE) derived from the 5° × 5° CRUTEM4v network[99] (30–90°). Wilson et al.[42] scaled their proxy record to CRUTEM4v (40–75° N) MJJA land temperatures during the 1880–1988 CE period. The combination of these differences in tree-ring networks, transfer functions, climatological reference datasets, calibration periods as well as target season inevitably results in differences in the cooling magnitudes for specific events.

Our study relies on state-of-the art reconstructions that efficiently capture post-volcanic summer cooling but several avenues may improve the peak cooling detection and refine the timing of HMP eruptions:

i.  Improvement of the spatial coverage of the millennium-long MXD network with new data from poorly represented regions of the globe.

ii. Quantitative wood anatomy (QWA). Edwards et al.[100,101] attempted to narrow the period of peak cooling associated with the Laki eruption to late summer 1783 CE using cellular-scale tree-ring proxy measurements. These findings contrast with MXD reconstructions that suggest that the entire 1783 CE summer was exceptionally cold and with tree RW reconstructions that mute the cooling. Such results indicate that QWA data can identify more precisely than tree RW and MXD records the timing of peak cooling following volcanic eruptions within the growing season. Including QWA analyses for the HMP eruptions in the four-step procedure proposed in this study may further refine estimates of eruption timing. Despite promising results, highly resolved QWA is at its early stage. Further, because highly resolved wood anatomical parameter chronologies are costly and labour-intensive, it is unlikely that an operational (Northern Hemisphere) network of QWA records will be available soon.

### Discriminating between tropospheric and stratospheric aerosol layers

Most of the refracted sunlight that illuminates the eclipsed Moon passes between 5 and 25 km above the surface of the Earth[19]. Upper tropospheric aerosols (5–10 km) may affect the brightness of the Moon[18] but their residence time is on the order of a few weeks[1,18]. Dark lunar eclipses thus more probably indicate high turbidity of the stratosphere after large volcanic events[16–18]. We thus assume that lunar eclipses of reddish or coppery colour (that is, with an *L* value >1) observed in the aftermath of HMP eruptions indicate that aerosol veils were mainly confined to the troposphere and probably had limited climatic impacts. The robustness of our approach was assessed by comparing our results with sulfur isotope records (Δ$^{33}$S) from Dome C (Antarctica)[3], which have proven a valuable proxy to distinguish between eruptions whose plumes reached the stratosphere at or above the ozone layer and those that remained below[3,102–107].

## Data availability

The historical data underlying this study can be found in Supplementary Dataset S1 and is available on Zenodo at https://doi.org/10.5281/zenodo.6907654. The tree-ring-based reconstructions can be downloaded at https://doi.org/10.5281/zenodo.3724674. Ice-core data can be retrieved from the following link and repository: https://doi.org/10.1038/nature14565 and https://doi.org/10.1594/WDCC/eVolv2k_v2. SAOD time series can be obtained from the following links: https://data.giss.nasa.gov/modelforce/strataer/ and https://doi.org/10.1594/WDCC/eVolv2k_v2. Source data are provided with this paper.

## Code availability

The codes used in the data processing is available on Zenodo at https://doi.org/10.5281/zenodo.6907654.

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

**Acknowledgements** S.G., C.C., M.K. and M. Stoffel were supported by the Swiss National Science Foundation Sinergia Project CALDERA (CRSII5_183571). S.G. acknowledges A. Harrak (Department of Near and Middle Eastern Civilizations, University of Toronto), F. Espenak (NASA Goddard Space Flight Center), F. Hierink (Institute for Environmental Sciences, University of Geneva) and P. Souyri (Department of East Asian Studies, University of Geneva) for providing advice on the manuscript. F. Lavigne was supported by Institut Universitaire de France (IUF, Academic Institute of France). M.K. received funding from the EUR IPSL – Climate Graduate School project, managed by the ANR within the "Investissements d'avenir" programme under reference ANR-11-IDEX-0004-17-EURE-0006. F. Ludlow received funding from an Irish Research Council Starting Laureate Award (CLICAB project, IRCLA/2017/303). F. Ludlow and Z.Y. also received funding from a European Research Council (ERC) Synergy Grant (4-OCEANS; grant agreement no. 951649) under the European Union's Horizon 2020 research and innovation programme. M. Sigl received funding from the ERC under the European Union's Horizon 2020 research and innovation programme (grant agreement no. 820047). This paper is a product of the Volcanic Impacts on Climate and Society (VICS) working group.

**Author contributions** S.G. designed the research, with inputs from C.C., M. Stoffel and F. Lavigne. S.G. investigated European, Russian and Middle Eastern historical sources. S.G., P.S.A., N.H. and T.M. investigated Japanese historical sources. Z.Y. and S.G. investigated Chinese historical sources. S.G. and Z.Y. investigated Korean historical sources. S.G. analysed historical sources, with contributions from Z.Y., P.S.A. and F. Ludlow. S.G., C.C., M. Sigl, C.O. and M.T. contributed to ice-core and tree-ring data interpretation. S.G., C.C., M.K. and M.T. contributed to aerosol model simulations interpretation. The manuscript was written by S.G., C.C., M. Stoffel and C.O., with contributions from M.K., F. Ludlow, F. Lavigne, M. Sigl and M.T.

**Funding** Open access funding provided by University of Geneva.

**Competing interests** The authors declare no competing interests.

**Additional information**
**Correspondence and requests for materials** should be addressed to Sébastien Guillet.

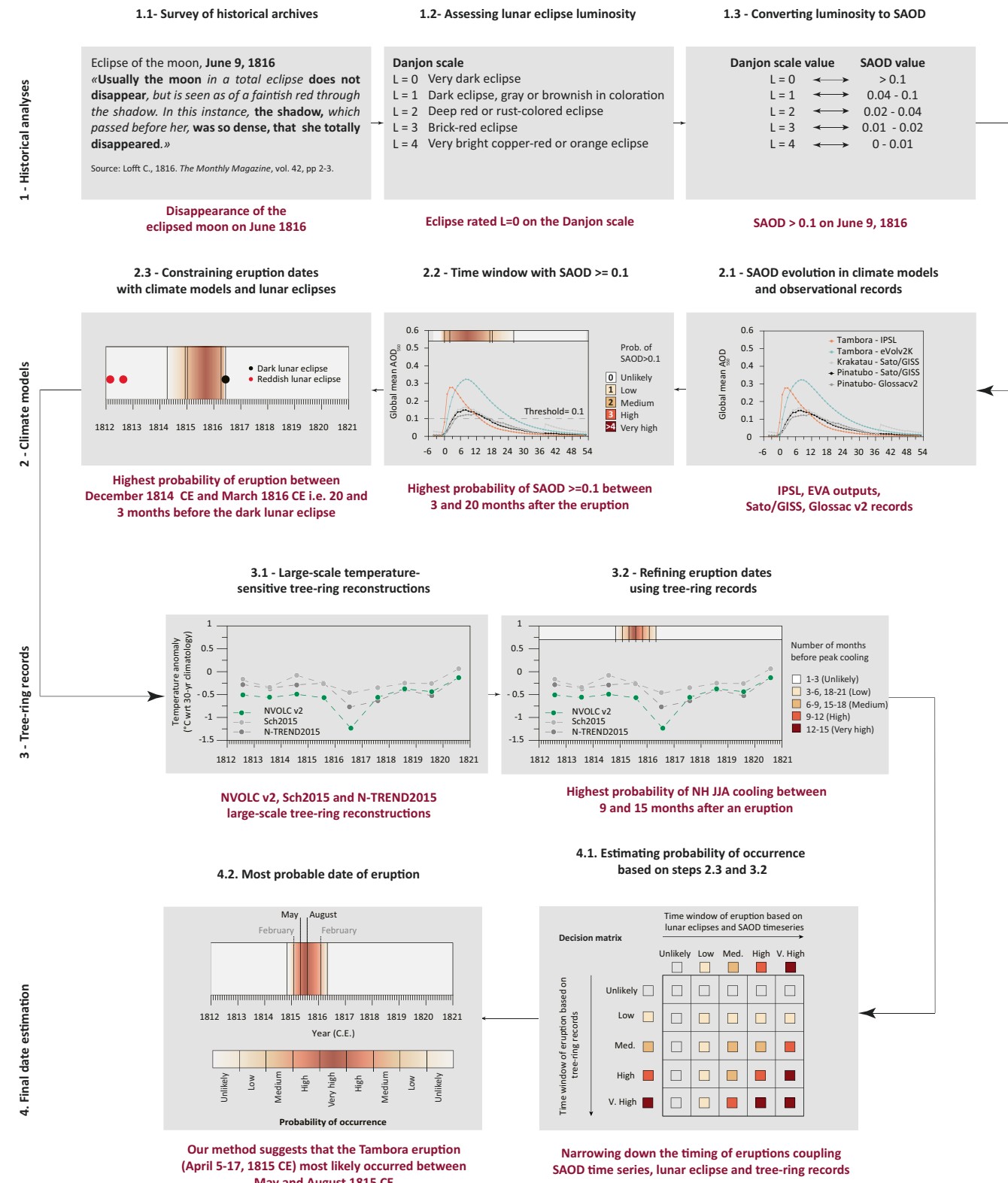

**Extended Data Fig. 1 | Testing the four-step procedure using the emblematic 1815 ᴄᴇ Tambora eruption.** A detailed description of the approach can be found in Methods.

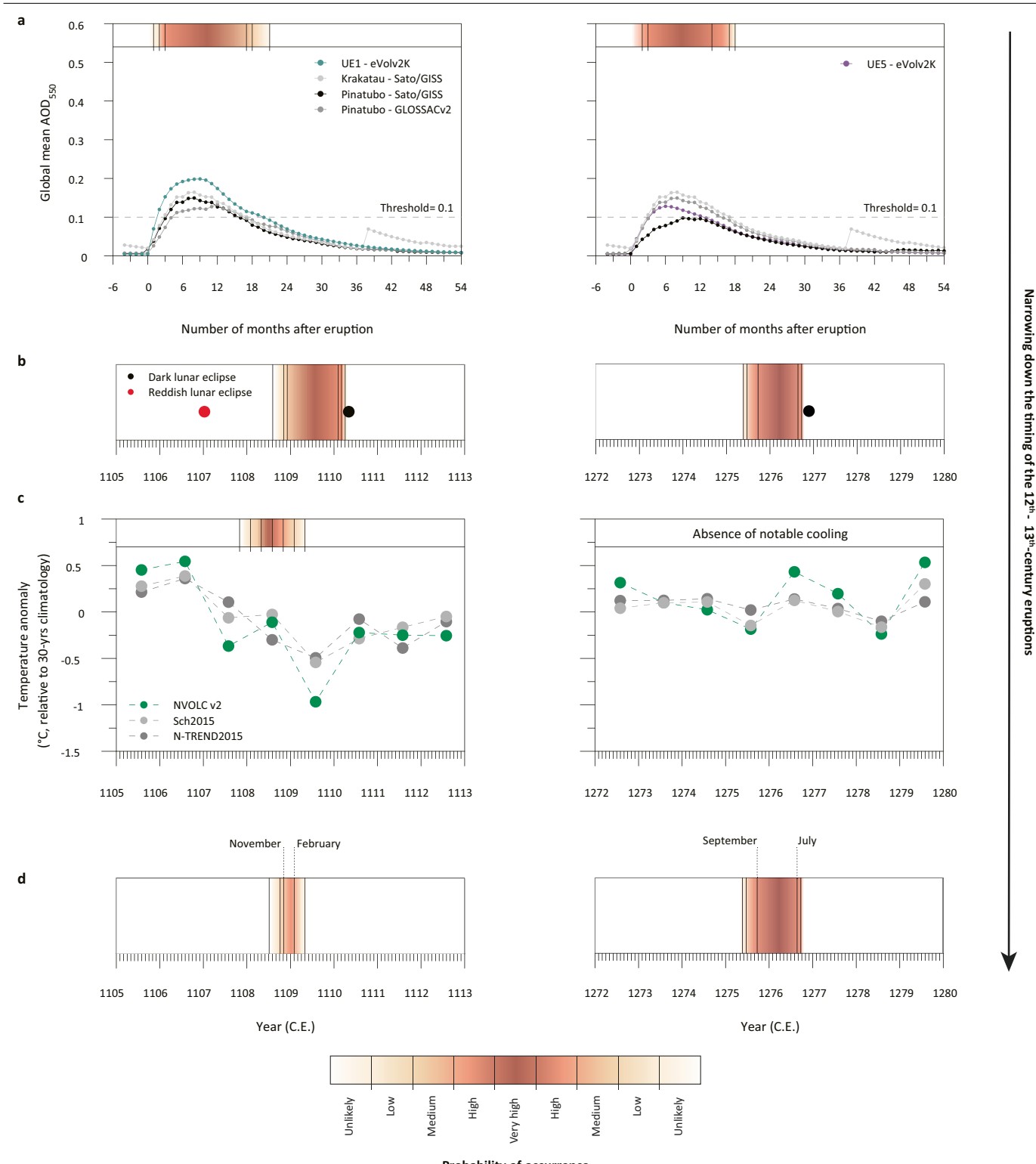

**Extended Data Fig. 2 | Potential dates of HMP (1100–1300 CE) volcanic eruptions (UE1 and UE5) using a four-step procedure. a**, Residence time of volcanic stratospheric aerosols and time windows with SAOD exceeding about 0.1. The residence time of aerosols is based on global mean SAOD_550 time series from the Sato/GISS[37] and GloSSAC v2 (ref. [38]) datasets (for the 1883 CE Krakatau and 1991 CE Pinatubo eruptions) and simulated by the EVA forcing generator[4,39] (for UE1 and UE5). Probability of occurrence of HMP eruptions based on the timing of dark lunar eclipse dates (**b**) and tree-ring records[7,41,42] (**c**). **d**, Integration of **b** and **c** to estimate the most probable time windows for the UE1 and UE5 eruptions.

**Extended Data Table 1 | Number and percentage of total lunar eclipses potentially visible and recorded in historical sources in Europe during the twelfth and thirteenth centuries**

| List of total lunar eclipses not recorded or found in European sources | Number of total lunar eclipses visible in Europe in the 12th-13th centuries | Number of total lunar eclipses recorded in the 12th and 13th centuries | Number of unrecorded total lunar eclipses in the 12th and 13th centuries | Percentage of total lunar eclipses recorded in European sources | Number of lunar eclipses for which we have information about the brightness of the moon | Number of lunar eclipses for which we have no information about the brightness of the moon | Number of lunar eclipses above L=1 |
|---|---|---|---|---|---|---|---|
| 1121-Sep-28<br>1139-Apr-16<br>1150-Sep-08<br>1157-Oct-19<br>1183-Jun-07<br>1190-Jul-18 | 32 | 26 | 6 | 81 | 17 | 9 | 15 |
| 1200-Dec-22<br>1201-Jun-18<br>1208-Jul-29<br>1237-Jan-12<br>1244-Aug-19<br>1295-May-30<br>1298-Sep-21 | 32 | 25 | 7 | 78 | 18 | 7 | 15 |
| 12 - 13th Century | 64 | 51 | 13 | 80 | 35 | 16 | 30 |

**Extended Data Table 2 | Description of lunar eclipses observed after the eight largest eruptions since 1600 CE (refs. [16,17,19,62,108–120])**

| Volcano | Huaynaputina | Parker | Laki | Tambora | Krakatoa | Katmai | Agung | El Chichon | Pinatubo |
|---|---|---|---|---|---|---|---|---|---|
| Eruption date | 19.02.1600 | Dec 1640 - Jan 1641 | Jun 1783 – Feb 1784 | 10.04.1815 | 27.08.1883 | Jun - Oct 1912 | 17.03.1963 | 03.03.1982 | Jun-91 |
| Lunar Eclipse 1 - Date | 09.12.1601[108,109] | 14.04.1642[109,110] | 10. 09. 1783[16] | 09.06.1816[17,62] | 04.10.1884[109,111-113] | 22.03.1913[114] | 30.12.1963[116,117] | 30.12.1982[118,119] | 09.12.1992[19] |
| Months after the eruption | 20 | 15 | 4 | 14 | 14 | 9 | 10 | 9 | 18 |
| Luminosity - Danjon Scale | 0 | 0 | 2 | 0 | 0 | 0 | 0-0.2 | 0-1 | 0 |
| Lunar Eclipse 2 - Date | - | - | | - | - | 15.09.1913[115] | 24.06.1964[116,117] | - | 28.11.1993[120] |
| Months after the eruption | - | - | | - | - | 15 | 15 | - | 29 |
| Luminosity - Danjon Scale | - | - | | - | - | 0 | 0 | - | 3 |
| Lunar Eclipse 3 - Date | - | - | | - | - | - | 19.12.1964[116,117] | - | - |
| Months after the eruption | - | - | | - | - | - | 21 | - | - |
| Luminosity - Danjon Scale | - | - | | - | - | - | 1.6 | - | - |