## [Peer Review File · Nature]

Manuscript Title: Lunar eclipses illuminate timing and climate impact of medieval volcanism

Reviewer Comments & Author Rebuttals

Reviewer Reports on the Initial Version:

Referee #1 (Remarks to the Author):

I write as a medieval historian and thus with a consciously limited area of expertise in relation to this paper. Overall I found the article extremely interesting and, as far as I am able to judge, compelling in its methodology and conclusions. The handling of sources, and the source-critical questions applied, are entirely appropriate and produce good results. My recommendation is that it should be published.

In relation to the handling of medieval European sources I do have a few comments, which are intended to be helpful. The first is that clear understanding of the exact conditions required to produce a total lunar eclipse was available to experts in western Europe in the twelfth century, but that many monastic institutions continued to rely on the more general models provided in works on computus. The experts could use the 'canons' or instructions accompanying Arabic astronomical tables, which were translated into Latin in C12. However, as the article says, accessible and wide-ranging treatises on astronomy and cosmology, like Sacrobosco's, only appeared in the thirteenth century. The relevance of this is that at least some twelfth-century sources appear to give dates for lunar eclipses calculated from old computistical texts rather than from direct observation, and that this may account for some of the dating errors noted in Table S1.

Another small point which might be mentioned is that accurate observation of the 'age' of the Moon was important not only for monastic time-keeping but also for the correct identification of the paschal full moon each year. McCluskey, who is cited, discusses this in chapter 5.

A minor point about terminology is that the article talks of evidence from 'archives' - but this would usually mean the original manuscripts. As published texts (and translations) have been used it might be clearer to say 'sources' at these points.

I am not clear whether dataset S1 will be published as it stands. If it will be then further proof reading will be needed as there are some typos. I also wondered whether it was necessary to repeat the full information about each author and chronicle every time the same chronicle was cited.

These comments are intended to be helpful and do not detract from my overall recommendation.

Referee #2 (Remarks to the Author):

Referee Report for MS 2021-10-15899

In this manuscript, the authors have allegedly surveyed historical records for the lunar eclipses to identify dark total lunar eclipses as indications for stratospheric turbidity of extremely large volcanic

eruptions in 1100 -- 1300 CE. This is an interesting attempt which would potentially benefit the scientific and historical communities. However, according to their supplementary materials, in most cases, they seem to have consulted not the original historical documents but the secondary catalogs for lunar eclipses unlike what is emphasised in their manuscript. This is especially the case with the non-European sources, which share mistakes of the previous catalogs, play little role in their discussions, and should be omitted from their manuscript. Their astronomical analyses owe much to the NASA website, whereas they have to philologically reconsider the observational sites, entirely renew their parameter with the state-of-the-art Earth's rotation variability. Overall, in my humble opinion, thorough revisions are needed before further consideration on this manuscript for publication in Nature.

Major Comments

1. The eclipse magnitude

Here, they have accurately computed the eclipse timing and magnitude in the supplementary table, on the basis of the Five Millennia Catalog of Lunar Eclipses. However, I have to raise two concerns. Firstly, their reference is misleading. They have cited this catalog as if a published book (see Ref 35), whereas this is actually NASA's online catalog. This is not acceptable. More seriously, since this catalog's release, the ΔT has been significantly updated (e.g., Stephenson et al., 2016). The authors must recalculate all these parameters based on the latest ΔT .

2. Observational sites

This is closely relevant with the source reliability, which the authors have described the source reliability in the supplementary materials. This is a good move. However, it is not clear what they have learnt from this entry, at least in the main text. In fact, a recent case study has identified a source report of an atmospheric optics report in the Anglo-Saxon Chronicle with the celestial phenomenon in Northern France, after tracing chains of duplications (DOI: 10.5194/hgss-11-81-2020). The supplementary material does not allow us to see such philological traces. Rather, it seems the authors have generally assumed the observational site as chroniclers' residential area or somewhere nearby. The authors need to be more cautious on the identification of the observational sites, as the authors are keen to derive the eclipse parameters in their supplemental datasets.

3. Analyses of the historical records (especially for non-European records)

The authors claimed that they have reexamined the original sources. However, this seems slightly dubious, especially for the non-European records. What they have consulted seems to be the catalogs of astronomical records. I suspect they have consulted Xu et al. (2000) [Ref. 42] for Chinese and Korean records, as their translations are virtually the same and frequently share the same mistakes. For example, some Japanese terminologies are transcribed into Chinese pronunciations. Similar issues are found throughout the East Asian and Middle Eastern eclipse records. The authors must distinguish the historical source documents from these secondary catalogs. At least, it would be excessive for the authors to claim that they have "extensively reexamined hundreds of annals and chronicles". They should explicitly clarify that they have mostly consulted not the original historical records but the catalogs of historical lunar-eclipse records.

4. East Asian records

While I appreciate the authors' reanalyses on the European eclipse records, I suspect their

reanalyses on the non-European records are mostly not on the basis of the original historical sources but on the secondary catalogs (see comments below). Their translations are extremely similar to what has been published in Xu et al. (2000) and Yuasa (2007, 2010) [NB: probably not cited in the main text] for example (see my comments below). They have also alleged that Korean records are written in Classic Chinese (P9L385), whereas this was actually Sino-Korean. If so, why are they transcribing 高麗史 not as Gaolishi but as Koryo-sa? Koryo-sa could be better transcribed as Goryeo-sa, following discussions in Korean studies (e.g., Ref. 43). Honestly, apart from some exceptions, these non-European records have played little role in their manuscript. Therefore, I think the authors should remove their datasets from the non-European records, which are not quite original. They include not only their records from East Asia but also those from Islamic Middle East (e.g., Stephenson and Said, 1997) and Russia (e.g., Vyssotsky, 1949). Instead, they should pinpoint the citation for the dark lunar eclipse in 1229.

5. Referencing in the Dataset S1 (Chinese records)

They have cataloged the East Asian records in Dataset S1. Especially for the Chinese records, they have referenced the sources only up to the chapter number without clarifying which page(s) in which critical edition(s) they have actually consulted. Instead, they have provided a link to Wikisource and C-text, as a convenient online source. This is not acceptable, as these texts frequently involve some weird transcriptions. They have to rely not on these unreliable online sources but on appropriate critical editions.

6. Romanizations of the Chinese terms in the Dataset S1

Their romanizations lack accents too. The Chinese language has four accents. Even for the same syllable, different accents will provide different meanings. If they insist in using Chinese sources, they have to put appropriate accents to their transcriptions.

7. Referencing in the Dataset S1 (Japanese records)

The original descriptions are mostly missing. They have to fill this entry or completely abandon the usage of the Japanese sources. For Gyokuyo and Azuma Kagami, they have specifically relied upon Yuasa's articles. Here, they must rely on an appropriate critical edition. Yuasa explicitly declared that he used Yoshikawa Kobunkan's critical edition in 1978-1979. They relied on the English translation mostly on Xu et al. (2000), whereas they sometimes lack translations for the entries with original records (e.g., 1175-May-08, 1175-Oct-31, and 1176-Apr-26).

8. Source Description for the Japanese sources

The source descriptions were repeated many times. For example, they have repeated similar (or likely the same) notes for Azuma Kagami more than 25 times. For the readership, it is probably easier to have another supplementary table for the source documentation in a separate file. Some notes look interesting, whereas this repetition makes it extremely difficult to find which information is scientifically valuable.

9. "Dark eclipses" in the Japanese sources

For some of their notes, I have found philological difficulties. Their interpretation for the dark lunar eclipse in 1229 is convincing, with an explicit source description "The Moon's disk changed and disappeared, but only as if covered by clouds." In contrast, the one in 1258 is too speculative. Here,

they have interpreted "We could not see anything" as an unusually dark lunar eclipse. However, this is not justified as this description can be applied to another lunar eclipse which was not actually visible from Japan (Yuasa, 2007, 2010). The authors have omitted a record with the same description ("We could not see anything"), possibly because these eclipses were not visible from Japan. This will probably mislead the readership to associate this description with the 'dark eclipse'.

10. The SAOD simulation

The authors have suggested the dark eclipses would be visible 3-20 months after the eruptions, using the SAOD simulation (Extended Data). This is interesting but difficult to see and understand. It would be nicer if the authors could graphically explain this procedure more understandably.

11. The European chroniclers' general attention to the celestial events

In P3L92-102, the authors have emphasized the European chroniclers' general attention to the celestial events, citing their remarkable coverage (~85%). However, this is not quite correct. This kind of trend is totally dependent on the nature and purpose of the chroniclers (Stephenson and Green, 2002, p. 140). It is well known that the SN1054 was missed from most of the European chroniclers (Stephenson and Green, 2002). Some chroniclers have paid significant attention to the celestial events, as exemplified with John of Worcester (e.g., DOI: 10.1080/03044181.2013.798742). In contrast, some chroniclers have apparently paid limited attention to the celestial phenomena, as exemplified with Bede. Stephenson et al. (2020) have recently examined Bede's accounts and have proven omissions of many large solar and lunar eclipses visible at Jarrow (DOI: 10.1177/0021828619899188). Some celestial reports have been even derived from hearsay (DOI: 10.5194/hgss-11-81-2020). In this case, it would be safer to avoid this generalisation, assuming the reason. In short, it is also questionable how far contemporary science (e.g., their Figure 1b) has been shared throughout the chroniclers.

Minor Comments

P2L67: solar coronae => atmospheric optics.

P3L88: Astronomical and Five-Elements treatise => Astronomical treatises and Five-Elements treatises

P3L88 Japan => Japanese

P3L89: "diaries of courtiers or temple records" => I suspect Azuma Kagami is not diaries or temple records but chronicles or annals.

P3L90: "In continental Europe and in the Middle East, annals and chronicles from monasteries and towns represent the main sources." => Is this really true for the Islamic annals in the Middle East? The Arabic sources tend to be universal chronicles (See "Ta'rikh" in: Encyclopedia of Islam (ver 2), v. 10, pp. 257-302). If they still wish to claim this, they have to show appropriate references.

P4L102: The medieval East Asians frequently recorded predicted eclipses (Stephenson, 1997; Yuasa,

2007). I wonder if the authors have considered this fact when they describe the proportion of the reported eclipses.

Referee #3 (Remarks to the Author):

REVIEW: The “Dark Side of the Moon”: Constraining the history of medieval volcanism with astronomical records” by Guillet et al.

This paper presents a novel multi-proxy approach for constraining the timing of medieval volcanic events (including the use of astronomical records of lunar eclipses), with one goal being to shed light on the transition from the Medieval Warm Period to the Little Ice Age. This work provides an additional source of information for comparison to climate models, ice cores, tree rings and other proxies of volcanism, and thus can improve our understanding of such events and their implications regarding the climatic forcings of the past millennium. The paper is thus of considerable interest and potentially broad appeal.

With regards to the application of dendrochronology as a source of independent confirmation of the dating/ calibration/evaluation of the results, my main suggestion would be that the authors should mention several important caveats regarding their selection and usage of the hemispheric tree-ring reconstructions of Schneider et al. (2015, based solely on Maximum Latewood Density or MXD; 2017) and of Wilson et al. (2016, NTREND), the latter based on 11 ring width series (RW), 18 density (MXD as well as Blue Light), as well as 25 mixed series using both RW and density.

It should be noted that RW and MXD parameters can vary significantly in their cooling response to major volcanic events, and display different spectral features and potential biases, which could influence their results. Typically, RW features substantial autocorrelation or biological memory (causing a more muted, delayed response to forcing) and MXD features much less, if at all (e.g. Esper et al. 2015). For Samalas, for example, Wilson et al. (2016) note that the MXD response is more immediate, while RW shows a stronger response in year $t+1$ and their mixed record response is in $Yr\ t+2$. RW and MXD have different seasonal sensitivities; e.g. RW can even integrate conditions in the non-growing season, although this is controversial. RW also tends to have a weaker, less robust volcanic signal and hence is not ideal for evaluating volcanic response. There are also differences at lower frequencies which may be likely relevant to the MWP-LIA discussions. NTREND, for example, features a longer and warmer medieval period from 900-1170. The Schneider MXD series shows relatively modest warmth from 850-1050 and more muted multidecadal variability, and a delayed LIA onset using MXD, as compared to RW records. Tree-ring series are also typically less reliable back in time, e.g. during the critical MWP interval, due to decreased data coverage, and thus there can be less coherency both between the tree-ring series themselves and also between the tree-ring series and climate models during this important time period, emphasizing the need to collect additional data for early in the past millennium.

The authors should thus address whether or not these differences have any significance to their analysis and results.

It is also worthwhile mentioning that there are substantial differences in spatial response to volcanism depending on the prevailing atmospheric circulation, latitude and timing of the eruption, spatial biases in tree-ring data coverage (eg. towards coverage in Europe during the MWP), among other factors, that are obscured in the hemispheric scale average.

The authors should also address how their results might be impacted by multiple events/eruptions – for example the Gomagatake, Japan eruption in summer 1640 may have complicated the timing calculated for Parker in the Philippines (Jan 1640).

A number of the most relevant eruptions in their study from the more recent period were mainly from lower latitudes. Higher latitude eruptions – Laki, Katmai – might show less of a delay in timing of response in temperature reconstructions based on middle to higher-latitude northern tree-ring sites than what is shown in their study.

REFERENCES CITED

Esper, J., et al., 2015. Signals and memory in tree-ring width and density data. *Dendrochronologia* 35, 62e70.

Schneider, L., Smerdon, J. E., Pretis, F., Hartl-Meier, C. & Esper, J. A new archive of large volcanic events over the past millennium derived from reconstructed summer temperatures. *Environ. Res. Lett.* 12, 094005 (2017).

Schneider, L., Smerdon, J., Büntgen, U., Wilson, R., Myglan, V., Kirilyanov, A., Esper, J., 2015. Revising midlatitude summer temperatures back to A.D. 600 based on a wood density network. *Geophys. Res. Lett.* 42 (11), 4556e4562. [http:// dx.doi.org/10.1002/2015GL063956](http://dx.doi.org/10.1002/2015GL063956).

Wilson, R. et al. 2016. Last millennium NH summer temperatures from tree rings: Part I: The long-term context. *QSR* 134: 1-18. (NTREND).

Referee #4 (Remarks to the Author):

The manuscript provides an estimate of strong volcano eruptions based on reported lunar eclipses. As a part of the procedure, simulated stratospheric aerosol optical depths from pre-existing global climate model simulations were used for estimating the correlation of volcano emission and the rating of the appearance of lunar eclipses. The use of global model simulations for this purpose is a valid approach although it has large uncertainty since the lifetime and transport of stratospheric aerosol is very much model dependent. It would be good to add some discussion on this in the description of the use of model data.

Author Rebuttals to Initial Comments:

Referee #1 (Remarks to the Author):

I write as a medieval historian and thus with a consciously limited area of expertise in relation to this paper. Overall, I found the article extremely interesting and, as far as I am able to judge, compelling in its methodology and conclusions. The handling of sources, and the source-critical questions applied, are entirely appropriate and produce good results. My recommendation is that it should be published.

In relation to the handling of medieval European sources I do have a few comments, which are intended to be helpful. The first is that clear understanding of the exact conditions required to produce a total lunar eclipse was available to experts in western Europe in the twelfth century, but that many monastic institutions continued to rely on the more general models provided in works on computus. The experts could use the 'canons' or instructions accompanying Arabic astronomical tables, which were translated into Latin in C12. However, as the article says, accessible and wide-ranging treatises on astronomy and cosmology, like Sacrobosco's, only appeared in the thirteenth century.

Reply: This point is very well taken. The 12th century has been considered by historians as a period of scientific renaissance in Western Europe (Haskins, 1927; Burnett, 2013; Nothaft, 2017). Notably, during the first half of the 12th century, a number of mathematical and astronomical treatises were translated from Greek, Arabic and Hebrew to Latin.

These translations were important because they allowed Latin Europe to recover writings of ancient Greek astronomers like Ptolemy, but also to disseminate the new learnings gained by Arab astronomers from their own observations.

One of earliest translators was Aldebard of Bath (ca. 1180-1150 CE) who produced the first known translation of the *Zīj* (astronomical handbook) by al-Khwārizmī around 1134 CE (Burnett, 1987). He also wrote a tract explaining the use of the astrolabe (ca. 1149-1150 CE).

Another prolific translator was Gerard of Cremona (ca. 1114-1187 CE), who travelled to Toledo in Spain to learn Arabic and translated numerous Greek and Arabic astronomical works into Latin, such as the *Toledan tables* of al-Zarqālī or Ptolemy's *Almagest* (1175 CE) (Burnett, 2001).

Walcher (d. 1135 CE), prior of the Malvern abbey (Worcestershire, England) was one of the first scholars to adopt methods from Arabic science to compute lunaison tables and perform eclipse predictions, (although with some inaccuracies) (Nothaft, 2017).

Therefore – and as the referee rightly emphasises – several treatises allowing scholars to understand the motion of the Moon and the mechanisms behind of lunar eclipses were already available in the 12th century.

Our manuscript, now, not only mentions the *De Sphera* written circa 1230 CE by Johannes de Sacrobosco but also *De lunationibus* by Walcher of Mavern (1112 CE) and *De Dracone* (1120-1121 CE) (see lines: 119-121).

The relevance of this is that at least some twelfth-century sources appear to give dates for lunar eclipses calculated from old computistical texts rather than from direct observation, and that this may account for some of the dating errors noted in Table S1.

Reply: Indeed, while some mistakes can be explained by copyists' errors, it is likely that some of the mistakes found in annals and chronicles result from use of calendar tables containing errors. An interesting example can be found in Dataset S1 (see Dataset_S1_Continental_Europe, cell U45) in a report by the chronicler Fulcher of Chartres about a lunar eclipse he witnessed on 16 June 1117 CE, near Antioch. Fulcher writes that the eclipse occurred on the 13th day of the Moon while he states that a lunar eclipse should normally be observed on the 14th day. He therefore considered that the reddish Moon he observed was a sign of God, a miracle. But he didn't question the possibility that the computistical table he had with him that night might contain errors.

Another small point which might be mentioned is that accurate observation of the 'age' of the Moon was important not only for monastic time-keeping but also for the correct identification of the paschal full moon each year. McCluskey, who is cited, discusses this in chapter 5.

Reply: The referee raises a very good point. We added a sentence to the revised version of the manuscript mentioning the importance of knowing the 'age' of the Moon to identify the date of Easter full moon, which served as the key point of reference for Easter Sunday and all the other mobile feast days of the liturgical year.

A minor point about terminology is that the article talks of evidence from 'archives' - but this would usually mean the original manuscripts. As published texts (and translations) have been used it might be clearer to say 'sources' at these points.

Reply: We fully agree with the referee. We have replaced the term "archives" to "sources" throughout the manuscript for consistency.

I am not clear whether dataset S1 will be published as it stands. If it will be then further proof reading will be needed as there are some typos.

Reply: The referee is right, several typos were present in the first version of Dataset S1. We carefully checked each entry and made the appropriate changes.

I also wondered whether it was necessary to repeat the full information about each author and chronicle every time the same chronicle was cited.

Reply: We appreciate this comment and have given it careful consideration but we feel that for future researchers using the dataset it is important that all the information is kept for each entry. Although the source description looks similar for some entries, there are nuances that make them unique.

Therefore, we feel that removing the source description for some entries may actually add complexity for future users so prefer to err on the side of some redundancy.

Nonetheless, we agree with the referee that Dataset S1 is large and not easy to digest. Therefore, we created an interactive atlas, which enables the reader to display, interact and access each historical record in a more visual way. The map provides information about all the lunar eclipses that were reported in historical sources over the period 1100-1300 CE. The atlas can be accessed freely here:

<https://arcg.is/XyOKX>

And additional information can be found in the manuscript, lines 447-449.

These comments are intended to be helpful and do not detract from my overall recommendation.

References:

Burnett, C. (1987). *Adelard of Bath: an English scientist and arabist of the early twelfth century*. London: Warburg institute.

Burnett, C. (2001). The Coherence of the Arabic-Latin Translation Program in Toledo in the Twelfth Century. *Science in Context*, 14(1–2), 249–288. <https://doi.org/10.1017/S0269889701000096>

Burnett, C. (2013). THE TWELFTH-CENTURY RENAISSANCE. In D. C. Lindberg & M. H. Shank (Eds.), *The Cambridge History of Science: Volume 2: Medieval Science* (Vol. 2, pp. 365–384). Cambridge: Cambridge University Press. <https://doi.org/10.1017/CHO9780511974007.017>

Nothaft, C. P. E. (Ed.). (2017). *Walcher of Malvern, De lunationibus and De Dracone*. Turnhout, Belgium: Brepols.

Referee #2 (Remarks to the Author):

Referee Report for MS 2021-10-15899

In this manuscript, the authors have allegedly surveyed historical records for the lunar eclipses to identify dark total lunar eclipses as indications for stratospheric turbidity of extremely large volcanic eruptions in 1100 -- 1300 CE. This is an interesting attempt which would potentially benefit the scientific and historical communities.

Reply: We warmly thank the referee for their thorough and critical review of the manuscript. We have addressed the referee's comments one by one below and revised our paper based on their suggestions.

However, according to their supplementary materials, in most cases, they seem to have consulted not the original historical documents but the secondary catalogs for lunar eclipses unlike what is emphasised in their manuscript. This is especially the case with the non-European sources, which share mistakes of the previous catalogs, play little role in their discussions, and should be omitted from their manuscript.

Reply: We agree with the referee that clarifications are needed. A detailed answer to the referee's comment can be found below.

Their astronomical analyses owe much to the NASA website, whereas they have to philologically reconsider the observational sites, entirely renew their parameter with the state-of-the-art Earth's rotation variability. Overall, in my humble opinion, thorough revisions are needed before further consideration on this manuscript for publication in Nature.

Reply: We thank the referee for the comment. Dataset S1 now includes new estimates of lunar eclipse timing based on the latest Delta T (ΔT) values. We have also removed from Dataset S1 the column entitled "Observational sites". More details are provided below.

Major Comments

1. The eclipse magnitude

Here, they have accurately computed the eclipse timing and magnitude in the supplementary table, on the basis of the Five Millennia Catalog of Lunar Eclipses. However, I have to raise two concerns. Firstly, their reference is misleading. They have cited this catalog as if a published book (see Ref 35), whereas this is actually NASA's online catalog. This is not acceptable.

Reply: The Five Millennium Canon Of Lunar Eclipses: -1999 To +3000 (2000 BCE TO 3000 CE) is indeed a web-based catalog (<https://eclipse.gsfc.nasa.gov/LEcat5/LEcatalog.html>). But it is based on a technical publication (NASA/TP-2009-214172) composed of 680 pages that can be downloaded here: <https://eclipse.gsfc.nasa.gov/SEpubs/5MCLE.html>

Fred Espenak and Jean Meeus also published a supplemental catalog (NASA/TP-2009-214173) of 278 pages to the Canon that is available here:

<https://eclipse.gsfc.nasa.gov/SEpubs/5MKLE.html>

More seriously, since this catalog's release, the ΔT has been significantly updated (e.g., Stephenson et al., 2016). The authors must recalculate all these parameters based on the latest ΔT .

Reply: Accurate Delta T (ΔT) values are needed for the calculation of exact times of an eclipse and for determining the zone of visibility. To perform their predictions Espenak & Meeus (2009) employed the historical ΔT values estimated by Morrison & Stephenson (2004). Since their catalog's release, updated ΔT values have indeed been published. For the 12th and 13th century, there is a difference of about two minutes between the ΔT values computed by Morrison and Stephenson (2004) and the revised values recently released by Morrison et al. (2021) (see *Table 1*).

Year	Morrison & Stephenson (2004)		Morrison et al. (2021)	
	ΔT (seconds)	Error estimate (seconds)	ΔT (seconds)	Error estimate (seconds)
1100	1090	± 40	1220	± 15
1200	740	± 30	910	± 15

Table 1. Historical ΔT values computed for the years 1100 and 1200 by Morrison & Stephenson (2004) and Morrison et al. (2021)

The astrophysicist Dr. Yuk Tung Liu, working at the University of Illinois (USA), recently updated the work of Espenak & Meeus (2009) and performed retrocalculations and predictions of solar and lunar for the period -4000 - + 4000 (Liu, 2021).

Liu's calculations are based on the equations described in the *Explanatory Supplement to the Astronomical Almanac* (Urban & Seidelmann, 2013). Liu used the ΔT values published by Morrison et al. (2021) and the calculations were made with two lunar ephemeris, *DE431* and *DE441*.

Table 2 shows the differences between the Espenak & Meeus (2009) and Liu (2021) retrocalculation for the lunar eclipse of 05 May 1110 CE. There is a shift of about 2 minutes between the local times computed by Espenak and Meeus (2009) and those computed by Liu (2021).

Study	Calendar Date	Ecl. Type	Pen. Mag.	Umbral Mag.	Pen. Eclipse Begins	Alt	Partial Eclipse Begins	Alt	Total Eclipse Begins	Alt	Mid. Eclipse	Alt	Total Eclipse Ends	Alt	Partial Eclipse Ends	Alt	Pen. Eclipse Ends	Alt
Espenak and Meeus (2009)	1110-May-05	T	2.694	1.750	20:20	+06	21:13	+12	22:09	+16	22:58	+18	23:47	+19	00:44	+19	01:37	+16
Liu (2021)	1110-May-05	T	2.694	1.750	20:17	6	21:10	11	22:07	16	22:56	18	23:45	19	00:41	19	01:35	16

Table 2. Local circumstances table for the city of London, United Kingdom, for the lunar eclipse of 05 May 1110 CE.

All the local circumstances tables that can be found in the Dataset S1 have been updated using the Liu (2021) calculations. The revised estimates of lunar eclipse timing have no implication for the interpretation of the historical records.

2. Observational sites

This is closely relevant with the source reliability, which the authors have described the source reliability in the supplementary materials. This is a good move. However, it is not clear what they have learnt from this entry, at least in the main text. In fact, a recent case study has identified a source report of an atmospheric optics report in the Anglo-Saxon Chronicle with the celestial phenomenon in Northern France, after tracing chains of duplications (DOI: [10.5194/hgss-11-81-2020](https://doi.org/10.5194/hgss-11-81-2020)). The supplementary material does not allow us to see such philological traces. Rather, it seems the authors have generally assumed the observational site as chroniclers' residential area or somewhere nearby. The authors need to be more cautious on the identification of the observational sites, as the authors are keen to derive the eclipse parameters in their supplemental datasets.

Reply: We agree. When we began work on this project a few years ago, we hoped to determine observational sites. But this proved unrealistic, at least for most records. The original Dataset S1 was a 'working version,' and showed some early and unsuccessful attempts we made. This column "place of observation" was intended to be removed in the final version and is now omitted.

3. Analyses of the historical records (especially for non-European records)

The authors claimed that they have reexamined the original sources. However, this seems slightly dubious, especially for the non-European records. What they have consulted seems to be the catalogs of astronomical records. I suspect they have consulted Xu et al. (2000) [Ref. 42] for Chinese and Korean records, as their translations are virtually the same and frequently share the same mistakes. For example, some Japanese terminologies are transcribed into Chinese pronunciations. Similar issues are found throughout the East Asian and Middle Eastern eclipse records. The authors must distinguish the historical source documents from these secondary catalogs. At least, it would be excessive for the authors to claim that they have "extensively reexamined hundreds of annals and chronicles". They should explicitly clarify that they have mostly consulted not the original historical records but the catalogs of historical lunar-eclipse records.

4. East Asian records

While I appreciate the authors' reanalyses on the European eclipse records, I suspect their reanalyses on the non-European records are mostly not on the basis of the original historical sources but on the secondary catalogs (see comments below). Their translations are extremely similar to what has been published in Xu et al. (2000) and Yuasa (2007, 2010) [NB: probably not cited in the main text] for example (see my comments below).

Reply: We fully understand the concerns expressed by the referee. The version of the Dataset S1 that the referee received indeed contained numerous flaws. The original text for several entries was missing, especially for Japanese sources. A number of entries contained typos.

All European sources listed in the Dataset S1 have been reexamined by the first author. Whenever possible, we also tried to obtain access to original manuscripts to verify the accuracy of the critical editions.

The same holds true for Korean historical sources. The first author has investigated and listed all the lunar eclipses found in the astronomical treatises of the Goryeosa (volumes 47, 48, 49). But we understand the referee's confusion. The Dataset S1 that the referee received lacked the Excel sheet entitled "Dataset_S1_Korea". Therefore, the referee had absolutely no way to know whether we consulted the original critical editions or whether we extracted this information from Xu et al. (2000). We apologize for this. The revised version of the Dataset S1 now includes the missing table, the missing text and all the proper references needed to access easily the original text. The translations found in the table were made by Dr. Zhen Yang.

When we started to investigate Chinese records, we compiled the information collected by Xu et al. (2000). But soon we realized that some records were omitted. That's why, we decided to reexamine carefully the Song Shi (宋史, 960 to 1279 CE), the Jin Shi (金史, 1115-1234 CE), the Yuan Shi (元史, 1261 to 1367 CE), as well as the Wenxian tongkao (文献通考) compiled by Ma Dualin in 1317. If the referee compares closely our work with the catalog published by Xu et al. (2000), he may notice that a significant number of lunar eclipses that we included in the Excel sheet "Dataset_China_S1" are not found in Xu et al. (2000).

We acknowledge that we should have provided more clarity and accept that the way we referenced Chinese records was unsuitable. Consequently, "Dataset_S1_China" has been significantly revised. We now provide for each entry all the information needed for the reader to access the original text in the critical editions. The translations listed in the table were made by Dr. Zhen Yang.

Investigating Japanese historical sources was more challenging because reports of lunar eclipses are scattered in a large number of diverse sources, which are not always readily available. Therefore, we indeed used the compilation made by Xu et al. (2000), as stated in the previous version of the manuscript and in the Dataset_S1_Japan.

It would however be incorrect to say that we didn't investigate any of the original records. The authors have for instance checked several of the entries found in Xu et al. (2000) with the original texts,

especially for the *Chuyuki* (中右記) and the *Azuma Kagami* (吾妻鏡). The *Meigetsuki* (明月記) has also been reexamined using a new critical edition published in 2018. We also had a look at a facsimile of the original manuscript. The translation that we made of the 1229 total lunar eclipse is entirely new and independent from Xu et al. (2000).

But we agree with the referee that the first version of the Dataset_S1_Japan was far from being satisfactory. Several entries were missing the original text and most importantly only a small fraction of all the lunar eclipses reported in Japanese historical archives were compiled by Xu et al. (2000).

To increase the completeness of the Dataset S1 and to ensure that we didn't miss any source providing information about the color of lunar eclipses, we consulted *日本天文史料*, published by Shigeru Kanda in 1935 and reprinted in 1978. The book is 760 pages long. It lists all the astronomical phenomena (solar and lunar eclipses, comets, aurorae) found by Kanda in Japanese historical sources. The author also provides the original texts. All the entries are written in Kanbun. The *日本天文史料* can only be found in a limited number of libraries worldwide and we had to ask the National Diet Library of Japan for a copy, which arrived with us at the end of December 2021. The revised version of the "Dataset_S1_Japan" now includes all the lunar obscurations reports found by Kanda for the period 1100-1300 CE. And we provide for the first time a translation of each of these texts in English. These translations will allow the reader to access a work that has so far largely been restricted to scholars able to read and understand Kanbun. All the translations were made by Prof. Paul Atkins and Nobuko Horikawa.

The referee also expressed doubts about our claim to have reexamined hundreds of annals and chronicles. It is worth mentioning that the first author has been investigating European eclipse records for a decade. The author carefully and thoroughly examined the annals and chronicles written in the *Monumenta Germaniae Historica*, (ii) *Rerum Britannicarum Medii Aevi*, (iii) *Recueil des historiens des Gaules et de la France* as well as in the (iv) *Rerum Italicarum scriptores*. Each of these critical editions comprises more than 20 volumes. And each volume is about 500 to 1000 pages long and contains between 5 and 20 annals and chronicles. While we cannot claim to have identified all existing records of lunar eclipse available in Europe, we feel that stating that we reexamined over the last 10 years hundreds of sources is a conservative estimate.

They have also alleged that Korean records are written in Classic Chinese (P9L385), whereas this was actually Sino-Korean.

Reply: We thank the referee for spotting that mistake. The *Goryeosa* is indeed written in Hanja. The manuscript has been revised accordingly.

If so, why are they transcribing 高麗史 not as Gaolishi but as Koryo-sa? Koryo-sa could be better transcribed as Goryeo-sa, following discussions in Korean studies (e.g., Ref. 43).

Reply: We used the McCune–Reischauer romanization which is one of the most widely used systems for Korean. The new version of the manuscript now uses the Revised Romanization of Korean.

Honestly, apart from some exceptions, these non-European records have played little role in their manuscript.

Reply: The referee is right. East-Asian and Middle Eastern records only provide limited information about lunar eclipses color. However, since the dating of 1229 CE volcanic eruption relies heavily on the December 1229 CE dark lunar eclipse reported in the *Meigetsuki* (明月記), we felt compelled to check as many Japanese, Chinese and Korean records as we could, to ensure that we didn't miss any interesting report. Since we added to our dataset all the European eclipse records we could find, we consider adding East-Asian records important.

Therefore, I think the authors should remove their datasets from the non-European records, which are not quite original.

Reply: While we understand the suggestion, we suggest the revised Dataset S1 addresses the underlying criticism. Our translation of the East Asian records provides access to texts that have so far remained unknown to a wider audience. In this respect, we think that the East-Asian dataset of lunar eclipse remains valuable for the scientific community.

They include not only their records from East Asia but also those from Islamic Middle East (e.g., Stephenson and Said, 1997) and Russia (e.g., Vyssotsky, 1949). Instead, they should pinpoint the citation for the dark lunar eclipse in 1229.

Reply: We indeed checked the Islamic record of lunar eclipses translated in Stephenson and Said (1997). We didn't include the records in the dataset because the records are readily accessible and because none of the reports provides information about lunar eclipse color. However, we decided to include the records published by Vyssotsky (1949) because several of them provide information about lunar eclipse color.

5. Referencing in the Dataset S1 (Chinese records)

They have cataloged the East Asian records in Dataset S1. Especially for the Chinese records, they have referenced the sources only up to the chapter number without clarifying which page(s) in which critical edition(s) they have actually consulted. Instead, they have provided a link to Wikisource and C-text, as a convenient online source. This is not acceptable, as these texts frequently involve some weird transcriptions. They have to rely not on these unreliable online sources but on appropriate critical editions.

Reply: We fully agree with the referee. As stated above, the revised "Dataset_S1_China" now relies on the critical editions. We also now provide for each record the appropriate pages.

6. Romanizations of the Chinese terms in the Dataset S1

Their romanizations lack accents too. The Chinese language has four accents. Even for the same syllable, different accents will provide different meanings. If they insist in using Chinese sources, they have to put appropriate accents to their transcriptions.

Reply: We are not completely sure we understand the referee's comment. Is the referee referring to tones? To our knowledge tones are not always added to romanization. But we did our best to add the tones to our dataset, especially for the emperors, the reign periods and the days's name. For instance, our romanization now defines the emperor Huizong as follows: Emperor Huizong (Hūizōng 徽宗).

Would that be suitable?

7. Referencing in the Dataset S1 (Japanese records)

The original descriptions are mostly missing. They have to fill this entry or completely abandon the usage of the Japanese sources. For Gyokuyo and Azuma Kagami, they have specifically relied upon Yuasa's articles. Here, they must rely on an appropriate critical edition. Yuasa explicitly declared that he used Yoshikawa Kobunkan's critical edition in 1978-1979. They relied on the English translation mostly on Xu et al. (2000), whereas they sometimes lack translations for the entries with original records (e.g., 1175-May-08, 1175-Oct-31, and 1176-Apr-26).

Reply: We agree. Several original descriptions were missing. As stated above, they have now been added to the Dataset S1. As mentioned, most of the original descriptions added to Dataset S1 were extracted from the book by Kanda entitled 日本天文史料. This constitutes an undisputed benchmark in the field of historical astronomy in Japan. For the most interesting sources (i.e, those referring to dark lunar eclipses), the original critical edition of the source was consulted as indicated in the Dataset S1.

8. Source Description for the Japanese sources

The source descriptions were repeated many times. For example, they have repeated similar (or likely the same) notes for Azuma Kagami more than 25 times. For the readership, it is probably easier to have another supplementary table for the source documentation in a separate file. Some notes look interesting, whereas this repetition makes it extremely difficult to find which information is scientifically valuable.

Reply: The comment made by the referee is very relevant and we have given it careful thought. Yet we think that removing the source description and expecting the reader to refer to another entry will complicate the reading of Dataset S1, which is already dense in information. Since other referees also invited us to improve the readability of Dataset S1, we created an interactive atlas of lunar eclipse covering the period 1100-1300 CE, which provides information about each historical source and lunar eclipse report. The atlas can be accessed here:

<https://arcg.is/XyOKX>

9. "Dark eclipses" in the Japanese sources

For some of their notes, I have found philological difficulties. Their interpretation for the dark lunar eclipse in 1229 is convincing, with an explicit source description "The Moon's disk changed and

disappeared, but only as if covered by clouds." In contrast, the one in 1258 is too speculative. Here, they have interpreted "We could not see anything" as an unusually dark lunar eclipse. However, this is not justified as this description can be applied to another lunar eclipse which was not actually visible from Japan (Yuasa, 2007, 2010). The authors have omitted a record with the same description ("We could not see anything"), possibly because these eclipses were not visible from Japan. This will probably mislead the readership to associate this description with the 'dark eclipse'.

Reply: We fully agree with the referee. The description referring to the lunar eclipse of 1258 CE in the *Azuma Kagami* (吾妻鏡) remains unfortunately too short and cryptic to undoubtedly classify this event as a dark lunar eclipse. Furthermore, it is important to note that this account is not contemporary. It was written several decades after the event from a source which is now lost. These limitations had been noted in the original version of the dataset. We agree with the referee that this text should be used with caution. Given the uncertainties surrounding the description found in the *Azuma Kagami* (吾妻鏡), we now decided not to attribute any luminosity value to this eclipse in the Dataset S1.

However, it is important to state that a very dark lunar eclipse rated $L=0$ on the Danjon scale was observed in Genoa (Italy) at the same time. The description from Genoa was written by a contemporary, soon after the event occurred and is one of the most detailed descriptions of lunar eclipses available for the 12th and 13th centuries. This is an intriguing coincidence and we cannot exclude the possibility that the description found in the *Azuma* also refers to an unusually dark lunar eclipse.

10. The SAOD simulation

The authors have suggested the dark eclipses would be visible 3-20 months after the eruptions, using the SAOD simulation (Extended Data). This is interesting but difficult to see and understand. It would be nicer if the authors could graphically explain this procedure more understandably.

Reply: Following the referee's recommendation, the Extended Data Figure 1 was modified. The 3-20 months period, during which SAOD exceeds the threshold of 0.1, is now more clearly highlighted in the figure.

11. The European chroniclers' general attention to the celestial events

In P3L92-102, the authors have emphasized the European chroniclers' general attention to the celestial events, citing their remarkable coverage (~85%). However, this is not quite correct. This kind of trend is totally dependent on the nature and purpose of the chroniclers (Stephenson and Green, 2002, p. 140). It is well known that the SN1054 was missed from most of the European chroniclers (Stephenson and Green, 2002). Some chroniclers have paid significant attention to the celestial events, as exemplified with John of Worcester (e.g., DOI: [10.1080/03044181.2013.798742](https://doi.org/10.1080/03044181.2013.798742)). In contrast, some chroniclers have apparently paid limited attention to the celestial phenomena, as exemplified with Bede. Stephenson et al. (2020) have recently examined Bede's accounts and have proven omissions of many large solar and lunar eclipses visible at Jarrow (DOI: [10.1177/0021828619899188](https://doi.org/10.1177/0021828619899188)). Some celestial reports have been even derived from hearsay (DOI: [10.5194/hgss-11-81-2020](https://doi.org/10.5194/hgss-11-81-2020)). In this case, it

would be safer to avoid this generalisation, assuming the reason. In short, it is also questionable how far contemporary science (e.g., their Figure 1b) has been shared throughout the chroniclers.

Reply: We agree that our statement should be more nuanced. Some annalists and chroniclers showed remarkable interest in astronomy, such as Robert of Torigny, the continuator of Cosma of Prague or the chronicler of Bury St. Edmund who reported the solar and lunar eclipses for several years in a row. Yet it is true that others paid little attention to celestial phenomena as they felt that they were irrelevant to daily life. Therefore, as suggested by the referee, we decided to remove the sentence from our manuscript.

Yet, we should not consider that all monks lacked any knowledge in astronomy. For instance, Walcher (d. 1135 CE), a prior at the abbey of Malvern (Worcestershire, England) was one of the first scholars to adopt methods from Arabic science to compute lunaison tables and perform eclipse predictions (Nothaft, 2017). In the 13th century, several monasteries also sent their monks to the University to extend their intellectual horizon (Clark, 2020). We also found during our investigations that a chronicler from Basel in Switzerland was proud to announce that one of the brothers in his monastery successfully forecast a lunar eclipse in 1276 CE (see Dataset_S1_Continental_Europe):

Original text: *In vigilia sancti Clementis infra matutinas Praedicatorum Basiliae fuit ecclipsis lunae, quam frater juvenis eiusdem ordinis praedixit et fratribus ostendit (Gerard et Liblin, 1854, pp. 58)*

Translation: *“In the night of Saint Clement, before the Matins were said in the church, there was a lunar eclipse that one young brother of our order predicted and showed to his [others] brothers”. (Translation by Sébastien Guillet)*

GERARD C., LIBLIN J.J. (ed. and transl.), 1854. *Les annales et la chronique des Dominicains de Colmar*, Imprimerie et Lithographie de Mme Veuve Decker, 367p

Although it is possible the brother based his prediction on an available astronomical table, it nevertheless demonstrates the interest and knowledge that some monks had in astronomical phenomena.

Minor Comments

P2L67: solar coroneae => atmospheric optics.

Reply: The introduction has been modified accordingly.

P3L88: Astronomical and Five-Elements treatise => Astronomical treatises and Five-Elements treatises

Reply: The text has been modified accordingly. Thank you!

P3L88 Japan => Japanese

Reply: The manuscript has been changed accordingly.

P3L89: "diaries of courtiers or temple records" => I suspect Azuma Kagami is not diaries or temple records but chronicles or annals.

Reply: We agree. The manuscript has now been modified accordingly.

P3L90: "In continental Europe and in the Middle East, annals and chronicles from monasteries and towns represent the main sources." => Is this really true for the Islamic annals in the Middle East? The Arabic sources tend to be universal chronicles (See "Ta'rikh" in: Encyclopedia of Islam (ver 2), v. 10, pp. 257-302). If they still wish to claim this, they have to show appropriate references.

Reply: Thank you for this relevant comment and for correcting us. The text has been clarified accordingly.

P4L102: The medieval East Asians frequently recorded predicted eclipses (Stephenson, 1997; Yuasa, 2007). I wonder if the authors have considered this fact when they describe the proportion of the reported eclipses.

Reply: This is, once again, a very relevant comment. East Asian historians indeed frequently recorded predicted eclipses. While predicted eclipses are difficult to distinguish from observed events in Chinese records, we do have clues for Korean and Japanese eclipses that sometimes allow us to make such a distinction. We here show an example of a predicted eclipse recorded found in the Sankaiki 山槐記, the Diary of the courtier Nakayama Tadachika (1132-95) concerning the lunar eclipse of 26 April 1167 as follows:

Original text: 〔山槐記〕

三月十五日癸丑、天晴、有月蝕、宿曜師經範勘送日、虧初申初刻 {七十二分}、加時申六刻 {七十九分}、復末酉五刻 {六十二分}、大子 (分力) 十五分之十半弱、後聞、陰陽道同可現之由勘申云々、而算博士行衡不可現之由申、果不現云々、

Translation: Sankaiki

3rd month, 15th day... Clear skies, lunar eclipse. The astrologer Tsunenori sent his opinion, occlusion to begin at the Monkey (3p-5p), initial mark, 72nd minute, peak at the Monkey, 6th mark, 79th minute, conclusion at the Rooster (5p-7p), 5th mark, 62nd minute. Totality less than 10.5 parts of 15. Later I heard that the Yin and Yang specialists agreed that it would appear, but the doctor of calculations Yukihiro said it would not appear. In the end it did not appear, it was said. (Translation by Paul Atkins)

Kanda, S., 1935. Nihon Tenmon Shiryô (日本天文史料, Japanese Historical Records of Astronomy),

Koseisha Koseikaku Co., Ltd. ed. Tokyo.

All lunar eclipses for which we were able to confidently tell they were predictions rather than observations were discarded from our calculations.

References:

Clark, J. G. (2020). Monks and the Universities, c. 1200–1500. In A. I. Beach & I. Cochelin (Eds.), *The Cambridge History of Medieval Monasticism in the Latin West* (1st ed., pp. 1074–1092). Cambridge University Press. <https://doi.org/10.1017/9781107323742.058>

Liu, Y. T. (2021). Eight Millenia of Eclipses (4000 BCE – 4000 CE). Retrieved February 1, 2022, from <http://ytlui.epizy.com/eclipse>

Morrison, L. V., & Stephenson, F. R. (2004). Historical Values of the Earth's Clock Error Δ and the Calculation of Eclipses. *Journal for the History of Astronomy*, 35(3), 327–336. <https://doi.org/10.1177/002182860403500305>

Morrison, L. V., Stephenson, F. R., Hohenkerk, C. Y., & Zawilski, M. (2021). Addendum 2020 to 'Measurement of the Earth's rotation: 720 BC to AD 2015.' *Proceedings of the Royal Society A: Mathematical, Physical and Engineering Sciences*, 477(2246), 20200776. <https://doi.org/10.1098/rspa.2020.0776>

Nothaft, C. P. E. (Ed.). (2017). *Walcher of Malvern, De lunationibus and De Dracone*. Turnhout, Belgium: Brepols.

Stephenson, F. R. & Said S. Said. (1997). Records of Lunar Eclipses in Medieval Arabic Chronicles. *Bulletin of the School of Oriental and African Studies, University of London*, 60(1), 1–34.

Urban, S. E., & Seidelmann, P. K. (Eds.). (2013). *Explanatory supplement to the Astronomical almanac* (3rd ed). Mill Valley, Calif: University Science Books.

Vyssotsky, A. N. (1949). *Astronomical Records in the Russian Chronicles from 1000 to 1600 A. D. (as Collected by D. O. Sviatsky)* (Vol. Meddelanden fran Lunds Astronomiska Observatorium Series II, 126). Lund.

Xu, Z., Pankenier, D. W., & Jiang, Y. (2000). *East Asian archaeoastronomy: historical records of astronomical observations of China, Japan and Korea*. Amsterdam, Netherlands: Amherst, N.Y: Published on behalf of the Earth Space Institute by Gordon and Breach Science Publishers; Cambria Press.

Referee #3 (Remarks to the Author):

REVIEW: The “Dark Side of the Moon”: Constraining the history of medieval volcanism with astronomical records” by Guillet et al.

This paper presents a novel multi-proxy approach for constraining the timing of medieval volcanic events (including the use of astronomical records of lunar eclipses), with one goal being to shed light on the transition from the Medieval Warm Period to the Little Ice Age. This work provides an additional source of information for comparison to climate models, ice cores, tree rings and other proxies of volcanism, and thus can improve our understanding of such events and their implications regarding the climatic forcings of the past millennium. The paper is thus of considerable interest and potentially broad appeal.

Reply: We are very grateful to the referee for their incisive review. We did our best to address all concerns raised by the referee and the manuscript is, we believe, significantly improved.

With regards to the application of dendrochronology as a source of independent confirmation of the dating/ calibration/evaluation of the results, my main suggestion would be that the authors should mention several important caveats regarding their selection and usage of the hemispheric tree-ring reconstructions of Schneider et al. (2015, based solely on Maximum Latewood Density or MXD; 2017) and of Wilson et al. (2016, NTREND), the latter based on 11 ring width series (RW), 18 density (MXD as well as Blue Light), as well as 25 mixed series using both RW and density. It should be noted that RW and MXD parameters can vary significantly in their cooling response to major volcanic events, and display different spectral features and potential biases, which could influence their results. Typically, RW features substantial autocorrelation or biological memory (causing a more muted, delayed response to forcing) and MXD features much less, if at all (e.g. Esper et al. 2015). For Samalas, for example, Wilson et al. (2016) note that the MXD response is more immediate, while RW shows a stronger response in year $t+1$ and their mixed record response is in Yr $t+2$. RW and MXD have different seasonal sensitivities; e.g. RW can even integrate conditions in the non-growing season, although this is controversial. RW also tends to have a weaker, less robust volcanic signal and hence is not ideal for evaluating volcanic response.

Reply: This is a very relevant comment. Following the referee’s suggestion, we decided to add to the Supplementary Information a new section (Text S2) focusing on the limitations and uncertainties of our approach. Several paragraphs specifically address the challenges associated with using tree ring proxies to identify and evaluate the timing and environmental consequences of volcanic eruptions. Following the referee’s recommendations, we:

- 1) State that tree-ring width (TRW) based temperature reconstructions can underestimate, delay and even artificially extend the cooling induced by large volcanic eruptions and that such records must be used with caution.
- 2) Discuss the limitations associated with the use of MXD (maximum latewood density) chronologies, due to the limited availability of records.

- 3) Explain the reasons that led us to favor the Schneider 2015, NTREND 2016, and NVOLC_v2 2020 large-scale temperature reconstructions and to discard for instance the recent TRW-based temperature reconstructions published by Büntgen et al. (2021).

There are also differences at lower frequencies which may be likely relevant to the MWP-LIA discussions. NTREND, for example, features a longer and warmer medieval period from 900-1170. The Schneider MXD series shows relatively modest warmth from 850-1050 and more muted multidecadal variability, and a delayed LIA onset using MXD, as compared to RW records. Tree-ring series are also typically less reliable back in time, e.g. during the critical MWP interval, due to decreased data coverage, and thus there can be less coherency both between the tree-ring series themselves and also between the tree-ring series and climate models during this important time period, emphasizing the need to collect additional data for early in the past millennium. The authors should thus address whether or not these differences have any significance to their analysis and results.

Reply: We agree with the referee that the limited number of available tree-ring chronologies and series extending back to the medieval period can impact the robustness of the reconstruction and the detection of volcanic cooling. We tried to minimize these biases by:

- 1) Selecting large-scale reconstructions based on high-quality tree-ring records with high values of Rbar and EPS over the medieval period.
- 2) Selecting large-scale reconstructions using a significant number of MXD records.

Although the Schneider 2015 (Schneider et al., 2015), NTREND 2016 (Wilson et al., 2016), and NVOLC_v2 2020 (Guillet et al., 2021) large-scale reconstructions are not a perfect record of past volcanic eruptions, they still remain the state of the art. Furthermore, despite the use of different tree-ring networks, different meteorological targets for calibration and different methodologies, the Schneider 2015, NTREND 2016, and NVOLC_v2 2020 reconstructions show synchronous cooling after HMP eruptions, which increases our confidence in the reliability of our estimates.

But there is clearly still room for improvement. We added to the Supplementary Information (Text S2) a paragraph in which we explore avenues of research that may help to refine even further our estimates. We discuss the importance of (i) collecting data in underrepresented regions of the Northern and Southern Hemisphere, (ii) developing new millennium-long and well-replicated MXD records. We also discuss the importance of investigating new tree-ring parameters such as wood anatomical traits, which may provide even higher resolution records than MXDs.

It is also worthwhile mentioning that there are substantial differences in spatial response to volcanism depending on the prevailing atmospheric circulation, latitude and timing of the eruption, spatial biases in tree-ring data coverage (eg. towards coverage in Europe during the MWP), among other factors, that are obscured in the hemispheric scale average.

Reply: Again, we fully agree with the referee. A note has been added to the Supplementary Information concerning the limitations/uncertainties of our approach.

The authors should also address how their results might be impacted by multiple events/eruptions – for example the Gomagatake, Japan eruption in summer 1640 may have complicated the timing calculated for Parker in the Philippines (Jan 1640).

Reply: We agree with this point. Closely-spaced volcanic eruptions can indeed add complexity. Multiple events can increase stratospheric aerosol loading, the period when the SAOD exceeds 0.1 and thus the interval during which a dark lunar may be observed.

Following the referee's suggestion, we here test our approach on the cluster of eruptions that occurred circa 1640-1641 CE and discuss how it might affect the dating of the Parker eruption:

Two distinct sulfur peaks are found in Greenland ice core records circa 1641-42 CE. In Antarctica, only one unambiguous sulfate spike is recorded, peaking in 1642 CE (Figure 1). The resulting bipolar sulfur peak in 1641-1642 CE has been ascribed to the eruption of Mount Parker (6°N, Philippines) on December 26, 1640 CE, but sulfate emitted from Koma-ga-take (42°N, Japan) volcano on July 31, 1641 CE, has potentially also contributed to the sulphate concentrations observed in Greenland at this time (Stoffel et al., 2021).

Figure 1. (a) Monthly-resolved non-sea-salt sulfur records from the NEEM-2011-S1 and the WDC06A ice cores and (b) brightness of the total lunar eclipses observed in Europe for the period 1100–1122 CE.

The astronomer Johannes Helvetius (1611-1687 CE) reports, in his *Selenographia* published in 1647 CE the occurrence of a dark lunar eclipse on April 25, 1642 CE and explains that the Moon could barely be distinguished even with the aid of the telescope, although the air was sufficiently pure to discern the stars (step 1.1):

Ejusmodi notabile exemplum & mihi animadvertere contigit, Anno 25 Aprilis: Luna enim, tempore totalis obscurationis penitus evanescebat, ita ut Spectatorum haud pauci, nec locum Lunae in coelo invenire, vel indigitare potuerint; & quamvis Telescopio instructi essemus, nihilominus visum Luna illudebat, cum tamen stellae quarti & quinti honoris, satis essent aspectabiles.

Selenographia. Johannis Hevelii Selenographia sive lunae descriptio, 1647. Rar 8932, ETH-BIB, Zürich, chapter 6, pp. 117

This account contrasts with usual records of total lunar eclipses depicting blood-red eclipsed moons and suggests that significant amounts of volcanic aerosol remained in the stratosphere on April 1642, i.e., 15 months after the eruption of Mount Parker. The eclipse can be rated L=0 on the Danjon scale (step 1.2).

Based on SAOD simulations from Evol2k and the IPSL climate model, we evaluated that the highest probability of SAOD to exceed 0.1 is between 3 to 20 months (step 2.2). Reciprocally, we assume that

the most probable eruption time window falls within 3 to 20 months prior to the dark lunar eclipse, i.e. between November 1640 and January 1642 CE (step 2.2).

This window can be further refined using the Schneider 2015, NTREND 2016 and NVOLC_v2 records (step 3.1), which all show major cooling during summer 1641 (-0.6°C/Schneider2015, -0.7°C / NVOLC_v2, -0.8°C / NTREND, wrt. 30-years climatology).

Here we assume that the peak cooling observed after a volcanic event generally occurred between 9 and 15 months after an eruption (step 3.2). We therefore attributed the strong cooling observed in tree-ring reconstructions during the 1641 CE summer to an eruption that probably occurred between May and October 1640 CE (step 3.2).

Finally, we combined probabilities of occurrence estimated from steps 2.3 and 3.2 (step 4.1) and estimated that the eruption most probably occurred between November 1640 CE and January 1641 CE (step 4.2). Our estimate agrees with the known date of the Parker eruption (December 1640 CE).

In that particular setting the occurrence of Koma-ga-take eruption in July 1641 CE didn't prevent our approach from estimating robustly the most likely date of the Parker eruption. But of course, we can't exclude the possibility that in other cases closely-spaced volcanic events will increase uncertainties in our dating. These uncertainties will depend on interval between eruptions, the timing of the dark eclipse and the cooling observed in the tree-ring reconstructions.

A number of the most relevant eruptions in their study from the more recent period were mainly from lower latitudes. Higher latitude eruptions – Laki, Katmai – might show less of a delay in timing of response in temperature reconstructions based on middle to higher-latitude northern tree-ring sites than what is shown in their study.

Reply: Since the large majority of HMP eruptions, analyzed in this manuscript are low-latitude eruptions, we considered that peak cooling observed after a volcanic event generally occurred between 9 and 15 months after an eruption. This time window was defined based on the peak cooling observed after the largest tropical eruptions of the last 500 years for which precise dates are known.

But we concur with the referee that the delay between eruption and peak cooling observed in tree-ring reconstructions might be shortened for high-latitudes eruptions. Given the flexibility of our approach, this time window can be easily adjusted for future studies aiming to date high-latitude events.

References

Büntgen, U., Allen, K., Anchukaitis, K. J., Arseneault, D., Boucher, É., Bräuning, A., et al. (2021). The influence of decision-making in tree ring-based climate reconstructions. *Nature Communications*, 12(1), 3411. <https://doi.org/10.1038/s41467-021-23627-6>

Guillet, S., Corona, C., Ludlow, F., Oppenheimer, C., & Stoffel, M. (2020). Climatic and societal impacts of a “forgotten” cluster of volcanic eruptions in 1108-1110 CE. *Scientific Reports*, 10(1). <https://doi.org/10.1038/s41598-020-63339-3>

Schneider, L., Smerdon, J. E., Büntgen, U., Wilson, R. J. S., Myglan, V. S., Kirilyanov, A. V., & Esper, J. (2015). Revising midlatitude summer temperatures back to A.D. 600 based on a wood density network. *Geophysical Research Letters*, *42*(11), 2015GL063956. <https://doi.org/10.1002/2015GL063956>

Stoffel, M., Corona, C., Ludlow, F., Sigl, M., Huhtamaa, H., Garnier, E., et al. (2021). *Climatic, weather and socio-economic conditions corresponding with the mid-17th century eruption cluster* (preprint). Continental Surface Processes/Historical Records/Decadal-Seasonal. <https://doi.org/10.5194/cp-2021-148>

Wilson, R., Anchukaitis, K., Briffa, K. R., Büntgen, U., Cook, E., D'Arrigo, R., et al. (2016). Last millennium northern hemisphere summer temperatures from tree rings: Part I: The long term context. *Quaternary Science Reviews*, *134*, 1–18. <https://doi.org/10.1016/j.quascirev.2015.12.005>

Referee #4 (Remarks to the Author):

The manuscript provides an estimate of strong volcano eruptions based on reported lunar eclipses. As a part of the procedure, simulated stratospheric aerosol optical depths from pre-existing global climate model simulations were used for estimating the correlation of volcano emission and the rating of the appearance of lunar eclipses. The use of global model simulations for this purpose is a valid approach although it has large uncertainty since the lifetime and transport of stratospheric aerosol is very much model dependent. It would be good to add some discussion on this in the description of the use of model data.

Reply: We are grateful for these constructive reflections and comments on our manuscript. Indeed, large uncertainties remain regarding the formation, transport and residence time of volcanic aerosols in the stratosphere for events larger than the 1991 Pinatubo eruption.

The suggestions convinced us to add to the Supplementary Information (see text S2) a section in which we now detail the current uncertainties relative to the formation and evolution of aerosols after large volcanic eruptions in climate models and how these uncertainties can affect our findings.

Reviewer Reports on the First Revision:

Referee #1 (Remarks to the Author):

I have read the revised text of this paper with great interest and am happy that the points I raised have been addressed.

Dataset S1, covering Europe, has fewer typos and is helpful in providing images of the entries for lunar eclipses as well as transcriptions of the text.

I noticed that the final 'h' in Peterborough has disappeared from some entries in column AA; and that the word Historiarum in 'Flores Historiarum' is several times spelt wrongly. These are very minor points which can easily be corrected during copy editing.

Some of the information given in the details for each entry still seems to me repetitive or redundant, but I can see that the aim is for completeness and accept it on those grounds.

The only detail which definitely needs correcting is in line 332 where the Apocalypse of Silos is incorrectly called the Silos of Apocalypse (which does not make sense).

Referee #2 (Remarks to the Author):

Summary

I confirm this manuscript has substantially modified the main manuscript and the ΔT issues upon the last revision. Overall, I think this is an interesting manuscript if it does not needlessly reduce its reliability with the unreliable supplementary material and discussions on the East Asian records which play little role in their own manuscript. I have some serious concerns on their eclipse parameters as well. In this regard, I think a major revision is still needed for further consideration, at least in the philological and astronomical viewpoints.

Unfortunately, the authors have kept some serious flaws in their supplementary datasets. The authors have confirmed they have not identified the exact observational sites for most of the cases. This means they cannot derive such seemingly accurate values for the eclipse parameter. This is something serious, as only total lunar eclipses are usable for their studies. They have also included too many non-total lunar eclipses in the supplementary datasets and make this dataset rather inaccessible to the readers. They need to show what they have actually used in their manuscript. This manuscript shows little about the source value for the individual manuscript variants (in their philological genealogy from the autograph manuscript). They have involved serious philological flaws in the East Asian record files, whereas only one of them has played some scientific roles in their manuscript (Japanese report in 1229) and other records will just needlessly reduce their philological reliability. This is actually a pity, as their manuscript itself is interesting almost purely based on European manuscripts. Overall, IMHO, they should remove their unnecessary description for the East Asian records and detail more about the European records which form their manuscript backbone.

1. Title

The authors have removed "Dark Side of the Moon" from the title. However, here, the authors have only exploited the lunar eclipse records. This modification will mislead the readers to consider the authors have comprehensively analysed various astronomical records in a wide range. The authors should clarify their analyses on the lunar eclipses in the title. Moreover, given the manuscript contents, the authors should modify the title as "with [occidental] astronomical records" or "with [occidental] lunar-eclipse records"

2. Contaminations of penumbral and partial lunar eclipses

I fully agree with the authors' own caveat (i) in the Supplementary Text 2: "(i) Only total lunar eclipses are suitable. Partial and penumbral eclipses cannot be used for reliable estimates of stratospheric turbidity". This is extremely an important point and forms a baseline for my most serious concern for this manuscript.

Despite the authors' caveat (i), the Supplementary Datasets have involved quite a number of penumbral and partial eclipses. They are making these tables rather inaccessible for the readership. They should be removed. The authors may wish the readers to use the Excel Tabs to sort only total lunar eclipses. However, not all the readership has a good access to MS Excel. Some even tries to print them out. Moreover, the file is already excessively large (~100 MB). The authors should present what are essentially important for their article.

3. The observational sites

The authors have omitted the "place of observation", as it proves unrealistic to trace their sources back to the original. This sounds a good move. However, the authors have still kept columns for the Lat. and Long., apparently to indicate the observational sites. They must be removed, as the authors removed the exact observational sites from their analyses.

Moreover, for the eclipse visibility condition, the 'Local Circumstances' is far more important than the 'General Circumstances' in Dr. Liu's website. This decides everything, timings and altitude at each timing and each magnitude in the authors' datasets. In short, if the authors cannot show the observational sites, most of their dataset parameters do not make sense. I will address more details later.

At least they cannot emphasize too much about successes in their retrospective calculations (P3L98-100), as their observational sites are unknown, do not allow exact calculation, and cannot give them a solid basis for quantitative discussions. They may state that they are mostly consistent, whereas it is far-fetched to raise such exact number.

4. Eclipse atlas

This decision force the authors to remove their ' Historical Lunar Eclipse Atlas in 1100-1300 CE' (P11L461), as the authors themselves have explicitly confirmed they gave up identifications of the

observational sites.

5. Eclipse parameters

I am skeptical about the validity of the authors' dataset without exact observational time. The visibility condition differs up to where the observers are situated. Even on the night of the total lunar eclipse, some places can see only a partial or penumbral eclipse. Even if the observers manage to see the total lunar eclipse, the timing, altitude, and exact magnitude may easily differ from each other up to their locations. IMHO, without identifying observational sites, it is extremely difficult to derive exact numerical values for most of the eclipse parameters (as shown in their Dataset).

The penumbral and umbral magnitudes are probably derived from "General Circumstances" of Dr. Liu's website (by the ways, thank you for accommodating my comments for the ΔT issues). Of course, without exact observational sites, we can never compute the "Local Circumstances" in Dr. Liu's website. The contact time for each timing has some strange offsets with Dr. Liu's website. For example, for the total lunar eclipses of 1103-Sep-17 and 1172-Jan-12 (but on 1172-Jan-13 in Dr. Liu's dataset), I have compared the authors' table with Dr. Liu's website.

Pen. Eclipse Begins 20:02 -- 19:02:11
Partial Eclipse Begins 20:57 -- 19:56:39
Total Eclipse Begins 21:53 -- 20:53:16
Mid. Eclipse 22:43 -- 21:42:44
Total Eclipse Ends 23:32 -- 22:32:13
Partial Eclipse Ends 0:29 -- 23:28:49
Pen. Eclipse Ends 1:23 -- 1103-Sep-18 00:23:21

Pen. Eclipse Begins 23:40 -- 1172-Jan-12 22:39:48
Partial Eclipse Begins 0:46 -- 1172-Jan-12 23:46:19
Total Eclipse Begins 1:49 -- 00:49:13
Mid. Eclipse 2:40 -- 01:39:41
Total Eclipse Ends 3:30 -- 02:30:11
Partial Eclipse Ends 4:33 -- 03:33:06
Pen. Eclipse Ends 5:39 -- 04:39:29

The authors seemingly introduced 1 hour offset in their table here. If these timestamps are given in UT, these errors cast serious doubts on the reliability of the authors' dataset. If these timestamps are given in the local time, it is impossible to derive the local time without identifying the exact observational sites. The authors may wish to cite time zones. However, the time zones were introduced only after the mid 19th century. It is erroneous to describe the medieval events with non-existing time zones.

The authors have to consider a better presentation method for their dates especially when the timestamps go beyond the midnight. Otherwise, this table will easily mislead the readership.

This uncertainty will immediately and inevitably affect altitude and time at each point, and the

maximum magnitude. Such flaws have significantly reduced the reliability of the authors' datasets. If the authors still wish to maintain their dataset files, they should remove columns C-U. These flaws are commonly shared in Dataset S1.

6. Continental European chronicles

They form a backbone for this manuscript. I appreciate their significant effort here. However, without observational site, how can the authors derive the exact lunar eclipse parameter here? It would make sense if the authors do not provide seemingly exact number for each eclipse. There should have been total lunar eclipses in most of these cases, whereas it is unrealistic to provide these exact number.

7. The selection of the manuscript images

Additionally, they have shown some manuscript images. How have they chosen these manuscripts? Especially if there are some variants, it is always important to trace back to their autograph or the quasi-original manuscript. Can the authors explain why they have chosen these manuscript images (supplementary materials) among several variants? Even for the Anglo-Saxon Chronicles, there are serious analyses for the manuscript genealogies (e.g., Swanton, 1998, *The Anglo-Saxon Chronicles*, New York, Routledge). The authors probably need to explain why they have chosen specific variants for their manuscript images for each cases.

8. Copyright/ownership of the Dataset S1: Eclipse records from Continental Europe

This is a backbone for this manuscript with the greatest value. Here, the authors have reproduced manuscript images from multiple historical archives such as the British Library, Lund University Library, Benediktinerkollegium, Melk of Benediktinerstift, Österreichische Nationalbibliothek, Bibliothèque Virtuelle du Mont Saint-Michel, Bibliothèque Municipale de Dijon, Admont Benediktinerstift, Bodleian Libraries. Each historical archive has their own rule for the record reproduction/reuse authorizations. The authors need to provide permission letters for these manuscript images before acceptance, as they need to follow the copyrights and ownership of these historical archives and avoid their potential violations. Furthermore, the authors are obliged to mention these historical archives in the acknowledgment.

9. Report of eclipse predictions

As confirmed in their manuscript, the medieval people had an ability to predict eclipses (P3L120). In this case, how have the authors secured the actual observational records from the predicted reports?

10. East Asian records

The authors have also agreed that the East Asian records have played little roles in their manuscript. This manuscript set their discussions up on their quasi-comprehensive surveys for the continental

European records and compare East Asian records for some specific cases. In fact, only one Japanese record in 1229 has made a significant contribution in the main manuscript (the only mentioned case). For the Japanese records, they have mostly consulted a secondary catalog (Kanda, 1935).

In this regard, I do not find any points to associate their East Asian dataset with their main manuscript. At least, the authors have not conducted comprehensive archival survey for Japanese records. Chinese and Korean records played little (or no) role at least in the main text. For the East Asian records, their rather their minor philological flaws have needlessly reduced the manuscript reliability which is actually using the occidental historical records.

I think the authors should remove the East Asian records from their Dataset S1 and their statements in P111436-453. If the authors wish to publish such translations, they should write an independent article. These East Asian records will gain more appropriate attention in the specialist journals for astronomy or astrophysics. It is not that appropriate to publish something irrelevant with the main discussion. After all, these datasets are not "supporting" their own manuscript.

11. Korean records

Their source description states they have consulted Goryeosa. No more details are given in the Dataset_S1_Korea, although the authors have claimed that they have consulted the original critical editions. The authors have not indicated neither the exact volume number nor the page number for their Korean source records. While they pasted individual URLs from <http://db.history.go.kr>, their texts are based on certain critical editions. The authors must provide individual sources with their exact volume number and page number.

The Readme_Fataset_S1_Korea confuses McCune-Reichauer romanization with the Revised Romanization of Korea: e.g., Goryeo dynasty vs Koryo dynasty, Kojong vs Gojong. They need to make their statement internally consistent.

The authors have agreed that these Korean records are written in Hanja. However, they transcribed some terms into Mandarin (especially for the dates). For example, bingshen (丙申) should be rather read as byeongsin or pyŏngsin. The authors need to revise these details throughout the dataset file.

There are some issues with their translations too. For example, for 1127-Nov-21, the authors have translated 救食 as "the special ritual" with some notes. This is understandable, whereas this is not quite accurate. I would rather suggest the authors to transcribe them in romanized Korean and add some explanatory notes.

For 1130-Oct-18, "密雲不見" is translated as "but it could not be observed because of the clouds". Here, the authors have overlooked "密". The authors have translated "見" as "observed" here but as "seen" in other places. What difference do they wish to emphasize here? This is rather translated as "but it could not be [seen] because of [dense] clouds".

For 1185-Apr-16, the authors have translated "木瓜" as "papaya". However, papayas were brought to Eurasia only after the geographical discovery. In the 12th century, I would not expect the Korean

to recognize papayas in the modern sense.

There are many such cases. I am afraid but their translations may needlessly reduce the reliability of the overall discussions which is not actually relying on the Japanese records that much. In this sense, it is probably more appropriate to remove this section.

12. Chinese records

For Songshi and Jinshi, the volume number is missing. For example, for the lunar eclipse of 1111-Oct-18, this would be volume 52 and page 1097. The authors need to revise these details throughout the dataset file.

Wenxian Tongkao also miss the volume number as well. It has multiple volumes in the original version. (The authors have translated them seemingly as ch [chapter?]). Their volume number should be clarified.

I think some translations are not quite accurate. For example, 1135-Jan-02, the authors have translated 食于井 and 月食于井 in exactly the same ways (the moon was eclipsed in JING). In the first sentence, the character for "moon" is missing. Therefore, the authors are clearly supplementing this character for Wenxian Tongkao in analogy with their parallel record in Songshi. This kind of interpretation needs some notes (not to be misled with "manipulation"), although I personally agree with the authors' interpretation.

Some predictive records are contaminated. For example, 1138-Oct-20, the authors have translated this as "the moon was eclipsed but it was hidden by the clouds". However, they probably missed "当" for Songshi report. This could be translated as "the moon [should have been] eclipsed but it was hidden by the clouds." Then, this is not observational but predictive. In contrast, for the Wenxian Tongkao report, they have missed "既". This is rather translated as: "the moon was totally eclipsed but it was hidden by the clouds."

For 1144-Jan-22, the authors have translated "当食， 雲蔽之不见" as "the moon was eclipsed but it was hidden by the clouds." Similar flaws are found here. This is rather like: "the moon [should have been] eclipsed but it was hidden by the [shady] clouds [and was not seen/visible]." I understand "雲" stands "overcast", "cloudy", or "shady" and similar with 雲. However, they need to be as loyal to the original text as possible.

For 1160-Feb-23, the authors have transcribed 陰雲 (traditional style) as 阴云 (simplified style). The critical editions are mostly loyal to the original text. The simplified Chinese characters has been used only recently. The authors should accurately transcribe the original characters following the critical editions, if they wish to show these texts in the Dataset file.

There are many such cases. For major issues, they have to remove all the predictive records so that they will not be misunderstood as observational records. For minor issues, there are a number of translation flaws. These issues may sound minor, whereas the authors must show accurate

translations if they really wish to show them in the Dataset file.

I am afraid but their translations may needlessly reduce the reliability of the overall discussions which is not actually relying on the Japanese records that much. In this sense, it is probably more appropriate to remove this section.

13. Japanese records

I agree that the authors have consulted the original critical editions for some specific interesting event. However, this is not true for the majority of their dataset descriptions. In the revision, the authors have explicitly admitted that they have relied on Kanda (1935). This ends up confusing citation method.

For example, Supplementary Text 2 cites Azuma Kagami as: "Kanda, S., 1935. Nihon Tenmon Shiryô (日本天文史料, Japanese Historical Records of Astronomy), Koseisha Koseikaku Co., Ltd. ed. Tokyo. (See Dataset S1 for more information)" without indicating page number. Here, the authors should have cited Azuma Kagami instead of Kanda (1935).

Alternatively, in the Dataset_S1_Japan, the authors cited the eclipse of 1181-Jun-28 as: "〔玉葉〕三十六 Gyokuyô, vol. 36. 五月十五日庚寅、陰晴不定、此日月蝕也、5th month, 15th day... Alternately cloudy and clear, lunar eclipse on this day. ○「オッポルツェル」食表ニヨレバコノ日月食ナシ、(Kanda note: according to Oppolzer's list there was no lunar eclipse on this day.)"

If the authors intend citation from the original records, they should consult Gyokuyo's critical editions instead of Kanda's catalog without page number. It is slightly misleading to cite Oppolzer's statement. It is not historical records but Kanda's note, after all.

I am not convinced with their English translations for more than a few cases. For example, 踏歌節會 should be translated as "nodal festival of stamping songs" rather than "Seasonal Festival of Dance and Song" (e.g., DOI: 10.1163/9789004264540_006). The authors have translated "今夜月蝕" as lunar eclipse tonight. Well, literally this might be fine. However, this is an ungrammatical sentence in English. The authors should modify their translation.

For the eclipse report on 1212-Nov-10, the authors have translated the 天晴 as "clarification sky". This is more appropriately translated as "the sky was clear", as this was not "晴天". 云々 was omitted in their translation.

"大陰虧" is translated as "total lunar eclipse". Not exactly true. This is better translated as "great lunar eclipse". "建曆二年十月九日" not as "Kenryaku 2.10.19" but as "Kenryaku 2.10.9".

There are many such cases. I am afraid but their translations may needlessly reduce the reliability of the overall discussions which is not actually relying on the Japanese records that much. In this sense, it is probably more appropriate to remove this section.

Yuasa articles have not been cited either in the main manuscript or the supplementary information, whereas these articles are what the authors relied on for Azuma Kagami for their eclipse identification. This is not the best practice. They should cite Yuasa articles in the main manuscript.

14. Islamic records

If they wish to use these records in their Dataset S1, they should show the volume number and page number of the original Arabic chronicles. Moreover, they should show the original Arabic text, as done for the occidental records and East Asian records. The same can be said to the Russian records. For the Syriac records, the original texts are missing too.

15. East Asian records themselves

These datasets have not played significant scientific roles in their main manuscript and involve a number of philological flaws. IMHO, this dataset will needlessly reduce their philological reliability. Moreover, if they wish to publish a 'the most complete compilations of lunar eclipse observations for the 12th and 13th centuries', they should do so in an astronomical journal with an appropriate title (not as a paper for volcanic studies).

Their supplementary text 2 explicitly confirms the lunar eclipse record in 1258 is not usable for their discussion: "Given these circumstances, we did not attribute any luminosity value to this reported eclipse." Moreover, the authors cite this account from Nihon Tenmon Shiryo, a compilation in 1935, as if it were a contemporary historical account. The authors have not addressed similar accounts "it was not properly visible" in Azuma Kagami, although Yuasa articles have explicitly listed these cases. In these regards, I think it is inappropriate to cite Azuma Kagami account for the eclipse record for their source 'data' table.

Again, I do not think the East Asian have played any scientific roles in their article, except for the Japanese lunar eclipse record in 1229. Therefore, they should drop their description about the East Asian records in the supplementary materials and just specifically cite a case report from 1229.

They should also replace the Azuma Kagami citation in Supplementary Text 2 to something from the occidental chronicles. Azuma Kagami is based on a different cultural background, in contrast with the occidental chronicles. They should visualize their "careful treatment and interpretation" for the historical records on the basis of the occidental records which they have actually used in this manuscript.

Minor Comments

P2L51: food crises => great famines?

P2L67: Bishop's rings => e.g., Bishop's rings

P3L87-91: Rather than describing East Asian records which are not much used here, the authors should explain how they have collected that amount of the European chronicles in this study in

which selection criteria.

P3L110: Middle Eastern => Eastern Christianity?

P11L444: Not 'original records' but 'transcriptions'

Author Rebuttals to First Revision:

Referees' comments:

Referee #1 (Remarks to the Author):

I have read the revised text of this paper with great interest and am happy that the points I raised have been addressed.

Reply: We thank the referee for re-evaluating our manuscript and for the relevant comments he/she made.

Dataset S1, covering Europe, has fewer typos and is helpful in providing images of the entries for lunar eclipses as well as transcriptions of the text.

I noticed that the final 'h' in Peterborough has disappeared from some entries in column AA; and that the word Historiarum in 'Flores Historiarum' is several times spelt wrongly. These are very minor points which can easily be corrected during copy editing.

Reply: Noted and corrected.

Some of the information given in the details for each entry still seems to me repetitive or redundant, but I can see that the aim is for completeness and accept it on those grounds.

The only detail which definitely needs correcting is in line 332 where the Apocalypse of Silos is incorrectly called the Silos of Apocalypse (which does not make sense).

Reply: Noted and corrected.

Referee #2 (Remarks to the Author):

Summary

I confirm this manuscript has substantially modified the main manuscript and the ΔT issues upon the last revision. Overall, I think this is an interesting manuscript if it does not needlessly reduce its reliability with the unreliable supplementary material and discussions on the East Asian records which play little role in their own manuscript. I have some serious concerns on their eclipse parameters as well. In this regard, I think a major revision is still needed for further consideration, at least in the philological and astronomical viewpoints. Unfortunately, the authors have kept some serious flaws in their supplementary datasets. The authors have confirmed they have not identified the exact observational sites for most of the cases. This means they cannot derive such seemingly accurate values for the eclipse parameter. This is something serious, as only total lunar eclipses are usable for their studies. They have also included too many non-total lunar eclipses in the supplementary datasets and make this dataset rather inaccessible to the readers. They need to show what they have actually used in their manuscript. This manuscript shows little about the source value for the individual manuscript variants (in their philological genealogy from the autograph manuscript). They have involved serious philological flaws in the East Asian record files, whereas only one of them has played some scientific roles in their manuscript (Japanese report in 1229) and other records will just needlessly reduce their philological reliability. This is actually a pity, as their manuscript itself is interesting almost purely based on European manuscripts. Overall, IMHO, they should remove their unnecessary description for the East Asian records and detail more about the European records which form their manuscript backbone.

Reply: We thank the referee again for the time and care put into this thorough review, which forces us to be as rigorous as possible. We did our best to take into consideration and accommodate the critique and comments. We summarize the revisions we made as follows:

- 1) The title has been modified following the referee's comments (see section 1).
- 2) All records pertaining to partial and penumbral have been removed from the manuscript (see sections 2 and 15).
- 3) We feel, however, that keeping all the total lunar eclipses from East-Asia is important. Our motivations are further explained in sections 10 and 15.
- 4) All eclipse parameters have been removed from the Dataset S1 (see section 5).
- 5) All the manuscript images have been removed (see section 8).
- 6) Observational Sites. In Europe, the *lat* and *long* column refer to the places where the sources were written. The coordinates provided in the Excel sheet "Dataset_S1_Continental_Europe" should not be seen as "Observational sites". More information can be found in section 3.
- 7) We provide more information about the online critical edition of the Goryeosa we used (see section 11).

- 8) Revised translations have been made for Chinese, Korean and Japanese sources (see sections 11, 12 and 13).

1. Title

The authors have removed “Dark Side of the Moon” from the title. However, here, the authors have only exploited the lunar eclipse records. This modification will mislead the readers to consider the authors have comprehensively analysed various astronomical records in a wide range. The authors should clarify their analyses on the lunar eclipses in the title. Moreover, given the manuscript contents, the authors should modify the title as "with [occidental] astronomical records" or "with [occidental] lunar-eclipse records"

Reply: We agree and have modified the title as follows:

Lunar eclipses illuminate timing and climate impact of medieval volcanism

Since the text pertaining to the dark lunar eclipse of 1229 was written in Kyoto by the Japanese courtier Fujiwara no Teika, we feel it is better not to add the word “occidental” to the title.

2. Contaminations of penumbral and partial lunar eclipses

I fully agree with the authors' own caveat (i) in the Supplementary Text 2: "(i) Only total lunar eclipses are suitable. Partial and penumbral eclipses cannot be used for reliable estimates of stratospheric turbidity". This is extremely an important point and forms a baseline for my most serious concern for this manuscript.

Despite the authors' caveat (i), the Supplementary Datasets have involved quite a number of penumbral and partial eclipses. They are making these tables rather inaccessible for the readership. They should be removed. The authors may wish the readers to use the Excel Tabs to sort only total lunar eclipses. However, not all the readership has a good access to MS Excel. Some even tries to print them out. Moreover, the file is already excessively large (~100 MB). The authors should present what are essentially important for their article.

Reply: We accept the comment and have removed from Dataset_S1 all texts pertaining to penumbral and partial lunar eclipses.

3. The observational sites

The authors have omitted the "place of observation", as it proves unrealistic to trace their sources back to the original. This sounds a good move. However, the authors have still kept columns for the Lat. and Long., apparently to indicate the observational sites. They must be removed, as the authors removed the exact observational sites from their analyses.

Reply: To address this comment, we must make a distinction between Continental Europe and East-Asian records.

Continental Europe

We agree with the referee that it is not possible to know the exact place of observation of all the lunar eclipses reports presented in the Dataset S1. Thus, we emphasise that the coordinates provided in the table Dataset_S1_Continental_Europe should not be regarded as “Observational sites”. This matter was already mentioned on the welcome page of the Historical Lunar Eclipse Atlas (<https://arcg.is/Xy0KX>).

The columns *Lat* and *Long* in “Dataset_S1_Continental_Europe” refer to locations (i.e., monasteries, the towns) where historical sources were written. Several monasteries mentioned in the dataset can be difficult to locate. Some are no longer extant, others are ruins. For this reason, we still consider that providing coordinates is useful.

There are also several cases where the lunar eclipses *were* probably observed at the location where the sources were written. For instance, there is evidence that the series of eleven lunar eclipses recorded in the *Chronicle of Bury St. Edmund* between 1258 and 1297 were observed at Bury Saint Edmund’s abbey (Suffolk, East England). The same holds for several lunar eclipses recorded by the chronicler Robert of Torigni in the 12th century that were very likely observed at Mont Saint Michel abbey.

To avoid any confusion, we have added a few sentences to the Excel sheet entitled: “Readme”.

East-Asia

The columns *Lat* and *Long* in “Dataset_S1_China”, “Dataset_S1_Korea” refer to the capitals of the Chinese, Korean dynasties.

In China and Korea, lunar eclipses were recorded by official astronomers and the place of observations can usually be taken as the dynastic capital of the time (Stephenson, 1997).

In Japan, the imperial court mostly remained in Kyoto for more than a thousand years and most of the courtier diaries that we consulted appear to have been written in Kyoto. It is likely that most astronomical observations were performed there.

Moreover, for the eclipse visibility condition, the 'Local Circumstances' is far more important than the 'General Circumstances' in Dr. Liu's website. This decides everything, timings and altitude at each timing and each magnitude in the authors' datasets. In short, if the authors cannot show the observational sites, most of their dataset parameters do not make sense. I will address more details later.

At least they cannot emphasize too much about successes in their retrospective calculations (P3L98-100), as their observational sites are unknown, do not allow exact calculation, and cannot give them a

solid basis for quantitative

discussions. They may state that they are mostly consistent, whereas it is far-fetched to raise such exact number.

Reply: Compared to solar eclipses, lunar eclipses have a wide footprint on the ground. An eclipse visible in Paris (for instance) is likely to span most of Europe (see Figure 1).

While we cannot know the exact place of observation of each lunar eclipse report, source criticism allows us to estimate the area where the eclipse was most likely observed. We made significant efforts to trace the origins of each lunar eclipse report that we found and to assess whether the reports were contemporary and original or later duplicates (See Dataset_S1_Continental_Reports).

In most cases, lunar eclipse observations were generally made near the monasteries that reported them.

There are of course exceptions. For example, we were able to identify that a partial lunar eclipse reported in England, under the year 1218, in the *Flores Historiarum*, by Roger of Wendover drew on an observation made by Oliver of Paderborn in Damietta in Egypt. More information on this interesting

Figure 1. Visibility maps of the 1110, 1258 and 1276 total lunar eclipses. (Espenak and Meeus, 2009)

example is provided in the version of Dataset S1 we resubmitted in March 2022.

Therefore, overall, we feel it is possible to estimate the proportion of observed and missed lunar eclipses in Europe.

The same holds true for China, Korea and Japan. We know that most observations were carried out in the dynastic capitals (Stephenson, 1997), which allows us to estimate the completeness of the records.

4. Eclipse atlas

This decision force the authors to remove their ' Historical Lunar Eclipse Atlas in 1100-1300 CE'

(P11L461), as the authors themselves have explicitly confirmed they gave up identifications of the observational sites.

Reply: The Historical Lunar Eclipse Atlas was developed in light of suggestions to make the Dataset S1 more user-friendly. We wanted to offer readers and referees the opportunity to visualize and access easily the sources we investigated.

As specified in the section 3, the coordinates do not show observational sites. But if the referee prefers us to leave the atlas aside we won't publish it. The revised version of the manuscript we resubmitted doesn't include the Historical Lunar Eclipse Atlas.

5. Eclipse parameters

I am skeptical about the validity of the authors' dataset without exact observational time. The visibility condition differs up to where the observers are situated. Even on the night of the total lunar eclipse, some places can see only a partial or penumbral eclipse. Even if the observers manage to see the total lunar eclipse, the timing, altitude, and exact magnitude may easily differ from each other up to their locations. IMHO, without identifying observational sites, it is extremely difficult to derive exact numerical values for most of the eclipse parameters (as shown in their Dataset).

The penumbral and umbral magnitudes are probably derived from "General Circumstances" of Dr. Liu's website (by the ways, thank you for accommodating my comments for the ΔT issues). Of course, without exact observational sites, we can never compute the "Local Circumstances" in Dr. Liu's website. The contact time for each timing has some strange offsets with Dr. Liu's website. For example, for the total lunar eclipses of 1103-Sep-17 and 1172-Jan-12 (but on 1172-Jan-13 in Dr. Liu's dataset), I have compared the authors' table with Dr. Liu's website.

Pen. Eclipse Begins 20:02 -- 19:02:11
Partial Eclipse Begins 20:57 -- 19:56:39
Total Eclipse Begins 21:53 -- 20:53:16
Mid. Eclipse 22:43 -- 21:42:44
Total Eclipse Ends 23:32 -- 22:32:13
Partial Eclipse Ends 0:29 -- 23:28:49
Pen. Eclipse Ends 1:23 -- 1103-Sep-18 00:23:21
Pen. Eclipse Begins 23:40 -- 1172-Jan-12 22:39:48
Partial Eclipse Begins 0:46 -- 1172-Jan-12 23:46:19
Total Eclipse Begins 1:49 -- 00:49:13
Mid. Eclipse 2:40 -- 01:39:41
Total Eclipse Ends 3:30 -- 02:30:11
Partial Eclipse Ends 4:33 -- 03:33:06
Pen. Eclipse Ends 5:39 -- 04:39:29

The authors seemingly introduced 1 hour offset in their table here. If these timestamps are given in UT, these errors cast serious doubts on the reliability of the authors' dataset. If these timestamps are given in the local time, it is impossible to derive the local time without identifying the exact observational sites. The authors may wish to cite time zones. However, the time zones were introduced

only after the mid 19th century. It is erroneous to describe the medieval events with non-existing time zones.

The authors have to consider a better presentation method for their dates especially when the timestamps go beyond the midnight. Otherwise, this table will easily mislead the readership.

This uncertainty will immediately and inevitably affect altitude and time at each point, and the maximum magnitude. Such flaws have significantly reduced the reliability of the authors' datasets. If the authors still wish to maintain their dataset files, they should remove columns C-U. These flaws are commonly shared in Dataset S1.

Reply: We understand and agree with the referee's concerns. We have therefore removed all eclipse parameters from Dataset_S1. Still we would like to provide the referee with a number of explanations so that he or she can better understand the reasons that led us to include the eclipse parameters.

As specified in the Excel sheet entitled "Readme", the eclipse parameters shown in the version Dataset_S1 submitted in March 2022 were based on the local circumstances computed by Liu (2021), which explains the offset noted by the referee.

The local circumstances tables provided at the beginning of "Dataset_S1_Continental_Europe", "Dataset_S1_China", "Dataset_S1_Korea", and "Dataset_S1_Japan" presented the lunar eclipse contact time for the cities of Paris, Bian (Kaifeng), Songdo (Kaesong) and Kyoto.

Once again, our purpose was to provide the reader with all the information on lunar eclipses considered in this study. But of course, this invites problems, especially for Europe. While the contact times differ little between Paris and a monastery in Germany, the uncertainties regarding contact time, altitude and magnitude increase as the distance from Paris increases.

We sincerely appreciate the referee's feedback on this matter.

6. Continental European chronicles

They form a backbone for this manuscript. I appreciate their significant effort here. However, without observational site, how can the authors derive the exact lunar eclipse parameter here? It would make sense if the authors do not provide seemingly exact number for each eclipse. There should have been total lunar eclipses in most of these cases, whereas it is unrealistic to provide these exact number.

Reply: As specified in the section 5, all the eclipse parameters have now been removed.

7. The selection of the manuscript images

Additionally, they have shown some manuscript images. How have they chosen these manuscripts? Especially if there are some variants, it is always important to trace back to their autograph or the

quasi-original manuscript. Can the authors explain why they have chosen these manuscript images (supplementary materials) among several variants? Even for the Anglo-Saxon Chronicles, there are serious analyses for the manuscript genealogies (e.g., Swanton, 1998, *The Anglo-Saxon Chronicles*, New York, Routledge). The authors probably need to explain why they have chosen specific variants for their manuscript images for each cases.

Reply: We agree with the referee that the manuscript images provided are not essential to the manuscript. We have therefore decided to remove all the manuscript images from Dataset_S1.

Again, we offer some background. Showing the manuscript images had two main purposes:

- 1) To demonstrate that we didn't blindly used the critical editions and that we did our best to consult, whenever possible, original manuscripts. Furthermore, consulting the original manuscript is often critical for source criticism. In a number of cases, it helped us to assess to whether the lunar report investigated was written by contemporary hand and whether it is original or derived from a source that is now lost.
- 2) It is evident from the detailed and thorough review we received that the referee has been working for many years on astronomical records. But one should not forget that most of readers of *Nature* and most of scientists working on the field of volcanism and climate have limited experience with medieval manuscripts. Therefore, we thought that it could be useful share illustrations of the manuscripts.

Almost all the critical editions that we consulted provide details about the original manuscripts they used to build the editions. We used such information to locate the most relevant manuscripts. Whenever possible, we consulted the autograph version (provided that it is still extent) or at least the oldest known extent version of the manuscript.

8. Copyright/ownership of the Dataset S1: Eclipse records from Continental Europe

This is a backbone for this manuscript with the greatest value. Here, the authors have reproduced manuscript images from multiple historical archives such as the British Library, Lund University Library, Benediktinerkollegium, Melk of Benediktinerstift, Österreichische Nationalbibliothek, Bibliothèque Virtuelle du Mont Saint-Michel, Bibliothèque Municipale de Dijon, Admont Benediktinerstift, Bodleian Libraries. Each historical archive has their own rule for the record reproduction/reuse authorizations. The authors need to provide permission letters for these manuscript images before acceptance, as they need to follow the copyrights and ownership of these historical archives and avoid their potential violations. Furthermore, the authors are obliged to mention these historical archives in the acknowledgment.

Reply: Each library has indeed its own policy regarding the use and reproduction of their manuscripts. For instance, the *Codex Gigas* which is currently hosted at the National Library of Sweden, (Stockholm, Sweden) can be reproduced freely and without specific permissions. The same holds true for the Gottfried Wilhelm Leibniz Bibliothek in Hanover (Germany), which very recently decided that for

publication is no longer required for scans of manuscript in the public domain. Permissions are however required for reproducing manuscripts from the Bodleian and British Libraries.

As specified in the section 7, we decided to remove all the manuscripts images from Dataset_S1. Therefore, the permission letters for these images are no longer needed. We also secured all the necessary permissions for the manuscript images shown in Figure 1. The supporting documents have been submitted to *Nature*, along with the revised manuscript.

9. Report of eclipse predictions

As confirmed in their manuscript, the medieval people had an ability to predict eclipses (P3L120). In this case, how have the authors secured the actual observational records from the predicted reports?

Reply: Annals and chronicles were the main genre of historical writing in Europe in the 12th and 13th centuries. One of their main objectives was to record major events (human deeds or natural prodigies). In this respect, annals and chronicles are extremely unlikely to have been contaminated by any predicted report. Most of the records that we investigated reported lunar eclipses using the past tense (“luna **passa est** eclisim”, “eclipsis lunae **facta est**”, “eclipsis lunae et tota sanguinae **fuit**”), which suggests that these events actually occurred and were witnessed.

In the last revision we submitted in March 2022, we quoted a chronicler from Basel proudly announcing that one brother from his monastery successfully predicted a lunar eclipse. But even in this case, the chronicler clearly specifies that the eclipse was observed by the community:

Original text: *In vigilia sancti Clementis infra matutinas Praedicatorum Basilae fuit ecclipsis lunae, quam frater juvenis eiusdem ordinis praedixit et fratribus ostendit (Gerard et Liblin, 1854, pp. 58)*

Translation: *“In the night of Saint Clement, before the Matins were said in the church, there was a lunar eclipse that one young brother of our order predicted and showed to his [others] brothers”. (Translation by Sébastien Guillet)*

GERARD C., LIBLIN J.J. (ed. and transl.), 1854. *Les annales et la chronique des Dominicains de Colmar*, Imprimerie et Lithographie de Mme Veuve Decker, 367p

We can also add that the numerous descriptions we found reporting the color of the eclipsed Moon leave no doubt on the fact that these reports are observations and not predictions.

10. East Asian records

The authors have also agreed that the East Asian records have played little roles in their manuscript.

This manuscript set their discussions up on their quasi-comprehensive surveys for the continental European records and compare East Asian records for some specific cases. In fact, only one Japanese

record in 1229 has made a significant contribution in the main manuscript (the only mentioned case). For the Japanese records, they have mostly consulted a secondary catalog (Kanda, 1935).

In this regard, I do not find any points to associate their East Asian dataset with their main manuscript. At least, the authors have not conducted comprehensive archival survey for Japanese records.

Reply: We have indeed used the work of Kanda published in 1935. But it is important to stress that Kanda's compilation remains to date the key reference. We are unaware of any more recent archival survey. The most eminent scholars working in the field of historical astronomy have relied on the work of Kanda, such as John Steele (Steele, 2000) and Richard Stephenson:

Japanese observations of solar eclipses are also numerous, but unlike those from Korea they are scattered - along with references to other celestial phenomena - in a large number of diverse writings. These works include privately compiled histories, diaries and temple records as well as official histories such as the Dainihonshi (History of Great Japan). Fortunately, early in the present century Kanda (1934) made an exhaustive collection of astronomical observations from this varied material; this extends down to AD 1600. Kanda's extensive publication contains a major section devoted to solar eclipses. I have used this as my principal source of Japanese data (see also Kanda, 1935).

Historical Eclipses and the Earth's Rotation, Stephenson R., 1997. Cambridge University Press, pp. 266.

Therefore, we feel that the work of Kanda remains valuable despite its age. We add that no attempt had hitherto been made to translate the work of Kanda into English. We made that effort and feel that our translation is suitable for inclusion in the supplementary materials of this manuscript.

Chinese and Korean records played little (or no) role at least in the main text.

Reply: Indeed, our work suggests that Chinese and Korean records contain almost no information on lunar eclipse coloration. But we need to show the readers of the manuscript that no information about lunar eclipses color could be found in these records. See next comment for further detail.

For the East Asian records, their rather their minor philological flaws have needlessly reduced the manuscript reliability which is actually using the occidental historical records. I think the authors should remove the East Asian records from their Dataset S1 and their statements in P11L436-453. If the authors wish to publish such translations, they should write an independent article. These East Asian records will gain more appropriate attention in the specialist journals for astronomy or astrophysics. It is not that appropriate to publish something irrelevant with the main discussion. After all, these datasets are not "supporting" their own manuscript.

Reply: We agree that the partial and penumbral records are not essential to the main discussion. These records have therefore been removed. Nevertheless, we maintain that including all the total lunar eclipse records that we investigated in the supplementary materials is important.

If we remove East-Asian records from our manuscript, we can anticipate criticism for overlooking hundreds of existing records from China, Korea and Japan.

Another alternative would be to state in the manuscript that all the East Asian records we consulted contained no information about eclipse color, except for the Meigetsuki, without providing any text in the supplementary materials. But this would require the reader to “trust” us. East-Asian records are difficult to access for non-specialists. Most readers cannot read classical Chinese, Hanja or Kanbun. Therefore, if we don’t provide these texts and their translations, they will have limited means to evaluate our conclusions. We note that in an opinion article in Nature in 2018, statistician Philip B. Stark states: “Science should be ‘show me’, not ‘trust me’; it should be ‘help me if you can’, not ‘catch me if you can’.” (Stark, 2018, <https://doi.org/10.1038/d41586-018-05256-0>). By showing all the lunar eclipse records we considered, even the one that don’t provide any information about eclipse luminosity, we feel that we actually help the readers and make our research reproducible.

It is also important to show readers that we didn’t simply “cherry-pick” the records that were interesting to us. By showing all the lunar eclipse records we considered, we aim for transparency. Since we listed in the Dataset_S1 the lunar eclipse records from Europe, it seems logical to do the same for the East-Asian records consulted.

11. Korean records

Their source description states they have consulted Goryeosa. No more details are given in the Dataset_S1_Korea, although the authors have claimed that they have consulted the original critical editions. The authors have not indicated neither the exact volume number nor the page number for their Korean source records. While they pasted individual URLs from <http://db.history.go.kr>, their texts are based on certain critical editions. The authors must provide individual sources with their exact volume number and page number.

Reply: The online edition (<http://db.history.go.kr>) presented by the National Institute of Korean History (NIKH) should be seen as a new critical edition. It is based on several woodblock prints of the Goryeosa owned, among others, by the Dong-A University (Busan) and the Gyujanggak Royal Library (Seoul) and the Yonsei University (Seoul).

For instance, the woodblock print owned by the Dong-A University is one of the oldest extant and most complete versions of the Goryeosa. It was published in 1613 and is thought to be a copy of an edition published during the reign of Seongjong of Joseon (1457 – 1495) (Bruneton, 2020, <https://doi.org/10.4000/books.pressesinalco.26386>). More information about the materials used to create the online edition can be found at: http://db.history.go.kr/introduction/intro_kr.html (in Korean).

The online edition provides the original text and annotations. It provides the exact volume number where the text of interest can be found. The dates are given using both lunar and Julian calendars, which is helpful for the users unfamiliar with inter-conversion. And each text is linked to one or several scanned images of the woodblock prints on which the online edition is based (see Figure 2). In this respect, we feel that the individual URLs we added to the Excel file provide all the information needed by users.

The same website also allows the reader to access a Korean translation of the Goryeosa, known as *Guyeok Goryeosa* (History of the Goryeo Dynasty translated into Modern Korean, Gyeongin munhwasa Publisher, 2008).

They are about 80 editions of Goryeosa worldwide (Gwangchul, 2014). We chose the online edition over other editions because:

- 1) It is accessible to a wide audience. We should indeed be aware that few readers will have access to the printed version of the Goryeosa.
- 2) It is – to our knowledge – one the latest attempts to create a critical edition of the Goryeosa. The National Institute of Korean History (NIKH) started the project in 2009 and completed it in 2015 (Park, 2019).
- 3) Even the most recent printed critical editions contain flaws (Breuker et al., 2012; Park, 2019). Asea Munhwasa published in 1972 an edition of the Goryeosa which became a standard. But the edition was only based one woodblock print preserved in the Gyujanggak Royal Library and didn't consider other existing versions of the text. The printing quality of some characters is also quite poor in the 1972 edition (see Breuker et al., 2012 for more information)

The total lunar eclipse of 06.05.1118 in the Goryeosa

http://db.history.go.kr/id/kr_047_0010_0030_0110_1560

Volume 47 → 卷四十七 > 조종책 -> 天文 -> 月五曜凌犯及星變 > 彗 > 隕석이 일어나다

Lunar calendar → 1118년 5월 14일(癸) 병신(丙申), 1118년 6월 5일(庚) **隕석이 일어나다**

Julian calendar → 五月丙申 月食. **Original text written in Hanja**

Links to scanned images → 고종실소장본, 국사편찬위원회 소장본(KOJ1636)

Figure 2. Online edition of the Goryeosa developed by the National Institute of Korean History (NIKH)

The `Readme_Fataset_S1_Korea` confuses McCune-Reichauer romanization with the Revised Romanization of Korea: e.g., Goryeo dynasty vs Koryo dynasty, Kojong vs Gojong. They need to make their statement internally consistent.

Reply: The `Readme_Dataset_S1_Korea` indeed contained several words still transcribed following the McCune-Reichauer romanization. We are grateful for this comment and have made all the necessary changes.

The authors have agreed that these Korean records are written in Hanja. However, they transcribed some terms into Mandarin (especially for the dates). For example, bingshen (丙申) should be rather read as byeongsin or pyöngsin. The authors need to revise these details throughout the dataset file.

Reply: We agree and accept this an excellent point. All the dates are now transcribed in Romanized Korean.

There are some issues with their translations too. For example, for 1127-Nov-21, the authors have translated 救食 as "the special ritual" with some notes. This is understandable, whereas this is not quite accurate. I would rather suggest the authors to transcribe them in romanized Korean and add some explanatory notes.

Reply: There are in the *Goryeosa*, as well as in *Veritable Records of the Joseon Dynasty* several references to lunar or solar eclipse ceremonies in which the king, dressed in white clothes and accompanied by his closest ministers, attempted to rescue the Moon or the Sun. There is no single or simple way to translate the character “救食”. The character “救” means “to save/to rescue” or “to forbid/to prevent”, while the character “食” here likely means in that context “eclipse”. “救食” has sometimes been translated as “eclipse prayer” or “ceremony of praying during an eclipse”:

Original text:

○壬子朔/日有食之既, 上素服救食。

The Veritable Records of King Taejo, Year 6 (1397), Month 5, Day 1, Entry 1

Translation:

An eclipse occurred, and the king donned a white robe and offered the eclipse prayer.

The Annals of King T'aejo. Founder of Korea's Chosön Dynasty, Byonghyon C. (transl.), 2014, Harvard University Press, Cambridge, Massachusetts pp. 673.

Original text:

○乙亥朔/日食。上素服御勤政殿楹外, 救食如儀, 陰雲不見。

Translation:

A solar eclipse occurred. The King, dressed in white mourning garments, went to the outside of the front pillars of Geunjeong Hall 勤政殿. He performed the ceremony of praying during an eclipse 救食, according to protocol. The solar eclipse was not observed due to dark clouds.

The Veritable Records of King Sejong, Year 11 (1429), Month 8, Day 1, Entry 1

Korean historians have sometimes transcribed “救食” as “구식 (gusig)” in modern Korean. We have chosen to revise our translation as follows:

The 2nd year of King Injong (仁宗) of Goryeo, 12th month, day muo (戊午), the moon was totally eclipsed. The King put on his white clothes and went to the palace for the eclipse ceremony.

This translation is now accompanied by a note providing information about this ceremony (see Dataset_S1_China for more information).

For 1130-Oct-18, “密雲不見” is translated as “but it could not be observed because of the clouds”. Here, the authors have overlooked “密”. The authors have translated “見” as “observed” here but as “seen” in other places. What difference do they wish to emphasize here? This is rather translated as “but it could not be [seen] because of [dense] clouds”.

Reply: We thank the referee for this valuable comment. All the necessary changes have been made to the Dataset S1.

For 1185-Apr-16, the authors have translated “木瓜” as “papaya”. However, papayas were brought to Eurasia only after the geographical discovery. In the 12th century, I would not expect the Korean to recognize papayas in the modern sense. There are many such cases. I am afraid but their translations may needlessly reduce the reliability of the overall discussions which is not actually relying on the Japanese records that much. In this sense, it is probably more appropriate to remove this section.

Reply: This is once again a very relevant comment. We were also quite unsatisfied with the translation.

Several authors have used the term “papaya” to translate the character “木瓜”. This the case, for instance, of the sinologist James Legge (1814-1897) – who became notorious for translating a substantial number of Chinese Classics to English (Huilin, 2011, <https://doi.org/10.3917/rhc.337.0085>). Legge translates, in his book *The Chinese Classics*, the poem *Muh Kwa*, written sometimes between 800 and 600 BC, as follows:

投我以木瓜、報之以瓊瑤。
匪報也、永以為好也。

*There was presented to me a papaya,
And I returned for it a beautiful Ju-gem;
Not as a return for it,
But that our friendship might be lasting.*

X – Muh Kwa, Odes of Wei, Legge J., (ed. and transl.), 1871. *The Chinese Classics*, volume IV, part 1, pp. 107.

But we agree with the referee that the use of papaya may not be accurate. In the aforementioned poem and in the Goryeosa, the character “木瓜” (mùguā) probably refers to the fruit of the tree *Chaenomeles sinensis* (Chinese Quince). This tree is native from China. In Korea, the tree is known as

mogwa-namu (모과나무) and the fruit is called mogwa (모과). We have now chosen to use “quince” and added an explanatory note at the end of our translation.

12. Chinese records

For Songshi and Jinshi, the volume number is missing. For example, for the lunar eclipse of 1111-Oct-18, this would be volume 52 and page 1097. The authors need to revise these details throughout the dataset file.

Reply: The text pertaining to the lunar eclipse of 1111-Oct-18 can indeed be found in the chapter 52, volume 4 and page 1097 of the version of the Song shi edited by Zhonghua Book Company. This information was presented in the revised Dataset S1 submitted in March 2022. We refer the referee to cells X22 and AB22.

Wenxian Tongkao also miss the volume number as well. It has multiple volumes in the original version. (The authors have translated them seemingly as ch [chapter?]). Their volume number should be clarified.

Reply: We are grateful this has been spotted. Wenxian Tongkao indeed missed the volume number. This information has been corrected throughout the Excel sheet entitled “Dataset_S1_China”.

“ch.” indeed stands for chapter. In order to avoid confusion, we have now replaced “ch.” by chapters.

The comments made by the referee prompted us to check carefully how we referenced the critical editions. Dataset_S1 now allows (1) to read the original text, (2) to know which editions we used and (3) to find easily the text of interest thanks to the volume, chapter and pages numbers we provided in the “Dataset_S1_China”.

I think some translations are not quite accurate. For example, 1135-Jan-02, the authors have translated 食于井 and 月食于井 in exactly the same ways (the moon was eclipsed in JING). In the first sentence, the character for "moon" is missing. Therefore, the authors are clearly supplementing this character for Wenxian Tongkao in analogy with their parallel record in Songshi. This kind of interpretation needs some notes (not to be misled with "manipulation"), although I personally agree with the authors' interpretation.

Reply: Chapter 285 of the Wenxian Tongkao is dedicated to the Moon and to lunar eclipses. The authors of Wenxian Tongkao intentionally omitted the character 月 meaning “moon” to avoid repetition. When we translated the sentence, we had no choice but to add the word “moon” to ensure our translation was meaningful. Square brackets [] have been added around the words that we added to fill a gap or an omission in the text.

Some predictive records are contaminated. For example, 1138-Oct-20, the authors have translated this as "the moon was eclipsed but it was hidden by the clouds". However, they probably missed "当"

for Songshi report. This could be translated as "the moon [should have been] eclipsed but it was hidden by the clouds." Then, this is not observational but predictive.

Reply: Indeed. The sentence has been corrected accordingly.

In contrast, for the Wenxian Tongkao report, they have missed "既". This is rather translated as: "the moon was totally eclipsed but it was hidden by the clouds."

Reply: We have investigated several editions of the Wenxian Tongkao but couldn't find in any of them the missing character "既" mentioned by the referee.

For 1144-Jan-22, the authors have translated "当食， 𩇛雲蔽之不见" as "the moon was eclipsed but it was hidden by the clouds." Similar flaws are found here. This is rather like: "the moon [should have been] eclipsed but it was hidden by the [shady] clouds [and was not seen/visible]." I understand "𩇛" stands "overcast", "cloudy", or "shady" and similar with 雲. However, they need to be as loyal to the original text as possible.

Reply: We agree. The sentence has been corrected.

For 1160-Feb-23, the authors have transcribed 陰雲 (traditional style) as 阴云 (simplified style). The critical editions are mostly loyal to the original text. The simplified Chinese characters has been used only recently. The authors should accurately transcribe the original characters following the critical editions, if they wish to show these texts in the Dataset file.

Reply: Critical editions also use in some cases simplified style. Following the referee's comment, we checked carefully all the texts found Dataset_S1_China to make sure our transcription is faithful to the critical editions of Zhonghua Book Company.

There are many such cases. For major issues, they have to remove all the predictive records so that they will not be misunderstood as observational records. For minor issues, there are a number of translation flaws. These issues may sound minor, whereas the authors must show accurate translations if they really wish to show them in the Dataset file.

Reply: We have added for all the records that could be predictive a note to avoid any confusion.

13. Japanese records

I agree that the authors have consulted the original critical editions for some specific interesting event. However, this is not true for the majority of their dataset descriptions. In the revision, the authors have explicitly admitted that they have relied on Kanda (1935). This ends up confusing citation method.

For example, Supplementary Text 2 cites Azuma Kagami as: "Kanda, S., 1935. Nihon Tenmon Shiryô (日本天文史料, Japanese Historical Records of Astronomy), Koseisha Koseikaku Co., Ltd. ed. Tokyo.

(See Dataset S1 for more information)" without indicating page number. Here, the authors should have cited Azuma Kagami instead of Kanda (1935).

Reply: The citation has been replaced with the original reference.

Alternatively, in the Dataset_S1_Japan, the authors cited the eclipse of 1181-Jun-28 as: "〔玉葉〕三十六 Gyokuyō, vol. 36. 五月十五日庚寅、陰晴不定、此日月蝕也、5th month, 15th day... Alternately cloudy and clear, lunar eclipse on this day. ○「オッポルツェル」食表ニヨレバコノ日月食ナシ、(Kanda note: according to Oppolzer's list there was no lunar eclipse on this day.)"

If the authors intend citation from the original records, they should consult Gyokuyo's critical editions instead of Kanda's catalog without page number.

It is slightly misleading to cite Oppolzer's statement. It is not historical records but Kanda's note, after all.

Reply: Indeed. When we translated the transcription made by Kanda, we also translated his notes. This note was not supposed to be added to the Excel file. We thank the referee for spotting the mistake.

I am not convinced with their English translations for more than a few cases. For example, 踏歌節會 should be translated as "nodal festival of stamping songs" rather than "Seasonal Festival of Dance and Song" (e.g., DOI: 10.1163/9789004264540_006).

Reply: Regarding the term *tōka no sechie* 踏歌節會, there is no standard translation.

The translation proposed by Reviewer 2 is from Duthie (2014), *Man'yōshū and the Imperial Imagination*, p. 100, n. 71.

However, we also find:

"Dance and Royal Song Banquet" in Piggott and Yoshida (2008), *Teishin kōki*;

"Ambulating Song-dance Feast" in Butler (2002), *Emperor and Aristocracy in Japan, 1467-1680*; and

"Ladies' performance of dance and song" in Miner, Morrell, and Odagiri (1985), in *Princeton Companion to Classical Japanese Literature*, p. 710.

In this context, "Seasonal Festival of Dance and Song" sounds convincing to us and more accessible to general readers than the proposed alternative.

The authors have translated "今夜月蝕" as lunar eclipse tonight. Well, literally this might be fine. However, this is an ungrammatical sentence in English. The authors should modify their translation.

Reply: Actually, this is a perfectly grammatical sentence in English—for a telegram, a newspaper headline, or a diary entry, which is what the original text is. Adding a verb would force us to specify tense. Omitting the verb is therefore not only grammatical, but more accurate.

For the eclipse report on 1212-Nov-10, the authors have translated the 天晴 as "clarification sky". This

is more appropriately translated as "the sky was clear", as this was not "晴天". 云々 was omitted in their translation.

Reply: We cannot find the phrase "clarification sky" in the resubmitted manuscript. We have translated 天晴 consistently as "clear skies" and it should be so in this instance as well. In Japanese kanbun the phrase 云々 has a variety of functions; sometimes it marks the end of a quotation; sometimes it indicates that details have been omitted (in which case "blah blah" might be an apt translation); other times it signals the end of a sentence. We have taken the third sense and rendered it with a period.

"大陰虧" is translated as "total lunar eclipse". Not exactly true. This is better translated as "great lunar eclipse".

Reply: We find 大陰 attested in Koji Ruien 古事類苑 as a synonym for 月 'moon.' Moreover, in Japanese it is pronounced "taiin" and is homophonous with the very similar and much more common 太陰 'moon', which may be intended here. The two compounds differ by only a dot.

"建曆二年十月九日" not as "Kenryaku 2.10.19" but as "Kenryaku 2.10.9".

Reply: We thank the referee for spotting the typo. The correction has been made.

There are many such cases. I am afraid but their translations may needlessly reduce the reliability of the overall discussions which is not actually relying on the Japanese records that much. In this sense, it is probably more appropriate to remove this section.

Reply: We would be grateful to have further details on the other cases so we can address them. We also would like to stress that two translations of the same text will never be exactly identical. Speaking about 5 modern translations of Aristophanes, Prof. Alan H. Sommerstein said in 1973:

There will never be a perfect translation of Aristophanes. There have been many translations, with many and diverse merits: one thinks of the poetical grace of F. L. Lucas, the deftness of Godley and Bailey, the easy naturalness of John and Patricia Easterling, the zest of B. B. Rogers, the modernity of Douglass Parker and his associates; but combine all these and you are still left a long way from a perfect translation, for there is much truth in the paradox that the only really perfect translation is the original.

Sommerstein, A. H., 1973. On Translating Aristophanes: Ends and Means. Greece & Rome, 20(2), 140–154.

The same holds true for ancient Chinese, Korea and Japanese records. And in this respect, the way we translated East-Asian lunar eclipse records will always slightly differ from the way the referee or other authors would translate the same text.

Yuasa articles have not been cited either in the main manuscript or the supplementary information, whereas these articles are what the authors relied on for Azuma Kagami for their eclipse identification. This is not the best practice. They should cite Yuasa articles in the main manuscript.

Reply: The following article by Yuasa was cited in the Excel sheet entitled "Dataset_S1_Japan" submitted in March 2022 (see cell X269):

Yuasa, Yoshimi 湯浅吉美. "Chūseibito no gesshoku-kan: 'Gyokuyō' to 'Azuma kagami' no kiji kara mite 中世びとの月蝕観：『玉葉』と『吾妻鏡』の記事から見て. Saitama Gakuen Daigaku kiyō: Ningen Gakubu-hen 埼玉学園大学紀要. 人間学部篇 10 (December, 2010), pp. 63-76. [Article on medieval Japanese views of lunar eclipses using entries from the texts Gyokuyō and Azuma kagami]

14. Islamic records

If they wish to use these records in their Dataset S1, they should show the volume number and page number of the original Arabic chronicles. Moreover, they should show the original Arabic text, as done for the occidental records and East Asian records.

Reply: We didn't include any records in the Dataset S1 due (1) to the limited number of lunar eclipse records (approximately 6, according to the survey made by Stephenson and Said in 1997), (2) because the records are readily accessible and (3) because none of the reports provides information about lunar eclipse color.

The same can be said to the Russian records.

Reply: The original texts have been added to the Dataset_S1.

For the Syriac records, the original texts are missing too.

Reply: The original texts have been added to the Dataset_S1.

15. East Asian records themselves

These datasets have not played significant scientific roles in their main manuscript and involve a number of philological flaws. IMHO, this dataset will needlessly reduce their philological reliability. Moreover, if they wish to publish a 'the most complete compilations of lunar eclipse observations for the 12th and 13th centuries', they should do so in an astronomical journal with an appropriate title (not as a paper for volcanic studies).

Their supplementary text 2 explicitly confirms the lunar eclipse record in 1258 is not usable for their discussion: "Given these circumstances, we did not attribute any luminosity value to this reported eclipse."

Reply: The lunar eclipse of 1258 recorded in the Azuma Kagami indeed cannot be used to estimate past stratospheric turbidity. However, we believe this record is of great interest. It is important to show that we considered this record, as well as the reasons that led us to discard it. Otherwise at a later stage some readers aware of this account may wonder why we didn't mention it.

Moreover, the authors cite this account from Nihon Tenmon Shiryo, a compilation in 1935, as if it were a contemporary historical account.

Reply: The citation has been replaced with the original reference.

The authors have not addressed similar accounts "it was not properly visible" in Azuma Kagami, although Yuasa articles have explicitly listed these cases. In these regards, I think it is inappropriate to cite Azuma Kagami account for the eclipse record for their source 'data' table.

Again, I do not think the East Asian have played any scientific roles in their article, except for the Japanese lunar eclipse record in 1229. Therefore, they should drop their description about the East Asian records in the supplementary materials and just specifically cite a case report from 1229.

Reply: Referring to only one single text from East-Asia could lead some readers to conclude we were overly selective, ignoring hundreds of other records of lunar eclipses available in East Asia. We consider it important to show that we considered all available records, and even more so to show that we didn't discard any record simply because it didn't fit our narrative.

The aim of this dataset is to be as transparent as possible and to make our research reproducible. Our dataset may not be perfect but it does offer the reader the possibility to validate or disprove our hypotheses.

They should also replace the Azuma Kagami citation in Supplementary Text 2 to something from the occidental chronicles. Azuma Kagami is based on a different cultural background, in contrast with the occidental chronicles. They should visualize their "careful treatment and interpretation" for the historical records on the basis of the occidental records which they have actually used in this manuscript.

Reply: We fully agree with the referee. The Azuma Kagami, and more generally all East Asian records, reflect a distinct cultural background. Therefore, one should be even more careful when using these sources to assess the impacts of volcanic eruptions on climate.

For this reason, we feel that taking the Azuma Kagami as an example is appropriate.

The authors have also listed a number of references showing that identifying dry fogs and dust veil in medieval European sources can be challenging.

Minor Comments

P2L51: food crises => great famines?

Reply: The link between large volcanic eruptions is complex. It may be better to keep the formulation "food crisis".

P2L67: Bishop's rings => e.g., Bishop's rings

Reply: The sentence has been modified.

P3L87-91: Rather than describing East Asian records which are not much used here, the authors should

explain how they have collected that amount of the European chronicles in this study in which selection criteria.

Reply: We feel that it is important to state that East-Asian recorded were considered during our investigation. Therefore, we prefer to retain the paragraph written page 3, lines 87-91. Regarding the European sources, there is little to add. The first author carefully and thoroughly examined for 10 years the annals and chronicles written in the Monumenta Germaniae Historica, Rerum Britannicarum Medii Aevi, Recueil des historiens des Gaules et de la France as well as in the Rerum Italicarum scriptores. He checked each source one by one. Each time he found a report, he transcribed it in a database, then tried to assess whether this report was original or whether it was a duplicate found in another sources. This information can already be found in the “Methods” section.

P3L110: Middle Eastern => Eastern Christianity?

Reply: Agreed. The sentence has been changed.

P11L444: Not 'original records' but 'transcriptions'

Reply: Unfortunately, we are not able to locate the text the referee would like us to change.

Once again, we would like to thank the referee. Although we do not concur with all of referee’s points, we sincerely appreciate his/her close attention to the manuscript and care.

References:

Breuker, R., Koh, G., and Lewis, J. B.: The Tradition of Historical Writing in Korea, in: The Oxford History of Historical Writing: Volume 2: 400-1400, vol. 1, edited by: Foot, S. and Robinson, C. F., Oxford University Press, <https://doi.org/10.1093/oso/9780199236428.003.0007>, 2012.

Bruneton, Y.: Koryŏsa 高麗史 고려사: l’histoire officielle du Koryŏ, in: Encyclopédie des historiographies : Afriques, Amériques, Asies, edited by: Kouamé, N., Meyer, É. P., and Viguier, A., Presses de l’Inalco, 973–988, <https://doi.org/10.4000/books.pressesinalco.26386>, 2020.

Butler, L.: Emperor and aristocracy in Japan, 1467-1680: resilience and renewal, Harvard University Asia Center : Distributed by Harvard University Press, Cambridge, Mass., 2002.

Duthie, T.: *Man’yōshū* and the Imperial Imagination in Early Japan, BRILL, <https://doi.org/10.1163/9789004264540>, 2014.

Gwangchul, K.: The Printed Books of 『Goryosa(高麗史)』 and Its Publication Date, J. Seokdang Acad., 145–174, <https://doi.org/10.17842/JSA.2014..59.145>, 2014.

Miner, E. R., Morrell, R. E., and Odagiri, H.: The Princeton companion to classical Japanese literature, Princeton University Press, Princeton, N.J, 570 pp., 1985.

Park, S.: A New Path for the Study of the Koryŏ Dynasty: Exploring the Future of Online Historical Source Archives, Int. J. Korean Hist., 24, 47–70, <https://doi.org/10.22372/ijkh.2019.24.2.47>, 2019.

Piggott, J. R., Yoshida, S., and Fujiwara, T. (Eds.): *Teishinkōki: the year 939 in the Journal of regent Fujiwara no Tadahira*, East Asia Program, Cornell University, Ithaca, NY, 264 pp., 2008.

Sommerstein, A. H.: *On Translating Aristophanes: Ends and Means*, *Greece & Rome*, 20, 140–154, 1973.

Sorimachi, S.: *Kōbunsō keiaisho zuroku. II* (弘文莊敬愛書図録. II, Images of respected and beloved books of Kōbunsō, vol. 2), Kōbunsō, Tokyo, 99 pp., 1984.

Stark, P. B.: *Before reproducibility must come preproducibility*, *Nature*, 557, 613–613, <https://doi.org/10.1038/d41586-018-05256-0>, 2018.

Steele, J. M.: *Observations and Predictions of Eclipse Times by Early Astronomers*, Springer Netherlands, Dordrecht, 2000.

Stephenson, F. R.: *Historical Eclipses And Earth's Rotation*, Cambridge University Press, Cambridge, <https://doi.org/10.1017/CBO9780511525186>, 1997.

Stephenson, F. R. and Said S. Said: *Records of Lunar Eclipses in Medieval Arabic Chronicles*, *Bulletin of the School of Oriental and African Studies, University of London*, 60, 1–34, 1997.

Reviewer Reports on the Second Revision:

Referee #2 (Remarks to the Author):

Summary:

I agree that this article has significantly updated chronologies of medieval volcanism, using historical reports for total lunar eclipse. However, this manuscript is still suffering from serious flaws on the astronomical and historical aspects that form backbone for the article methodology. The authors cannot derive accurate estimates for lunar eclipses without accurate identifications of observational sites. The authors have admitted that they have given up such identifications, whereas they are still maintaining 'Historical Lunar Eclipse Atlas' and success proportions. Moreover, I have detected previously overlooked flaws here. While they have reportedly included only total-eclipse reports, what they are actually doing is significantly different (Major Comment 3). As a result, they have involved enormous contaminations from probable partial eclipses in their database files. The authors have failed addressing how they have removed prediction reports for East Asian records either. There are still a number of philological flaws in their database file. Overall, I have to confirm that this manuscript fails satisfying publication thresholds on astronomical aspects (overall) and philological aspects (especially for East Asian records).

1. Title:

The authors have improved the title to some extent. However, the authors are overemphasizing their access to oriental records. The authors have mainly used the occidental records here, as they have agreed as well. On contrary, the authors have used only one East Asian record for their discussion (one from Kyoto in 1229). For this manuscript, the occidental records have played a main role, whereas the East Asian records have played only little -- or at best supporting -- role. The authors cannot claim equal emphases for their use on the occidental and East Asian records. They have to revise the title as: "[Occidental] lunar-eclipse [record]s illuminate timing and climate impact of medieval volcanism".

2. Observational sites

2.1. Clarification is needed in the main text

The authors have admitted that the sites in their datasets are not necessarily the observational sites. Accordingly, they first have to explicitly clarify this difficulty in the main text of their manuscript, appropriately citing some case studies (e.g., DOI: 10.1177/00218286221097111 and DOI: 10.5194/hgss-11-81-2020). These caveats are too grave to mention in one of their supplements. At this rate, casual readers may misunderstand that the authors had successfully identified observational sites for these lunar eclipses.

2.2. Concerns on observational sites of the occidental eclipse reports

In fact, the authors claim, "we still consider that providing coordinates is useful". This would be true, if they had carried out philological case studies for individual eclipse reports, as done in DOI: 10.1177/00218286221097111 and DOI: 10.5194/hgss-11-81-2020, citing these previous studies. Of course, some eclipses were probably seen in the sites where the chronicles were written, whereas the authors have not identified which are such cases and have not explained why they consider so. Unfortunately, the authors have tried a shortcut for discussions on the observational sites in their

manuscript. Without individual careful case studies, these coordinates would at best mislead the readership. They have to remove the 'Eclipse Atlas' and location columns.

2.3. Concerns on observational sites of the East Asian eclipse reports

For East Asia, DOI: 10.1177/0021828618789850 has clarified that the eclipse reports from Astronomical Treatises are from imperial capitals but those from imperial annals are not necessarily from the imperial capital. These issues have not been taken into consideration in this manuscript. Therefore, I cannot remove my concern on the authors' columns for the geographical coordinates.

2.4. Dating errors in their database

Such uncertainty has inevitably affected their local time calculations. In fact, the authors have removed their timing calculations. That was a good move. However, such issues have affected their eclipse dates too. For example, for the total lunar eclipse in 1222 Oct, the authors have dated this eclipse on 1222 Oct 21, whereas my calculation indicates its maximum at Milan at 02:34 on 1222 Oct 22 in the apparent solar time. This is just a case. There are more mistakes with similar kinds. Such discrepancies are not tolerable for publication.

3. Do the database entries really show reports for total lunar eclipses?

The authors have described, "Note that accounts referring to penumbral and partial eclipses were excluded from analysis as only total lunar eclipse observations are suited to this method" (P13 L511). This is extremely important, as the authors derive Danjon scale for each total-eclipse account to estimate the atmospheric turbidity. Following this description, the authors must involve only total-eclipse reports.

However, the authors have done in the database files is significantly different from their own descriptions. Their database actually shows lunar eclipse records on the dates of total lunar eclipses. Accordingly, the authors have failed clarifying if these reports indeed recorded total lunar eclipses or not. In East Asia, the eclipse totality has been described as "既 (ji)" (e.g., Stephenson, 1997; DOI: 10.1177/0021828618789850). The database has involved reports without ji as well (e.g., 1117-Dec-11 and 1118-Jun-05). This has clearly caused some contaminations from partial eclipses. For example, on 1118-Jun-05, the eclipse report stated, "Emperor Huizong (Hūizōng 徽宗) of Song, 8th year of the Zhenghe (Zhènghé 政和) reign period, 5th month, day bingshen (bǐngshēn 丙申); the moon was 9 fen (fēn 分) eclipsed". The moon was only "9 fen (fēn 分) eclipsed". This is clearly a partial eclipse.

This contamination is significant in their database. For example, roughly counting, in Chinese and Korean records, only 30 in 70 entries and 12 in 41 entries have involved this marker, respectively. I have not counted them in Japanese records, as their entire cell contents were difficult to display in my computer. This contradicts their methodological descriptions. Therefore, the authors should just list up what has been described as total eclipses. The authors also have to clarify what criteria they have used to select reports for total eclipses from the European historical documents.

4. Eclipse visibility condition

The authors have claimed, "we feel it is possible to estimate the proportion of observed and missed lunar eclipses in Europe". In order to claim so, they need to provide an error bar for eclipse

magnitudes at each case, clarifying the expected geographical extents of each observational site for each eclipse report with convincing explanations. The authors have claimed, "In most cases, lunar eclipse observations were generally made near the monasteries that reported them". If this were truly the case, the authors would be able to clarify their geographical extents and provide error bars for magnitude of each lunar eclipse. Without such explanations, they cannot provide a success proportion for the eclipse observations.

In fact, the authors' assumptions are not quite convincing, as British chronicles have involved some celestial reports even from Northern France and Italy (DOI: 10.1177/00218286221097111 and DOI: 10.5194/hgss-11-81-2020). Unfortunately, I am rather skeptical about the authors' claims, as some British reports were imported all the way from Italy and Northern France.

Can we consider such geographical uncertainty something negligible? That is not true. For example, the authors have assumed the observational site as Bayern and Colmar for the lunar eclipse on 1241 April 27. Of – by any chance – their provenances are expected from Königsberg, the moon would have been set (04:15) before the onset of total eclipse (04:46). As shown in this example, lunar eclipses can get significant variations for the visibility condition even in the European sector. Here, I am not claiming this report is actually from Königsberg. Rather, I am casting caveats on the geographical uncertainty. One of solar eclipse report in the Anglo-Saxon Chronicle was seemingly derived from Italy. As such, it is far from secure to assume their geographical uncertainty to be negligible in their studies. Therefore, if they indeed insist maintaining their discussions on the success proportion, they must show geographical error margins for each eclipse report and resultant error bars for the success proportion for these eclipse reports.

5. 'Historical Lunar Eclipse Atlas' must be removed

The authors are still maintaining their 'Historical Lunar Eclipse Atlas' with some excuses. At least, the URL is still valid. However, if these sites are not observational sites, this atlas cannot be called as a 'Historical Lunar Eclipse Atlas', as the contents contradict the title. Some readers could be easily misled that this were truly an 'eclipse atlas'. This is scientifically incorrect. They must remove this misleading online supplement at <https://arcg.is/Xy0KX>.

6. The authors' discussions on the proportion of observed and missed lunar eclipses

Their discussions on the success proportion are overemphasized. Their paragraphs in P3 L102-110 should be either removed or completely rewritten. Firstly, without knowing the exact observational sites, it is extremely difficult to assess if certain observers managed to see given lunar eclipses from certain observational sites. Of course, in many cases, the European sectors may have been completely covered by the eclipse visibility zones. However, there are some marginal lunar eclipses where Europe is situated in the visibility boundary. Even slight ΔT modifications can change visibility conditions for total lunar eclipses (DOI: 10.1111/j.1468-4004.2005.46511.x).

It would be far-fetched to derive exact percentage without exact observational sites (See Major Comment 4 too). After all, the authors have avoided identifications of the observational sites, as they have explicitly clarified in the previous rebuttal letters.

Secondly, there are significant contaminations of prediction records, especially in the East Asian

records (See Major Comment 7). The authors have not distinguished the observations from predictions. Then, it is quite possible that the alleged high success rate in East Asia is not because of their observational efforts but because of their accurate prediction capability. Therefore, the authors cannot derive success rates on the basis of what they have provided.

Thirdly, the authors have not clarified the cases of impossible eclipse reports (e.g., those with wrong dates). This has been detailed in another major comment. Apparently, the authors have coupled eclipse records with the lunar eclipses in Espenak and Meeus (2009). Then, the unreal eclipse reports were omitted here and exaggerated the 'success rate' as well.

7. Contaminations of prediction reports in East Asian records

The authors' "data" from East Asia are seriously contaminated by prediction reports. The authors have claimed: "The proportion of eclipses recorded by observers in Asia is also substantial (91%) (Extended data Table 2), motivated by the potential of astronomical phenomena to act as portents with political significance". However, it is far-fetched to claim them as "recorded by observers in Asia" without any critical assessments, as East Asian eclipse records involve significant amounts of predictions among them.

This danger is clearly confirmed in the data table too. For example, on 1103-09-17, Honchō seiki reported that the lunar eclipse did not appear owing to cloud cover. This is clearly a prediction. The authors have admitted this too, while they have listed this report as an one of "eclipses recorded by observers in Asia". This is a self contradiction, as the authors have listed these database "in search of credible lunar eclipse observations".

In fact, there are a number of previous studies which have confirmed significant contaminations of eclipse prediction reports in the East Asian historical documents. Foley (1989) have analysed solar eclipse records in the Chinese and Korean historical records and concluded that many of them were not actually visible from China or Korea.

This was also the case with Japanese eclipse records. Soma and Tanikawa (2015) analyzed Japanese eclipse records and concluded as follows: "The reliabilities of eclipse records between 7th and 11th centuries were discussed in this paper. Eclipses in the 7th century were recorded in the Nihongi and their reliability changed according to the volumes, namely the records in the volumes in the β group were based on actual observations whereas those in the α group were based on predictions at the time. Eclipses in the 8th century were recorded in the Shoku-Nihongi but most of them were written according to the predictions at the time. Eclipses between the 9th and 11th centuries were recorded in various literatures and diaries but it is not easy to decide whether they were based on actual observations or not." In fact, Yuasa (2010) have identified 4 impossible eclipse records among 37 realistic eclipse records in Azuma Kagami and 1 impossible eclipse records among 19 realistic eclipse records in Gyokuyo.

These cases have evidently shown that there are significant amount of prediction reports among the East Asian eclipse records. Of course, some of the eclipse predictions were successful enough. In this case, there should be numerous prediction reports in what the authors have tabulated as their 'data base'. However, the authors have not clarified which records are actually on the observational basis

(not predictions) at all. Contaminations of the predictions records do not allow us to derive any success rates and endanger reliability of the report accuracy as 'observations'. This is a fatal flaw for the authors' datasets for the East Asian records, which cannot be tolerated in astronomical and philological contexts.

7. East Asian "records"

The authors claimed to maintain East Asian 'records', while they have also admitted by themselves that the East Asian records played little role in this manuscript. However, their stance self-contradicts their own stance for Islamic records. According to the authors, the authors "didn't include any records in the Dataset S1 due (1) to the limited number of lunar eclipse records (approximately 6, according to the survey made by Stephenson and Said in 1997), (2) because the records are readily accessible and (3) because none of the reports provides information about lunar eclipse color". Apart from (1), the situation is virtually the same with East Asian records. In fact, for (2), the East Asian records are readily accessible with previous publications such as Xu et al. (2000) and Kanda (1935). For (3), none of these reports provide information about lunar eclipse color except for a single Japanese report. As they are omitting Arabic records in this logic, they have to be self-consistent to East Asian records too. This self-contradiction should not be tolerated.

The authors paid little attention to the probable contaminations of the prediction records although they cited East Asian records "in search of credible lunar eclipse observations". Moreover, their translations are suffering from numerous philological flaws and needlessly reducing the reliability of their discussions. If they still insist on claiming their searches on East Asian records, they should just cite Xu et al. (2000) for Chinese and Korean records and Kanda (1935) for Japanese records. These publications have already provided the text. The readers do not need to 'trust' the authors but just need to consult these previous publications.

Moreover, the authors are not showing "all the lunar eclipse records [they] considered". For example, for the East Asian records, the authors have already 'cherry-picked' the eclipse reports following Espenak and Meeus' database. As shown in Yuasa (2010), there are some impossible eclipse reports (see Major Comment 6), which has been ignored in the authors' dataset.

In fact, at the beginning, the authors have shown translations for what they had been interested in in the East Asian records. This "cherry-picking" has been modified only after I had raised some concerns.

8. Dataset S1

They should remove columns for Lat, Long, Country, and locality. This is because it is misleading to provide geographical coordinates for where chronicles were written as "locality". For example, Anglo-Saxon Chronicle has involved a number of astronomical reports even from outside of England, as exemplified in some eclipse records (seemingly from Rome; DOI: 10.1177/00218286221097111) and halo records (seemingly from Northern France; DOI: 10.5194/hgss-11-81-2020).

In this regard, it looks inappropriate to classify these records into countries in the modern definition. Even worse, back in the medieval periods, these countries have had different borders. It looks slightly weird to separate Scotland from the United Kingdom. The authors may wish to emphasize a

benefit for modern readers. However, this crude classification will be probably subjected to criticisms from medieval historians. If they wish to keep their columns for the countries, they need to clarify the country name at that time.

Dataset S1 is entitled as Continental Europe, while it involves reports from the United Kingdom and Continental Europe. They may need to either remove the British records from their supplementary files or add one more dataset tab to accommodate British records.

9. Continental European records

Here, in many cases, their English translations are missing. This stance significantly contradicts their claims on the values of English translations for the East Asian records.

10. Do they need English translations for East Asian records?

The authors have cited Sommerstein (1973) here to justify their philological flaws.

"There will never be a perfect translation of Aristophanes. There have been many translations, with many and diverse merits: one thinks of the poetical grace of F. L. Lucas, the deftness of Godley and Bailey, the easy naturalness of John and Patricia Easterling, the zest of B. B. Rogers, the modernity of Douglass Parker and his associates; but combine all these and you are still left a long way from a perfect translation, for there is much truth in the paradox that the only really perfect translation is the original."

Sommerstein, A. H., 1973. On Translating Aristophanes: Ends and Means. *Greece & Rome*, 20(2), 140–154.

Sommerstein is also right. I appreciate this opinion. Then, in this case, why do they need these inaccurate translations here? As the authors have also agreed, their translations for East Asian records have played little roles for their manuscript. Infact, they are mostly irrelevant. They have numerous philological flaws. The authors have treated translations of East Asian records too casually. Unfortunately, their translations only reduce their data reliability needlessly. They should just remove these needless tables for East Asian materials which was not actually used in this manuscript with a single exception (a Japanese report in 1229).

11. Korean records

I disagree with the authors' claims. The authors should refer what is permanently accessible. If the URL gets changed, everything will lose traceability. This happens to many databases. This is especially the case when the readers need to consult individual pages. Yes, it happens even to the NGDC websites! They need to refer what is permanently accessible. This is important, especially if the authors intend this article to be consulted not only over coming years but also over coming decades.

There are numerous philological flaws too.

For example, the authors should have transcribed Korean terms not to Mandarin but to Korean, as they have also admitted. However, for example, on 1132-Mar-04, the authors have transcribed "丙子" as "bingzi" in Mandarin, although they have claimed modifying such issues.

The authors have wrongly used square brackets. For example, on 1168-Mar-25, the authors have translated "密雲不見" as "it could not be seen because of the [dense] clouds", although "密" indicates "dense".

The authors have inconsistently translated similar phrases. For example, on 1277-May-19, the authors have translated "雨不見" as "it could not be seen because of the rains" in a plural form. However, on 1259-May-08, the authors have translated "因雨不見" as "but it was not seen because of the rain" in a single form. What is the cause if this discrepancy? Moreover, the translation of "因" is not well clarified here.

They are just initial examples. There are more philological flaws here.

12. Chinese records

There are still numerous philological flaws here. Here are only a few examples.

Some phrases like "不見" have been translated inconsistently. For example, on 1208-Feb-03, the authors translated "雨不見" as "it was not seen because of the rain". However, on 1186-Apr-05, the authors have translated "陰雲不見" as "it was hidden by the [shady] clouds [and was not seen]". As the authors are translating the same phrase, they should make such translations consistent (e.g., "it was not seen because of the shady cloud").

Some phrases with different wordings resulted in the same translation. For example, on 1219-Jun-29, the authors have presented two variants "雲陰不見" & "霧雲不見" but provided the same translations as "it was hidden by the [shady] clouds [and was not seen]". The authors should translate different Chinese characters (雲陰 and 霧雲) differently, although they certainly mean something similar with one another.

Some phrases added what its not included in the original text without any caveats. For example, on 1226-Aug-09, the authors have translated "陰雨不見" as "could not be seen because of the overcast and rainy weather". The word "weather" does not appear in the original text. To be more loyal to the original text, it could be something like "could not be seen because of cloud and rain".

Some phrases have been wrongly translated too. For example, on 1277-May-18, the authors have translated "寅四刻" as "By 4 marks in the hour of yin" whereas this should be "By 4th mark in the hour of yin".

They are just initial examples. There are more philological flaws here.

13. Japanese records

As I stated, the authors have to cite Yuasa (2010) in the main text. They have initially shortcut their access to Azuma Kagami using Yuasa (2010). They should appropriately appreciate Yuasa (2010) with their citation in the main text.

For the 1258 reports, the authors have claimed, "The lunar eclipse of 1258 recorded in the Azuma Kagami indeed cannot be used to estimate past stratospheric turbidity. However, we believe this record is of great interest. It is important to show that we considered this record, as well as the reasons that led us to discard it. Otherwise at a later stage some readers aware of this account may wonder why we didn't mention it". This statement is self-contradictory. The authors must remove this unusable report from their supplementary text 2. Moreover, this report does not involve a technical term to clarify eclipse totality (既). These difficulties do not allow the authors to claim this report as an account for a total lunar eclipse, unfortunately.

There are numerous philological flaws too.

For example, the authors have not appropriately translated "中略" on 1103-Sep-18.

Their translation style for Japanese records is inconsistent with those for Chinese and Korean records. For example, on 1103-Sep-18, the authors have translated "辰刻" as "hour of the Dragon", whereas they should have translated this as "Tatsunokoku" following their style to show local time descriptions with transcriptions as done in Chinese and Korean cases.

Some English translations are grammatically incorrect. For example, on 1103-Sep-18, they have translated an excerpt from Honchō seiki as "8th month, 14th day, lunar eclipse tonight, clouds covered it, did not appear". This translation is grammatically unacceptable.

The authors have ignored some phrases in the translations. For example, 勸脱力 is missing on 1103-09-17. This is probably this is an insertion by Kanda (1935). However, if this is the case, why has this been included in the 'original text'?

The local time descriptions are inconsistently given for the Japanese translations. For example, on 1114-Feb-21, they have shown a translation of "Boar [9p-11p]". I suspect p indicates pm. However, for Chinese text, it is for example shown as "zi {Time = 23-1h}". Such an inconsistency should be resolved.

Some phrases are missed in the translations. For example, on 1110-Oct-29, "誠如指掌" is missing in their translation.

14. West Asian records

They have omitted the Islamic records. If they insist in maintaining East Asian records, they must include Islamic records with original texts as well. They show a double standard for East Asian records and Islamic records.

The Syriac original texts are still missing, although the authors claimed to have tweaked this flaw.

15. Supplementary Text 2

Here, the authors have tried to show their 'careful treatments' on the historical records, comparing a western record with a Japanese record. However, as I have repeated, these records have different

cultural backgrounds and cannot be used for comparative references. Upon comparisons, we should compare two different objects with a similar background. Therefore, here, the authors have to show occidental records for their comparisons.

In the previous response, the authors have claimed, "We fully agree with the referee. The Azuma Kagami, and more generally all East Asian records, reflect a distinct cultural background. Therefore, one should be even more careful when using these sources to assess the impacts of volcanic eruptions on climate. For this reason, we feel that taking the Azuma Kagami as an example is appropriate." This statement looks self-contradictory and does not make sense to me, unfortunately.

Minor Comments

P2 L71: "and their occurrences known precisely from astronomical retro-calculation."

=> This sentence looks unclear. Please better rephrase what the authors have done for astronomical retro-calculations. Looking at P3 L96, the authors have tried a short cut with Espenak and Meeus' database. They should clarify that they have entirely relied the astronomical calculations on Espenak and Meeus (2009) and these database has some issues with the recent ΔT revisions (Stephenson et al., 2016; Morrison et al., 2021).

P3 L90-92: Again, the authors should remove the mentions to the East Asian records. The East Asian records have not played significant roles in their manuscript. If they still insist, they should just mention their comparison for the European records with the Japanese report in 1229.

P3 L92: Here, the authors have to explicitly clarify that they have tried a short cut with Kanda (19365) to avoid comprehensive investigations for original historical records. This sentence will mislead the readers, as if the authors have examined a number of Japanese historical records by themselves.

Additional References [those without DOI]

Foley, N. B. 1989, A statistical study of the solar eclipses recorded in Chinese and Korean history during the pre-telescopic era, M.Sc. thesis, Durham, University of Durham.

Sôma, M., Tanikawa, K. 2015, Reliability of Eclipse Records in Japanese Ancient Periods, Proceedings for Intensive Workshop on Ancient and Medieval Eclipse Data (2014 Nov 27—28), 2014, 5.
[<https://www2.nao.ac.jp/~mitsurusoma/WS2014/soma.pdf>]

Author Rebuttals to Second Revision:

Referees' comments:

Referee #2 (Remarks to the Author):

Summary:

I agree that this article has significantly updated chronologies of medieval volcanism, using historical reports for total lunar eclipse.

Reply: We sincerely thank Referee#2 for his/her comments and for the time and effort s/he took to review our manuscript.

However, this manuscript is still suffering from serious flaws on the astronomical and historical aspects that form backbone for the article methodology. The authors cannot derive accurate estimates for lunar eclipses without accurate identifications of observational sites. The authors have admitted that they have given up such identifications, whereas they are still maintaining 'Historical Lunar Eclipse Atlas' and success proportions.

Reply: The revised version of the manuscript we resubmitted in July 2022 did not include the Historical Lunar Eclipse Atlas. Indeed, we maintain that it is possible to estimate the proportion of observed and missed lunar eclipses. For more information, we would refer Referee#2 to Section 4.

Moreover, I have detected previously overlooked flaws here. While they have reportedly included only total-eclipse reports, what they are actually doing is significantly different (Major Comment 3). As a result, they have involved enormous contaminations from probable partial eclipses in their database files.

Reply: This statement is not correct. There is no contamination from partial lunar eclipses. No penumbral or partial lunar eclipses were used to estimate stratospheric turbidity. For more information, please see Section 3.

The authors have failed addressing how they have removed prediction reports for East Asian records either.

Reply: We do not agree with the referee. We have identified predictions from East Asian sources. The prediction records were excluded from the analysis as described in the methods section. For more information, we refer Referee#2 to Section 7.

There are still a number of philological flaws in their database file. Overall, I have to confirm that this manuscript fails satisfying publication thresholds on astronomical aspects (overall) and philological aspects (especially for East Asian records).

Reply: We do not agree with this statement. The East Asian records are philologically sound. For more information, see sections 10, 11, 12 and 13.

1. Title:

The authors have improved the title to some extent. However, the authors are overemphasizing their access to oriental records. The authors have mainly used the occidental records here, as they have agreed as well. On contrary, the authors have used only one East Asian record for their discussion (one from Kyoto in 1229). For this manuscript, the occidental records have played a main role, whereas the East Asian records have played only little – or at best supporting – role. The authors cannot claim equal emphases for their use on the occidental and East Asian records. They have to revise the title as: “[Occidental] lunar-eclipse [record]s illuminate timing and climate impact of medieval volcanism”.

Reply: Since the text pertaining to the dark lunar eclipse of 1229 was written in Kyoto by the Japanese courtier Fujiwara no Teika, the use of the word "occidental" in the title is not appropriate. We further note that titles in *Nature* cannot exceed 75 characters (including spaces). If we add "Occidental" and "record", we arrive at 95 characters. In the event that the manuscript is accepted for publication, we are confident that the editorial board will also provide advice and guidance on the title of the manuscript.

2. Observational sites

2.1. Clarification is needed in the main text

The authors have admitted that the sites in their datasets are not necessarily the observational sites. Accordingly, they first have to explicitly clarify this difficulty in the main text of their manuscript, appropriately citing some case studies (e.g., DOI: 10.1177/00218286221097111 and DOI: 10.5194/hgss-11-81-2020). These caveats are too grave to mention in one of their supplements. At this rate, casual readers may misunderstand that the authors had successfully identified observational sites for these lunar eclipses.

Reply: From the comments received from Referee#2, it appears s/he has been using historical observations of eclipses to estimate variations in the Earth's rotation for many years. Reliable ΔT estimates indeed require precise locations for observations (Stephenson, 1997). However, we emphasise that the methods and requirements that may apply to the field of the referee are not necessarily pertinent to our case, for which fixing observational sites with precision is important but not critical. To estimate past stratospheric turbidity and to refine the dating of past volcanic eruptions, we primarily need to know when the eclipse occurred, that it was total and to have detailed information about the colour and luminosity of the Moon during the eclipse. Accordingly, we do not wish to overdo this topic in the main text. We are also mindful of word and reference limits.

Nevertheless, we have revised the methods section as well as Dataset S1 (see Excel spreadsheet "Readme") to further respond to the referee's concerns. The Readme file also includes the two references suggested by the referee.

We reiterate that nowhere in the manuscript or supplementary material did we claim that we had successfully identified observational sites for all the lunar eclipses studied.

2.2. Concerns on observational sites of the occidental eclipse reports

Reply: We feel that the concerns of Referee#2 regarding the observational sites are not entirely justified.

Would the annalists and chroniclers from Europe have reported total lunar eclipses if the eclipsed Moon had not been visible at all? Would they have described the colour of lunar eclipses in so much detail if conditions had not been good enough to see the eclipsed Moon?

We emphasise that we have carefully assessed every single eclipse report found in historical sources. We checked the date accuracy of each report against the Five Millennium Catalogue of Lunar Eclipses (1999 BCE-3000 CE) (Espenak and Meeus, 2009) and the Eight Millennium Catalogue of Lunar Eclipses (4000 BCE - 4000 CE) (Liu, 2021). We also ensured that all reported eclipses were visible in Europe by using the visibility maps and the local circumstances

tables provided by both catalogues

We add that very often lunar eclipses were reported and described in more than one historical source. For example, the lunar eclipse of 12 February 1161 was described in 16 different sources (see Dataset S1). These span several countries, e.g. France, England, Germany, Ukraine and even Spain. A critical evaluation of the sources suggests that at least eight of these 16 descriptions are entirely original (i.e., independent of each other) and that the eclipses were very likely observed in or near the monastic centre where the texts were written (see Dataset S1 for more information). Further, if we inspect the visibility map of Liu et al. (2021) (Figure 1), it is evident that the eclipse of 12 February 1161 was visible in most parts of North and South America, Africa and Europe (from Spain to Ukraine). This is only one example, but it

demonstrates how we have given full consideration to observational sites.

Total Lunar Eclipse on 1161-Feb-12 (TD)

Figure 1. Visibility map of the total lunar eclipse of 12 February 1161.
Source: Liu (2021)

In fact, the authors claim, “we still consider that providing coordinates is useful”. This would be true, if they had carried out philological case studies for individual eclipse reports, as done in DOI: 10.1177/00218286221097111 and DOI: 10.5194/hgss-11-81-2020, citing these previous studies. Of course, some eclipses were probably seen in the sites where the chronicles were written, whereas the authors have not identified which are such cases and have not explained why they consider so.

Reply: We feel that this comment doesn't do justice to our work. We made significant efforts to trace the origins of each lunar eclipse report that we found and to assess whether the reports were contemporary and original or later duplicates, an issue we are very well aware of. Whenever possible, we also provided information about the author of each source and even tried to identify whether the author was an eyewitness or not.

Our work can be found in Dataset S1. We would invite the referee to read the column “Source Description” in the Excel sheet entitled Dataset_S1_Europe. A few examples are also provided below (Figure 2):

AA	AB	AC
References	Source description	Original reports?
Cosmae chronica Boemorum – Canonici Wissegradensis continuatio (1126-1142), KOFERKE R. (ed.), 1851. Monumenta Germaniae Historica, Scriptores 9, Hanover, pp. 132-148.	The Canonici Wissegradensis continuatio was written in Latin by an unidentified canon of Vyšehrad. It spans the years 1126–1142 and comprises a continuation of the Chronica Boemorum of Cosmas of Prague. Of particular interest are the reports of celestial and weather-related events, for which it deserves to be better known. The author was perhaps of only modest education (e.g., not evincing familiarity with the names of planets) but is quite accurate in his reporting. The value of his account is enhanced in being an eyewitness to many of the events reported, or in drawing from apparent eyewitness testimony of others (Bláhová, 2016). The entry referring to the 1128 total lunar eclipse is original.	Yes
References	Source description	Original reports?
Roberti Canonici S. Mariani Autissiodorensis Chronicon, HÖLDER-EGGER O. (ed.), 1882. Monumenta Germaniae Historica, Scriptores 26, Hanover, pp. 219-276.	The Roberti Canonici S. Mariani Autissiodorensis Chronicon was written by the chronicler Robert of Auxerre. Robert was a monk and canon at the Abbey of St. Marianus (Auxerre, France). He wrote his chronicle at the request of his abbot, Milo of Trainel (1156-1202). The chronicle covers the period between the creation of the world and 1211. For the years previous to 1181, the work is merely a compilation from Prosper of Aquitaine, Sigebert of Gembloux and his continuators, the Historia ecclesiastica of Hugh of Fleury, and the Chronicon quod dicitur Guillelmi Gadel, but it is an original authority for the period from 1181 to 1211. After 1181, Robert makes much use of oral and eyewitness evidence, including letters from the crusaders (Rech, 2016). The entry pertaining to the 1204 total lunar eclipse is original.	Yes
References	Source description	Original reports?
Reineri Annales (1066-1230), PERTZ G.H. (ed.), 1859. Monumenta Germaniae Historica, Scriptores 16, Hanover, pp. 651-680.	The Reineri Annales were written by Reinier (1157-1230), prior at the Benedictine Abbey of Saint Jacques at Liège (Wallonia, Belgium). The annals are a continuation of the Annals of Lambert le Petit (988-1193). They cover the years 1066-1230. The author explains at the beginning of the annals that he was born in 1157 and that he became subdeacon in 1175, deacon in 1179, monk in 1180, priest in 1191 and prior in 1197. The author shows great interest in climate and natural phenomena. His principal sources are eyewitness accounts. The annals are entirely original. (Dury, 2016). The entry referring to the lunar eclipse of 1208 is original.	Yes
References	Source description	Original reports?
Gransden A. (ed. and transl.), 1964. The chronicle of Bury St. Edmunds, 1212-1301, Nelson, London, 187p.	The Bury Chronicle was written at Bury Saint Edmund's Abbey, Suffolk, East, England. It narrates events from the Creation of the World till 1301. It was the work of at least three men. The first part of the chronicle of Bury Saint Edmund was written by John of Taxster who became a monk of Bury on 20 November 1244. Up to the annal for 1212, the chronicle is a compilation from numerous well-known works. Its primary source to 1131 is the chronicle of John of Worcester. From 1131 to the end of the twelfth century, its main sources are the chronicles of Ralph de Diceto and then the Annales Sancti Edmundi until 1212. From the last paragraph of the annal for 1212 until 1265, Taxster's chronicle is independent of all known literary authorities (Gransden, 2015). The author of the continuation from 1265 to 1296 has never been identified. However historians have suggested that the author may have been William of Hoo, sacrist of Bury St Edmunds, from 1280 to 1294, or someone from his office. The first continuation is written up more or less contemporaneously with the events recorded. The Bury St. Edmund's Chronicle is well-known for its numerous, accurate and original observations of lunar and solar eclipses. One solar eclipse (under 1261) and eleven lunar eclipses (under 1258, 1261, 1270, 1276, 1280, two under 1281, 1287, 1288, 1291 and 1297) were recorded. Most chronicles' dates are accurate (Roy et Stones, 1970).	Yes

Figure 2. Excerpts of the Excel sheet entitled “Dataset_S1_Europe”. The column “Source Description” provides information about the annals/chronicles, their author(s), where and when they were written, and whether the lunar eclipse description is original or if it is a second-hand report.

Unfortunately, the authors have tried a shortcut for discussions on the observational sites in their manuscript. Without individual careful case studies, these coordinates would at best mislead the readership. They have to remove the ‘Eclipse Atlas’ and location columns.

Reply: There is no short cut. Knowing precisely where the lunar eclipses were observed is important but not critical in order to estimate past stratospheric turbidity and refine the dating of past volcanic eruptions.

As mentioned many times in the previous revisions and in the Readme file of the Dataset_S1, we have been very cautious in compiling these data. The coordinates provided in the Excel sheet entitled “Dataset_S1_Europe” give information about the location where the sources were written. We see no impediment to a clear understanding of our procedures. The “Historical Eclipse Atlas” has also been removed during the last revisions as requested by the referee.

2.3. Concerns on observational sites of the East Asian eclipse reports

For East Asia, DOI: 10.1177/0021828618789850 has clarified that the eclipse reports from Astronomical Treatises are from imperial capitals but those from imperial annals are not necessarily from the imperial capital. These issues have not been taken into consideration in this manuscript. Therefore, I cannot remove my concern on the authors' columns for the geographical coordinates.

Reply: We do not follow the referee's concern. The geographical coordinates provided are entirely consistent with Stephenson (1998), Stephenson (1997), Stephenson (2002) and Stephenson (2018):

China

Almost all the eclipses records presented in Dataset_S1_China were retrieved from Astronomical Treatises, as specified in the Excel sheet entitled "Readme_Dataset_S1_China". Therefore, most of these records are probably from the dynastic capitals, which is consistent with Stephenson et al. (2018) and with our arguments in the earlier versions of the ms. We have also examined whether total lunar eclipses were visible in the capitals and made clear in Dataset S1 when this was not the case (please see the column entitled "accuracy of the eclipse reports").

Korea

The capital presented in the Dataset_S1_Korea is identical to the one used by Stephenson (1988) and Stephenson (2004). In these two papers, the author used the capital of Songdo (modern Kaesong) to compute the lunar eclipses visible in Korea between 918 and 1392 CE and assess completeness of the eclipse records in the *Goryeosa*.

Japan

In Japan, the imperial court was transferred to Kyoto in 784 CE and remained there until 1868 CE. Most of the courtier diaries we consulted appear to have been written in Kyoto. It is likely that several astronomical observations were performed there. This statement is again fully consistent with Stephenson (1997, see page 266).

2.4. Dating errors in their database

Such uncertainty has inevitably affected their local time calculations. In fact, the authors have removed their timing calculations. That was a good move. However, such issues have affected their eclipse dates too. For example, for the total lunar eclipse in 1222 Oct, the authors have dated this eclipse on 1222 Oct 21, whereas my calculation indicates its maximum at Milan at 02:34 on 1222 Oct 22 in the apparent solar time. This is just a case. There are more mistakes with similar kinds. Such discrepancies are not tolerable for publication.

Reply: We do not fully follow the referee's comment... There are no errors in the dating of the eclipses. The dates provided in the first column of Dataset S1 are based on the Five Millennium Catalogue of

Lunar Eclipses (Espenak and Meeus, 2009), as well as on the Eight Millennium Catalogue of Lunar Eclipses (Liu, 2021), as specified in the main manuscript and in the Readme of Dataset S1.

Let's now focus on the example taken by the referee. The eclipse indeed reached its maximum on 1222 Oct 22, but the referee overlooks that the eclipse actually started on 1222 Oct 21, shortly before midnight. Please see Table 1.

Calendar Date	Ecl. Type	Pen. Mag.	Umbral Mag.	Pen. Eclipse Begins	Alt (°)	Partial Eclipse Begins	Alt (°)	Total Eclipse Begins	Alt (°)	Mid. Eclipse	Alt (°)	Total Eclipse Ends	Alt (°)	Partial Eclipse Ends	Alt (°)	Pen. Eclipse Ends	Alt (°)
1212-Nov-10	P	1.768	0.785	15:08	-15	16:11	-6	—	—	17:38	7	—	—	19:05	21	20:09	32
1213-Apr-06	Pen	0.063	-0.951	23:26	32	—	—	—	—	00:03	33	—	—	—	—	00:39	33
1213-Oct-31	Pen	0.436	-0.595	04:07	30	—	—	—	—	05:39	15	—	—	—	—	07:11	0
1214-Sep-20	P	1.353	0.266	14:45	-32	16:23	-18	—	—	17:25	-8	—	—	18:26	3	20:04	20
1215-Mar-17	T	2.790	1.834	02:20	36	03:13	30	04:10	22	04:59	14	05:48	6	06:44	-4	07:38	-13
1215-Sep-09	T	2.674	1.602	14:22	-40	15:30	-30	16:36	-20	17:26	-12	18:17	-3	19:22	8	20:31	19
1216-Mar-05	P	1.575	0.588	17:56	-4	19:05	8	—	—	20:24	21	—	—	21:43	32	22:52	41
1216-Aug-28	P	1.731	0.713	20:06	12	21:17	22	—	—	22:45	32	—	—	00:13	36	01:24	35
1217-Feb-23	Pen	0.180	-0.866	05:25	18	—	—	—	—	06:27	7	—	—	—	—	07:29	-3
1217-Jul-20	Pen	0.571	-0.375	02:03	21	—	—	—	—	03:39	12	—	—	—	—	05:15	-1
1218-Jul-09	P	1.890	0.940	18:15	-17	19:14	-7	—	—	20:47	6	—	—	22:19	16	23:18	21
1219-Jan-02	T	2.755	1.681	11:57	-23	13:05	-23	14:08	-20	14:59	-16	15:50	-10	16:54	-2	18:02	8
1219-Dec-22	P	1.737	0.720	19:41	30	20:50	41	—	—	22:17	55	—	—	23:43	66	00:52	68
1220-Jun-17	Pen	0.945	-0.102	16:18	-39	—	—	—	—	18:33	-17	—	—	—	—	20:48	3
1220-Nov-12	Pen	0.243	-0.726	00:12	62	—	—	—	—	01:17	59	—	—	—	—	02:21	52
1221-May-08	Pen	1.032	-0.033	02:44	18	—	—	—	—	05:07	-2	—	—	—	—	07:29	-25
1221-Nov-01	P	1.420	0.427	14:00	-24	15:14	-16	—	—	16:22	-7	—	—	17:31	4	18:45	16
1222-Oct-21	T	2.713	1.667	23:45	57	00:50	56	01:52	50	02:42	44	03:32	36	04:34	26	05:39	15
1223-Apr-16	P	1.861	0.890	18:20	-9	19:22	1	—	—	20:53	14	—	—	22:25	24	23:27	29
1223-Oct-11	P	1.737	0.648	02:59	37	04:21	24	—	—	05:51	9	—	—	07:21	-5	08:43	-18

Table 1. Local Circumstances for Milan between 1212 and 1223. Eight millennium catalogue of Lunar eclipses (Liu, 2021). This table can be reproduced here: http://ytlui.epizy.com/eclipse/lunar_local.html

3. Do the database entries really show reports for total lunar eclipses?

The authors have described, “Note that accounts referring to penumbral and partial eclipses were excluded from analysis as only total lunar eclipse observations are suited to this method” (P13 L511). This is extremely important, as the authors derive Danjon scale for each total-eclipse account to estimate the atmospheric turbidity. Following this description, the authors must involve only total-eclipse reports.

Reply: We stand by our statement; no penumbral or partial lunar eclipses has been used to estimate stratospheric turbidity. The reader can very easily see that our statement is valid by checking Dataset S1.

However, the authors have done in the database files is significantly different from their own descriptions. Their database actually shows lunar eclipse records on the dates of total lunar eclipses. Accordingly, the authors have failed clarifying if these reports indeed recorded total lunar eclipses or

not. In East Asia, the eclipse totality has been described as “既 (ji)” (e.g., Stephenson, 1997; DOI: 10.1177/0021828618789850). The database has involved reports without ji as well (e.g., 1117-Dec-11 and 1118-Jun-05). This has clearly caused some contaminations from partial eclipses. For example, on 1118-Jun-05, the eclipse report stated, “Emperor Huizong (Hūizōng 徽宗) of Song, 8th year of the Zhenghe (Zhènghé 政和) reign period, 5th month, day bingshen (bǐngshēn 丙申); the Moon was 9 fen (fēn 分) eclipsed”. The Moon was only “9 fen (fēn 分) eclipsed”. This is clearly a partial eclipse.

Reply: We disagree. The lunar eclipse that took place on 1118-Jun-05 was total. It was visible in China, in Korea and in Japan (Liu, 2021). Scribes sometimes make mistakes. The text cited by the referee is from the *Wenxian tongkao* (文献通考, comprehensive examination of existing literature) compiled in 1317 CE by Ma Duanlin 馬端臨 (1254-ca.1325 CE). After 200 years and despite all the care taken by the historian responsible for this impressive work, errors and copying mistakes are to be expected. This text should not be taken as evidence of contamination. We kept this text in Dataset S1 for the sake of transparency.

The referee overlooks the second text, from the Song Shi 宋史, that is provided in the same entry and which doesn't refer to a partial eclipse (Figure 3).

Neither of these texts provided any information about the colour of the eclipse, which precluded the use of the Danjon scale.

Calendar Date	Eclipse Type	Luminosity Danjon scale	Li	Long	Capital	Original description	Translation
1118-Jun-05	Total	-	34.47	114.2	Bian (Kaifeng)	(1) 八年五月丙申，月食九分。 《文献通考》卷二百八十五·象緯八 (2) 重和元年五月丙申，月食。 《宋史》卷五十二·志第五·天文五	(1) Emperor Huizong (Hūizōng 徽宗) of Song, 8th year of the Zhenghe (Zhènghé 政和) reign period, 5th month, day bingshen (bǐngshēn 丙申); the moon was 9 fen (fēn 分) eclipsed. [Wenxian tongkao • Xiangwei] chapter 285 (2) Emperor Huizong (Hūizōng 徽宗) of Song, 1st year of the Chonghe (Chónghé 重和) reign period, 5th month, day bingshen (bǐngshēn 丙申); the moon was eclipsed. [Song shi • Tianwen zhi] chapter 52

Figure 3. Screen shot of the excel sheet "Dataset S1_China" for the entry pertaining to the total lunar eclipse of 1118-Jun-05.

This contamination is significant in their database. For example, roughly counting, in Chinese and Korean records, only 30 in 70 entries and 12 in 41 entries have involved this marker, respectively. I have not counted them in Japanese records, as their entire cell contents were difficult to display in my computer. This contradicts their methodological descriptions. Therefore, the authors should just list up what has been described as total eclipses. The authors also have to clarify what criteria they have used to select reports for total eclipses from the European historical documents.

Reply: This statement is incorrect. There is no contamination. As long as we have the precise date, we can determine whether the eclipse was total, without a specific notation. The absence of the notation 既 “total” alone does not prove that the eclipse was partial. If the record mentions an observation of an eclipse that we independently know to have been total at that time, then it is a valid entry.

It is well known that the Korean and Chinese records of eclipses from the 12th and 13th centuries contain only summaries of the original astronomical records (Stephenson, 1998 & 2002).

We also note that in 1998 and 2002, Stephenson examined the completeness and reliability of the eclipse records in the *Goryeosa* (高麗史) and concluded that “some 90 per cent of the eclipse reports ... are reliable.” (Stephenson, 1998, p. 711)

Stephenson later notes that “the dating accuracy for eclipses of the Moon is also extremely high, both for observations and predictions” (Stephenson, 2002, p. 243).

The author never observed any contamination from partial lunar eclipses in the *Goryeosa*. And our results agree with Stephenson (1998, 2002).

4. Eclipse visibility condition

The authors have claimed, "we feel it is possible to estimate the proportion of observed and missed lunar eclipses in Europe". In order to claim so, they need to provide an error bar for eclipse magnitudes at each case, clarifying the expected geographical extents of each observational site for each eclipse report with convincing explanations.

Reply: Espenak and Meeus (2009) and more recently Liu (2021) have developed several useful tools to calculate the number of solar and lunar eclipses visible over a given country and region. They also produced visibility maps for each solar and lunar eclipse that occurred in the last 4000 years, showing the geographical extent of each eclipse. If we use these tools and compare them with the number of eclipses found in historical sources, it is possible to estimate the proportion of observed and missed lunar eclipses.

All the visibility maps of the 12th and 13th century lunar eclipses considered in this study are attached to this response. They have also been included in the Zenodo repository so that the reader can readily reproduce our results. These maps were obtained from Liu (2021).

Inspection of these maps makes it clear that lunar eclipses have a wide footprint on the ground and that most of them were visible across much of Europe.

The authors have claimed, "In most cases, lunar eclipse observations were generally made near the monasteries that reported them". If this were truly the case, the authors would be able to clarify their geographical extents and provide error bars for magnitude of each lunar eclipse. Without such explanations, they cannot provide a success proportion for the eclipse observations.

Reply: We stand by our statement that in the 12th and 13th centuries a number of lunar eclipse observations were made in or near the monasteries that reported them. This statement agrees with the observations of Stephenson (1997, page 378, see also Figure 4), who wrote the following description in the book *Historical Eclipses and the Earth's Rotation*:

chronicles, translated most of the records cited by Celoria and Ginzel. However, several of his quotations are incomplete and his translations occasionally lack technical accuracy. In this chapter, I have either re-translated or cited reliable translations of all of the medieval European texts known to me which either (a) assert that a central eclipse was witnessed or (b) specifically deny totality or the ring phase.

Chronicles of medieval Europe can be divided into two main types: regional and local. Regional annals, typified by the *Anglo-Saxon Chronicle* and many other British chronicles, drew on a wide area. These often noted eclipses, but usually the place of observation is very uncertain. In contrast to the regional annals, monastic and town chronicles – especially those of continental Europe -- were mainly concerned with their own locality. It is thus usually quite reasonable to assume that eclipses noted in these works were observed at the centre where the chronicle was compiled. Sometimes this same place is specifically mentioned in the report of an eclipse, while occasionally the annalist tells us that he was an eyewitness. In general, if an obscuration of the Sun was particularly large, descriptions from even quite neighbouring locations differ in so many details as to be obviously independent. Annals of monasteries and towns provide the main source of the observations discussed in this chapter.

Figure 4. Print screen of the page 378 of the book *Historical Eclipses and the Earth's Rotation* written by Richard Stephenson (1997).

In fact, the authors' assumptions are not quite convincing, as British chronicles have involved some celestial reports even from Northern France and Italy (DOI: 10.1177/00218286221097111 and DOI: 10.5194/hgss-11-81-2020). Unfortunately, I am rather skeptical about the authors' claims, as some British reports were imported all the ways from Italy and Northern France.

Reply: The statement made by the referee is not entirely correct and must be viewed with caution. The two studies cited by the referee are the following:

(1) *Accuracy of eclipse records in the Anglo-Saxon Chronicle* by Leslie V. Morrison, F. Richard Stephenson, and Catherine Y. Hohenkerk.

(2) *Provenance of the cross sign of 806 in the Anglo-Saxon Chronicle: a possible lunar halo over continental Europe?* by Yuta Uchikawa, Les Cowley, Hisashi Hayakawa, David M. Willis, and F. Richard Stephenson.

The authors of articles (1) and (2) have indeed identified three celestial events that were probably observed beyond England. The first two first events are solar eclipses that occurred in 538 and 540 CE and were probably observed in Rome, Italy. The last celestial event is a halo that took place in 806 CE and was probably observed in Sens, France. However, these events took place several centuries before the period we are investigating. The *Anglo-Saxon Chronicle* (60 BCE - 1154 CE) has a complicated history and it is well known that a number of early entries are not original and that the authors used material from other sources, such as Bede.

Interestingly, the authors of study (1) concluded that the lunar and solar eclipses documented in the 12th century in the *Anglo-Saxon Chronicle* were visible in England, where the text was compiled.

The referee must bear in mind that in the 12th and 13th centuries there was an increasing interest in local and regional affairs. Annalists and chroniclers reported more frequently on the history of their monastery and described matters that directly affected their community (famines, natural disasters, conflicts, etc.), and a number of celestial events were observed locally.

Despite the referee's assertion, we have done our best - whenever possible - to trace the origins of the eclipse reports we have investigated. In some cases, we were able to show that the reports were not original, while in others we were able to demonstrate that the reports were original and probably even witnessed by the author. This work can be found in Dataset S1, in the column "Source description".

We stand by our statement that it is possible to estimate the proportion of eclipses missed and observed in Europe.

Can we consider such geographical uncertainty something negligible? That is not true. For example, the authors have assumed the observational site as Bayern and Colmar for the lunar eclipse on 1241 April 27. Of – by any chance – their provenances are expected from Königsberg, the Moon would have been set (04:15) before the onset of total eclipse (04:46). As shown in this example, lunar eclipses can get significant variations for the visibility condition even in the European sector. Here, I am not claiming this report is actually from Königsberg. Rather, I am casting caveats on the geographical uncertainty. One of solar eclipse report in the *Anglo-Saxon Chronicle* was seemingly derived from Italy. As such, it is far from secure to assume their geographical uncertainty to be negligible in their studies. Therefore, if they indeed insist maintaining their discussions on the success proportion, they must show geographical error margins for each eclipse report and resultant error bars for the success proportion for these eclipse reports.

Reply: As mentioned in the readme of Dataset S1, and in the response to the referee submitted in July 2022, all the locations given in Dataset S1 refer to the places where the sources reporting the eclipse were written. We did not assume the observational sites to be Schäftlarn and Colmar for the lunar eclipse that took place on 1241 April 27.

But the example chosen by referee gives us the opportunity to further illustrate that a number of astronomical observations were indeed made near the monasteries that reported them:

Annales Scheftlarienses maiores

The *Annales Scheftlarienses maiores* were written at the abbey of Schäftlarn (Bavaria, Germany) and cover the period 1092-1247 CE. Source criticism suggests that the first part of the annals is based on sources from Melk abbey (Austria) and Prüfening abbey (Ratisbonne, Germany). After 1172 CE, the annals focus mainly on local matters and seem to be independent from other sources. It is therefore likely that the eclipse was witnessed by a member of the monastic community of Schäftlarn (Bavaria, Germany). A look at the lunar eclipse visibility map confirms that the eclipse was indeed visible in Bavaria.

Annales Colmarienses Minores

The *Annales Colmarienses Minores* were written in Colmar (Alsace, France). They span 1211 to 1298 CE. Source criticism suggests that the part covering the period 1211-1260 CE derives from sources written in Alsace and in particular in the city of Strasbourg, which is about 60 km from Colmar. And a look at the visibility map confirms visibility of the eclipse across Alsace.

5. 'Historical Lunar Eclipse Atlas' must be removed

The authors are still maintaining their 'Historical Lunar Eclipse Atlas' with some excuses. At least, the URL is still valid. However, if these sites are not observational sites, this atlas cannot be called as a 'Historical Lunar Eclipse Atlas', as the contents contradict the title. Some readers could be easily misled that this were truly an 'eclipse atlas'. This is scientifically incorrect. They must remove this misleading online supplement at <https://arcg.is/XyOKX>.

Reply: We do not follow the referee's reaction... The revised version of the manuscript we resubmitted in July 2022 did not include the *Historical Lunar Eclipse Atlas*, and any material at this link is no longer formally associated with the present submission.

6. The authors' discussions on the proportion of observed and missed lunar eclipses

Their discussions on the success proportion are overemphasized. Their paragraphs in P3 L102-110 should be either removed or completely rewritten. Firstly, without knowing the exact observational sites, it is extremely difficult to assess if certain observers managed to see given lunar eclipses from certain observational sites. Of course, in many cases, the European sectors may have been completely covered by the eclipse visibility zones. However, there are some marginal lunar eclipses where Europe is situated in the visibility boundary. Even slight ΔT modifications can change visibility conditions for total lunar eclipses (DOI: 10.1111/j.1468-4004.2005.46511.x). It would be far-fetched to derive exact percentage without exact observational sites (See Major Comment 4 too).

Reply: Indeed. But we should bear in mind that totality, which is the most important stage for assessing stratospheric turbidity, was not always fully visible over Europe during these "marginal lunar eclipses". And there are many cases where the Moon was so low on the horizon that it would have been difficult for inexperienced observers to detect totality and report the colour of the eclipse. Schaefer (1990) has pointed out that lunar eclipses are unlikely to be seen under good conditions when the Moon is 2° below the horizon, due to daylight/twilight as well as obstacles obstructing the view.

Let's consider as an example the case of the marginal lunar eclipse of 21 January 1125. The visibility map in Figure 5 shows that totality reached Iceland, but not the rest of Europe. We would thus question the utility of considering an eclipse that had almost no chance of being observed in most of Europe.

Figure 5. Visibility map of the total lunar eclipse of 21 January 1125

To address the referee's concerns, we have updated our calculations. We have calculated the number of potentially visible total lunar eclipses for 12 locations in Europe, namely Reykjavik (Iceland), Madrid (Spain), London (England), Paris (France), Brussels (Belgium), Rome (Italy), Berlin (Germany), Prague (Czech Republic), Vienna (Austria), Stockholm (Sweden), Warsaw (Poland) and Kyiv (Ukraine). These locations were chosen because they cover most of Europe and should provide a reasonably good estimate of the number of potentially visible eclipses in England, France, Belgium, Italy, Germany and Austria, where the vast majority of lunar eclipse observations were made in the 12th and 13th centuries from our survey of the sources. The circumstance tables are from Liu (2021), who uses the most recent historical values of Delta T (ΔT) (Stephenson et al., 2015; Morrison et al., 2021).

A total lunar eclipse was considered "visible under good conditions and over large parts of Europe" if the Moon was at least 1° above the horizon in at least two of the 12 locations listed above during the totality phase. After removing the eclipses that did not meet the above criteria, we are left with 64 total lunar eclipses that were visible, weather permitting, over much of Europe and under good conditions. And we were able to find 188 historical accounts pertaining to 51 of these eclipses. We thereby conclude that we can estimate fairly accurately the number of eclipses missed and observed in England, France, Belgium, Italy, Germany and Austria, where most observations were made.

We have written a four-page Word document explaining in detail the methodology for calculating the potentially visible eclipses. The document is available in the Zenodo repository and can be downloaded directly here:

https://www.dropbox.com/sh/g7jso03xbw6nxzz/AAcTd_s5ThxGHShcUyt_X5lHa?dl=0

We also provide in the Zenodo archive all the R codes and circumstance tables needed to reproduce our calculations. The reader only needs to download the open-source statistical software R and copy our script into the software's console.

After all, the authors have avoided identifications of the observational sites, as they have explicitly

clarified in the previous rebuttal letters.

Reply: This statement is not entirely correct. Every record has been carefully examined. For example, we found that a total lunar eclipse reported by the Scottish historian John of Fordun in the year 1186 was actually observed in Jerusalem, not Scotland.

We have also recovered a lunar eclipse observed in 1277 in the town of Placenzia, Italy, and have stated that the totality phase was probably not visible from that location. This eclipse was not included in our calculations, but we kept the record for the sake of transparency. The text can be found in the Dataset S1_Extended. This applies not only to European but also to East Asian sources.

Our work is certainly not perfect, but it cannot be said that we did not scrutinise every record critically, consistently and with great attention.

Secondly, there are significant contaminations of prediction records, especially in the East Asian records (See Major Comment 7). The authors have not distinguished the observations from predictions. Then, it is quite possible that the alleged high success rate in East Asia is not because of their observational efforts but because of their accurate prediction capability. Therefore, the authors cannot derive success rates on the basis of what they have provided.

Reply: This assertion is false. As stated in the methodology section of the manuscript we submitted in July 2022, no predictive record has been used to calculate success rates. We invite the referee to read our responses in Section 7 for further details.

After considering the referee's comments and after discussion with the manuscript editor, we have decided to give less weight to the East Asian sources. Discussion of the number of lunar eclipses observed and missed in East Asia has also now been omitted.

Thirdly, the authors have not clarified the cases of impossible eclipse reports (e.g., those with wrong dates). This has been detailed in another major comment. Apparently, the authors have coupled eclipse records with the lunar eclipses in Espenak and Meeus (2009). Then, the unreal eclipse reports were omitted here and exaggerated the 'success rate' as well.

Reply: Dataset S1 contains several columns in which we examine the dating accuracy of each record. Whenever possible, we have tried to identify dating errors and to determine whether the eclipse was visible from a particular location. In some cases, we even added a warning to indicate that the record should be considered with caution. An example is presented in Figure 6.

Calendar Date	Eclipse Type	Luminosity	Danjon scale	Lat	Long	Capital	Original description	Translation	Accuracy of the eclipse reports
1197-Aug-26	Total	-	37.97	126.55	Songdo	七月西黄月食	The 7th month, day bysongin (西黄), the moon was eclipsed.	The eclipse is said to have occurred on the 8th of September 1197 (after the recorded date, originally expressed on the lunar calendar, is converted to the Julian calendar). But no eclipse occurred at that time. The closest eclipse occurred on the 29th of August 1197. This report must be considered with caution.	

Figure 6. Screen shot of Dataset S1 showing that the authors tried to identify dating errors.

7. Contaminations of prediction reports in East Asian records

The authors' "data" from East Asia are seriously contaminated by prediction reports. The authors have claimed: "The proportion of eclipses recorded by observers in Asia is also substantial (91%) (Extended data Table 2), motivated by the potential of astronomical phenomena to act as portents with political significance". However, it is far-fetched to claim them as "recorded by observers in Asia" without any critical assessments, as East Asian eclipse records involve significant amounts of predictions among them.

This danger is clearly confirmed in the data table too. For example, on 1103-09-17, Honchō seiki reported that the lunar eclipse did not appear owing to cloud cover. This is clearly a prediction. The authors have admitted this too, while they have listed this report as an one of "eclipses recorded by observers in Asia". This is a self contradiction, as the authors have listed these database "in search of credible lunar eclipse observations".

Reply: False claims. The lunar eclipse of 1103-09-17 was clearly stated as a prediction in Dataset S1 when submitted in July 2022 and was not used to calculate any success rate. We kept the record in Dataset S1 out of transparency for the reader. The authors have taken pains to distinguish predictions from observations. For more information, see the response below.

In fact, there are a number of precious studies which have confirmed significant contaminations of eclipse prediction reports in the East Asian historical documents. Foley (1989) have analysed solar eclipse records in the Chinese and Korean historical records and concluded that many of them were not actually visible from China or Korea.

Reply: In Dataset S1 submitted in July 2022, we did our best to identify the lunar eclipses that were not visible in Japan, Korea and China and added a note of warning where necessary (Figure 7).

Calendar Date	Eclipse Type	Luminosity y Daajon scale	Original descriptions and translations	Comments on the eclipse report
1200-Dec-22	Total	-	正治二年十一月十五日 (1200年12月22日) Shōji 2.11.15 1200.12.22 (1) 〔玉葉〕 六十六 Gyokuyō, vol. 66 十一月十五日丁卯、月蝕。 11th month, 15th day... lunar eclipse (2) 〔猪俣圃日記〕 十一 Inokuma Kankoku-kō, vol. 11 十一月十五日丁卯、天霽、月蝕也。但不見、他州能云々。 11th month, 15th day... clear skies. Lunar eclipse. But it was not visible, said to be of another country. (3) 〔明月記〕 九 Meigetsuki, vol. 9 十一月十五日、天晴、...月蝕、但不見云々。 11th month, 15th day. Clear skies. ... lunar eclipse, but it did not appear.	Most of the eclipse was not visible in Japan, as specified by the Inokuma Kankoku-kō. (1), (2) & (3) These reports are most likely predictions, not observations.

Figure 7. Screen shot of Dataset S1, showing that we issued warning when a lunar eclipse was not visible in East-Asia.

This was also the case with Japanese eclipse records. Soma and Tanikawa (2015) analyzed Japanese eclipse records and concluded as follows: "The reliabilities of eclipse records between 7th and 11th centuries were discussed in this paper. Eclipses in the 7th century were recorded in the Nihongi and their reliability changed according to the volumes, namely the records in the volumes in the β group were based on actual observations whereas those in the α group were based on predictions at the time. Eclipses in the 8th century were recorded in the Shoku-Nihongi but most of them were written according to the predictions at the time. Eclipses between the 9th and 11th centuries were recorded in various literatures and diaries but it is not easy to decide whether they were based on actual observations or not." In fact, Yuasa (2010) have identified 4 impossible eclipse records among 37 realistic eclipse records in Azuma Kagami and 1 impossible eclipse records among 19 realistic eclipse records in Gyokuyo.

These cases have evidently shown that there are significant amount of prediction reports among the East Asian eclipse records. Of course, some of the eclipse predictions were successful enough. In this case, there should be Renumerous prediction reports in what the authors have tabulated as their 'data base'.

Reply: We agree with Yuasa (2010) and with Soma and Tanikawa (2015). The Japanese records do indeed contain predictive records. Whenever possible, we have tried to identify them. More details below.

However, the authors have not clarified which records are actually on the observational basis (not predictions) at all. Contaminations of the predictions records do not allow us to derive any success rates and endanger reliability of the report accuracy as 'observations'. This is a fatal flaw for the authors' datasets for the East Asian records, which cannot tolerated in astronomical and philological contexts.

Reply: This statement is false. We clarified in Dataset S1 which records are deemed observations and which records are predictions, as exemplified in the screenshots below (see Figure 8):

Figure 8. Screen shot of Dataset S1 indicating care taken to identify prediction records.

None of the predictive records that we identified in this study was used to derive any estimated success rate, as indicated in the methods section of the manuscript we resubmitted in July, lines 481-484 (see also Figure 9). But for the sake of transparency, we decided to keep them in Dataset S1 submitted in July 2022, accompanied by the information needed to distinguish between apparent observations and predictions.

476 *Assessing the reliability of historical sources.* The reliability of each eclipse observation was assessed via historical
 477 source criticism and by reference to the five millennia catalogue of lunar eclipses (1999 BCE–3000 CE)²⁴ and
 478 eight millennia catalogue of lunar eclipses (4000 BCE – 4000 CE)²⁴. Care was taken to identify second-hand
 479 reports, i.e., those which a given author did not witness but instead drew information from another source. Frequent
 480 duplication occurred in Western and Eastern Christian sources due to common underlying source materials and
 481 the scribal practices of copying, synthesizing and piecemeal updating of annals and chronicles. East-Asian records
 482 often contain predictions rather than actual records of observation²⁵. Prediction records were excluded from
 483 analysis. A table listing all total lunar eclipses records and providing summary context on the respective sources
 484 is presented in Dataset S1.

Figure 9. Screen shot of the methodology section submitted to Nature in July 2022.

7. East Asian "records"

The authors claimed to maintain East Asian 'records', while they have also admitted by themselves that the East Asian records played little role in this manuscript.

Reply: In previous submissions, we did our best to convince the referee that providing the East Asian records was important as a reference for scholars (and which we legitimately explored as part of this research to identify whether East Asian total eclipse coloration observations existed, given that even a single observation would be valuable for the purposes of the manuscript). We accept that we did not succeed in convincing the reviewer. As mentioned above, following a valuable discussion with the editor of the manuscript and after reading the reviewer's comments, we have removed the East Asian sources from Dataset S1, except for the texts explicitly mentioned in the manuscript.

Nevertheless, for the sake of transparency and to allow readers to fully reproduce our results, we provide all the historical sources that we considered in the Zenodo repository, which also contains all the underlying data and codes used. We hope that this compromise is acceptable.

However, their stance self-contradicts their own stance for Islamic records. According to the authors, the authors "didn't include any records in the Dataset S1 due (1) to the limited number of lunar eclipse records (approximately 6, according to the survey made by Stephenson and Said in 1997), (2) because the records are readily accessible and (3) because none of the reports provides information about lunar eclipse color". Apart from (1), the situation is virtually the same with East Asian records. In fact, for (2), the East Asian records are readily accessible with previous publications such as Xu et al. (2000) and Kanda (1935). For (3), none of these reports provide information about lunar eclipse color except for a single Japanese report. As they are omitting Arabic records in this logic, they have to be self-consistent to East Asian records too. This self-contradiction should not be tolerated.

Reply: We do not agree with the referee. We have raised this point in previous revisions. Kanda (1935) is not "readily accessible". His volume has long been out of print and is held by very few, if any, libraries outside Japan; it was necessary to ask the National Diet Library to send us a digital scan. Moreover, his records are only transcriptions, with no translations into modern Japanese, much less English.

A significant number of eclipse reports from China and Korea were, moreover, omitted by Xu et al. (2000). We therefore decided to reexamine the Song Shi (宋史), the Jin Shi (金史), and the Yuan Shi (元史), the Wenxian tongkao (文獻通考), the Qinding Xu wenxian tongkao (欽定續文獻通考), and the Goryeosa (高麗史).

To our knowledge, only six records pertaining to total lunar eclipses in the 12th and 13th centuries have been found in Islamic sources, which is far fewer than the hundreds of records found in East Asia sources. A translation of some of these texts was presented by Said and Stephenson (1997). Unfortunately, the authors did not provide the original texts and our authorship does not include competency in Arabic. Referee#2 specifically asked us to provide the original texts of each report in Dataset S1, so we had to omit these texts in the previous resubmissions. Otherwise, the English translations of these texts would have been provided in Database S1 along with the reference to Stephenson and Said (1997).

The authors paid little attention to the probable contaminations of the prediction records although they cited East Asian records "in search of credible lunar eclipse observations".

Reply: This statement is false. We refer the referee to Section 6 entitled "Contaminations of prediction reports in East Asian records".

Moreover, their translations are suffering from numerous philological flaws and needlessly reducing the reliability of their discussions. If they still insist on claiming their searches on East Asian records, they should just cite Xu et al. (2000) for Chinese and Korean records and Kanda (1935) for Japanese records.

Reply: For more details, we refer the referee to our previous response. Xu et al. (2000) provided only a limited number of records of lunar eclipses from Chinese, Korean and Japanese sources. Kanda (1935) cannot currently be found in most libraries and no English translation is available.

Moreover, the authors are not showing "all the lunar eclipse records [they] considered". For example, for the East Asian records, the authors have already 'cherry-picked' the eclipse reports following Espenak and Meeus' database. As shown in Yuasa (2010), there are some impossible eclipse reports (see Major Comment 6), which has been ignored in the authors' dataset.

Reply: We do not follow this argument. How can we use "impossible eclipse reports"? We have indeed excluded from our analyses the reports of lunar eclipses whose dates were so wrong that they could not be associated with the eclipse dates given by Espenak and Meeus (2009) and Liu (2021).

The impossible eclipse reports mentioned by Yuasa (2010) were not, however, ignored by the authors. Although we did not use these reports, they were included in Dataset S1 submitted in March 2022. At that time, Dataset S1 contained not only reports of total lunar eclipses, but also reports of partial and penumbral eclipses that had been examined. Following the referee's request, we removed these records from Dataset S1.

In fact, at the beginning, the authors have shown translations for what they had been interested in in the East Asian records. This "cherry-picking" has been modified only after I had raised some concerns.

Reply: Referee#2's initial comments were very constructive and prompted us to provide the full translations of all the reliable East Asian texts that we had encountered.

8. Dataset S1

They should remove columns for Lat, Long, Country, and locality. This is because it is misleading to provide geographical coordinates for where chronicles were written as "locality". For example, Anglo-Saxon Chronicle has involved a number of astronomical reports even from outside of England, as exemplified in some eclipse records (seemingly from Rome; DOI: 10.1177/00218286221097111) and halo records (seemingly from Northern France; DOI: 10.5194/hgss-11-81-2020).

Reply: We re-emphasise that the columns "Latitude", "Longitude", "Country" and "Locality" in "Dataset_S1_Europe" do not refer to the places where celestial phenomena were observed, but to places (i.e. monasteries, towns) where historical sources were written.

In this regard, it looks inappropriate to classify these records into countries in the modern definition. Even worse, back in the medieval periods, these countries have had different borders. It looks slightly weird to separate Scotland from the United Kingdom. The authors may wish to emphasize a benefit for modern readers. However, this crude classification will be probably subjected to criticisms from medieval historians. If they wish to keep their columns for the countries, they need to clarify the country name at that time.

Reply: The demarcation of state borders in Europe during the Middle Ages is often not precisely known. We also note that the borders were not static and changed between 1100 and 1300 CE. There is therefore a risk of introducing errors into Dataset S1. Our aim is to help modern readers who do not necessarily know the ancient names of the kingdoms and states of medieval Europe. Therefore, we feel that keeping the modern countries is not only more accurate in providing a geographical reference known with certainty, but also more helpful to *Nature* readers. As for Scotland, we thank the referee for pointing out the typo.

Dataset S1 is entitled as Continental Europe, while it involves reports from the United Kingdom and Continental Europe. They may need to either remove the British records from their supplementary files or add one more dataset tab to accommodate British records.

Reply: 'Continental Europe' was replaced by 'Europe'.

9. Continental European records

Here, in many cases, their English translations are missing. This stance significantly contradicts their claims on the values of English translations for the East Asian records.

Reply: The European records were investigated by the first author. The records were so numerous that a decision had to be made. Either providing a description for each source or providing a translation. The author in charge of the records decided to focus his effort on the source criticism rather than providing translations. For an English-reading audience, European records are more easily read, or machine-translated, than East Asian records. And we will be happy to provide a translation to any researcher eager to know more about the texts we used.

10. Do they need English translations for East Asian records?

The authors have cited Sommerstein (1973) here to justify their philological flaws.

"There will never be a perfect translation of Aristophanes. There have been many translations, with many and diverse merits: one thinks of the poetical grace of F. L. Lucas, the deftness of Godley and Bailey, the easy naturalness of John and Patricia Easterling, the zest of B. B. Rogers, the modernity of Douglass Parker and his associates; but combine all these and you are still left a long way from a perfect translation, for there is much truth in the paradox that the only really perfect translation is the original."

Sommerstein, A. H., 1973. On Translating Aristophanes: Ends and Means. *Greece & Rome*, 20(2), 140–154.

Sommerstein is also right. I appreciate this opinion. Then, in this case, why do they need these inaccurate translations here?

Reply: Sommerstein (1973) reminds us that the perfect translation does not exist; all translations are inaccurate in one way or another. And translations of the same text made by different translators will always be different. That our translations might diverge on some points from those that the referee might make does not mean that our translations are wrong or inaccurate.

As the authors have also agreed, their translations for East Asian records have played little roles for their manuscript. In fact, they are mostly irrelevant.

Reply: We have outlined in the last two revisions why we believe the East Asian records are relevant to the manuscript. But the referee unfortunately does not share our view.

They have numerous philological flaws. The authors have treated translations of East Asian records too casually. Unfortunately, their translations only reduce their data reliability needlessly.

Reply: We have taken the translations of the East Asian records very seriously and have tried to accommodate almost all the referee's wishes. For example, we added tone marks to our Chinese

romanization, although this practice is not common in this field. Our translations were not done carelessly, and we offer them confident that the majority of readers (including experts in the respective languages) will agree.

They should just remove these needless tables for East Asian materials which was not actually used in this manuscript with a single exception (a Japanese report in 1229).

Reply: As mentioned in our previous responses, we decided to remove the East Asian sources from Dataset S1. They will now only appear in the Zenodo repository.

11. Korean records

I disagree with the authors' claims. The authors should refer what is permanently accessible. If the URL gets changed, everything will lose traceability. This happens to many databases. This is especially the case when the readers need to consult individual pages. Yes, it happens even to the NGDC websites! They need to refer what is permanently accessible. This is important, especially if the authors intend this article to be consulted not only over coming years but also over coming decades.

Reply: Should we really prefer a book published in 1972, accessible only to a limited number of readers and known to have several shortcomings (Breuker, 2012), to a new critical edition prepared by a team of experts at the National Institute of Korean History (NIKH) and accessible worldwide? The NIKH has made enormous efforts since 2009 to make the Korean historical sources accessible to a wide audience. We respect the referee's opinion, but feel that such an initiative should rather be commended. All the links we have provided in Dataset S1 are permalinks, i.e. URLs that are intended to remain unchanged for many years.

There are numerous philological flaws too. For example, the authors should have transcribed Korean terms not to Mandarin but to Korean, as they have also admitted. However, for example, on 1132-Mar-04, the authors have transcribed "丙子" as "bingzi" in Mandarin, although they have claimed modifying such issues.

Reply: We have indeed forgotten to use the Korean term. This is not a philological flaw, but a typo. "Bingzi" has been replaced by "byeongja".

The authors have wrongly used square brackets. For example, on 1168-Mar-25, the authors have translated "密雲不見" as "it could not be seen because of the [dense] clouds", although "密" indicates "dense".

Reply: Again, this is a simple typo, not a philological flaw. It has been corrected.

The authors have inconsistently translated similar phrases. For example, on 1277-May-19, the authors have translated "雨不見" as "it could not be seen because of the rains" in a plural form. However, on 1259-May-08, the authors have translated "因雨不見" as "but it was not seen because of the rain" in a single form. What is the cause if this discrepancy?

Reply: We have just added an "s" by mistake. This is a typo, now corrected.

Moreover, the translation of "因" is not well clarified here.

Reply: We do not understand the referee's comment very well. The translation is clear. We have rendered "因" as "because", just as it should be.

They are just initial examples. There are more philological flaws here.

Reply: We disagree with the blanket statement of the referee, supported only by the above few typographical errors likely to have been identified during our routine proofing of the submitted materials before any publication.

12. Chinese records

There are still numerous philological flaws here. Here are only a few examples.

Some phrases like "不見" have been translated inconsistently. For example, on 1208-Feb-03, the authors translated "雨不見" as "it was not seen because of the rain". However, on 1186-Apr-05, the authors have translated "陰雲不見" as "it was hidden by the [shady] clouds [and was not seen]". As the authors are translating the same phrase, they should make such translations consistent (e.g., "it was not seen because of the shady cloud").

Reply: We do not agree with the referee. Our translation is correct. "不見" does not always have the same meaning. It has various pronunciations and context-dependent meanings. When pronounced as "jiàn", the term "不見" should be translated as "was not seen". Sometimes, however, it also makes sense to pronounce the character "見" as "xiàn", which means "not shown" or "hidden" (and therefore was not seen). To our knowledge, the description of the lunar eclipse of 1186-Apr-05 has never been translated into English. However, we have consulted several translations of the same expression and found that "不見" has been translated both ways. There are many differences between Chinese and English. And depending on the context, it is not always possible to use the same wording for all sentences.

Some phrases with different wordings resulted in the same translation. For example, on 1219-Jun-29, the authors have presented two variants "雲陰不見" & "霧雲不見" but provided the same translations as "it was hidden by the [shady] clouds [and was not seen]". The authors should translate different Chinese characters (雲陰 and 霧雲) differently, although they certainly mean something similar with one another.

Reply: This statement is not correct. 霧 is simply a variant form of 陰. Therefore, there is no meaningful difference to be captured by varying the translation. They are not "similar" in meaning; they are identical.

Some phrases added what its not included in the original text without any caveats. For example, on 1226-Aug-09, the authors have translated "陰雨不見" as "could not be seen because of the overcast and rainy weather". The word "weather" does not appear in the original text. To be more loyal to the original text, it could be something like "could not be seen because of cloud and rain".

Reply: We do not entirely agree with the referee. Cloud is “雲” in classical Chinese. It is not the same as “陰雨”, which implies the meaning of cloudy, dark and rainy weather. We think that "overcast" is a more appropriate translation. We could also have translated the sentence as follows “could not be seen because it was overcast and raining”. But in the end, should we really argue about this? The translations proposed by the referee and the authors of this manuscript capture the same meaning. This illustrates once again that there is often no unique or correct way to translate a text. Every translation is subjective. Again, the translations we provided were simply meant to help the reader accessing texts that can only be read by a limited number of scholars.

Some phrases have been wrongly translated too. For example, on 1277-May-18, the authors have translated "寅四刻" as "By 4 marks in the hour of yin" whereas this should be "By 4th mark in the hour of yin".

Reply: There is actually no clear standard. For example, Stephenson (1997) and Xu et al. (2000) do not always use the suffix “th” in their translations (see Figures 10 and 11), while Steele (2000) does (Figure 12):

(4) AD 761 Aug 5 (total, mag. = 1.05): Ch'ang-an [class A]

Shang-yuan reign period, 2nd year, 7th month, day *kuei-wei* [20], the first day of the month. The Sun was eclipsed; the large stars were all seen. The Astronomer-Royal Ch'u T'an reported (to the Emperor): "On day *kuei-wei* the Sun diminished. The loss began at 6 marks (*k'o*) after the hour of *ch'en*. At 1 mark after the hour of *szu* it was total. At 1 mark before the hour of *wu* it was restored to fullness. The eclipse was 4 deg in *Chang*."

[*Chiu-t'ang-shu*, chap. 36.]

Figure 10. Screen shot of the page 248 of the book *Historical Eclipses and the Earth's Rotation* (Stephenson, 1997).

AD 1277 May 18 [China]

"Emperor Shizu of Yuan, 14th year of the Zhiyuan reign period, 4th month, day *guiyou* [10], full Moon; the Moon was eclipsed. The eclipse began at 6 marks in the hour of *zi* (LT = 23-1h). The eclipse was total at 3 marks in the hour of *chou* (LT = 1-3h). Maximum was reached by 5 marks in the hour of *chou*. Light reappeared by 7 marks in the hour of *chou*. By 4 marks in the hour of *yin* (LT = 3-5h) [the Moon] was restored to fullness." [*Yuan shi* • *Li zhi*] er ch. 53

Figure 11. Screen shot of the page 82 of the book *East-Asian Archeoastronomy: Historical Records of Astronomical Observations of China, Japan and Korea* (Xu et al., 2000).

• 7 October 1074 AD

"At the 1st mark of *ch'ou* the loss began on the eastern side. At the 6th mark, the eclipse reached its maximum ... Dawn broke and its return to fullness was not seen." [*Wen-hsien T'ung-k'ao*, 285]

"Beginning of loss at the 5th point of the 4th watch. Eclipse total at the 3rd point of the 5th watch." [*Yuan-shih*, 53]

Figure 12. Screen shot of the page 207 of the book *Observations and Predictions of Eclipse Times by Early Astronomers* (Steele, 2000).

They are just initial examples. There are more philological flaws here.

Reply: Again, none of the points raised by the referee qualifies as a philological flaw.

13. Japanese records

As I stated, the authors have to cite Yuasa (2010) in the main text. They have initially shortcut their access to Azuma Kagami using Yuasa (2010). They should appropriately appreciate Yuasa (2010) with their citation in the main text.

Reply: The authors have cited Yuasa (2010) more than 20 times in Dataset S1. And we have also consulted the original text of the Azuma Kagami, as can be seen from the reference in Supplementary Text 2. All the eclipse records found in the Azuma Kagami were collected by Kanda in 1935, several years before Yuasa (2010). And Kanda (1935) is cited in the main text.

For the 1258 reports, the authors have claimed, "The lunar eclipse of 1258 recorded in the Azuma Kagami indeed cannot be used to estimate past stratospheric turbidity. However, we believe this record is of great interest. It is important to show that we considered this record, as well as the reasons that led us to discard it. Otherwise at a later stage some readers aware of this account may wonder why we didn't mention it". This statement is self-contradictory. The authors must remove this unusable report from their supplementary text 2.

Reply: This is precisely the point we make in the Supplementary Text 2. The descriptions of lunar eclipses need to be analysed carefully. The Azuma Kagami is a perfect example in this respect.

Moreover, this report does not involve a technical term to clarify eclipse totality (既). These difficulties do not allow the authors to claim this report as an account for a total lunar eclipse, unfortunately.

Reply: We disagree. The entry from Azuma Kagami does not mention totality, but an entry from Hyakurenshō for the same date does. Clearly, they are referring to the same event. Retro-calculations by Liu (2021) and Espenak and Meeus (2009) confirm that a lunar eclipse occurred on that date and that it was total.

There are numerous philological flaws too. For example, the authors have not appropriately translated "中略" on 1103-Sep-18.

Reply: This is not the case. “中略” simply indicates that text was elided from the middle of the sentence. It is an indicator inserted by the compiler (Kanda). There are two such elisions in this entry. The first one is represented by an ellipsis "..."; the second one was inadvertently omitted, not inappropriately translated. The omission is not material. This is not a philological flaw but an insignificant omission.

It would have been very constructive and useful to the scientific community to list the typos that the referee encountered while reading the tables, and we would have been very happy to review and correct each of these. We thus invited the referee to do so in the last revisions we submitted. The table contains several hundred texts, and it is possible that a number of typos may remain despite all our efforts and care. But we feel that stating that our manuscript is riddled with philological flaws is not fair.

Their translation style for Japanese records is inconsistent with those for Chinese and Korean records. For example, on 1103-Sep-18, the authors have translated "辰刻" as "hour of the Dragon", whereas they should have translated this as "Tatsunokoku" following their style to show local time descriptions with transcriptions as done in Chinese and Korean cases.

Reply: The Japanese hours were translated into English for the sake of accessibility, as “Tatsu no koku” means nothing to the vast majority of readers. The translators of the Chinese and Korean texts took a different approach. It is not a philological flaw but diversity in translation style that does not materially affect the accuracy of the translation.

Some English translations are grammatically incorrect. For example, on 1103-Sep-18, they have translated an excerpt from Honchō seiki as "8th month, 14th day, lunar eclipse tonight, clouds covered it, did not appear". This translation is grammatically unacceptable.

Reply: There is nothing wrong with it grammatically; this is the laconic style used in English to write diaries. We addressed this point in our response to the previous resubmission.

The authors have ignored some phrases in the translations. For example, 勘脱力 is missing on 1103-09-17. This is probably this is an insertion by Kanda (1935). However, if this is the case, why has this been included in the 'original text'?

Reply: It is indeed an insertion by Kanda, which is our source text. This insertion could indeed have been omitted, but the lack of a translation of it and its inclusion are both immaterial to the accuracy of the translation.

The local time descriptions are inconsistently given for the Japanese translations. For example, on 1114-Feb-21, they have shown a translation of "Boar [9p-11p]". I suspect p indicates pm. However, for Chinese text, it is for example shown as "zi {Time = 23-1h}". Such an inconsistency should be resolved.

Reply: Indeed, p stands for pm. The equivalents are provided by us as a convenience to the reader, who can readily understand what 9p and 23 mean by the context.

We explained in Dataset S1 that the translations of the Japanese, Korean and Chinese records were made by different authors who used their own standard for the translations (and reflect the legitimately varying cultures of translations prevailing for each language). Each readme file provides a section entitled "notes on the sources", "notes on the transcriptions" and "notes on the translation" to help the reader understand the translator's work. The referee fails to explain how this minor inconsistency in the local time description impairs the readers' use of the translations.

Some phrases are missed in the translations. For example, on 1110-Oct-29, "誠如指掌" is missing in their translation.

Reply: False. This is a metaphor that means literally "truly as if pointing to one's palm." The connotation is rendered in the translation as "precisely." There is no missing sentence.

14. West Asian records

They have omitted the Islamic records. If they insist in maintaining East Asian records, they must include Islamic records with original texts as well. They show a double standard for East Asian records and Islamic records.

Reply: We would have liked to provide the original-language texts, but unfortunately they are not available in Said and Stephenson (1997), upon whose work we drew for our knowledge of the relevant Arabic sources.

The authors of the manuscript can read, understand and translate Latin, Russian, Chinese, Korean and Japanese sources from this period. They are also able to work with Syriac sources to a very limited extent, but unfortunately not Arabic texts.

We have now added the six missing Arabic texts to the Excel spreadsheet entitled "Dataset S1_Extended", available via the Zenodo repository. We added the translations as well as all the references needed to access the original Arabic texts.

The Syriac original texts are still missing, although the authors claimed to have tweaked this flaw.

Reply: This is not true. The original text written by Michael the Syrian was provided in the version of Dataset S1 submitted in July 2022 (Figure 13).

Figure 13. Screen shot of Dataset S1 showing that the Syriac original texts were already in the prior iteration of the database.

15. Supplementary Text 2

Here, the authors have tried to show their ‘careful treatments’ on the historical records, comparing a western record with a Japanese record. However, as I have repeated, these records have different cultural backgrounds and cannot be used for comparative references. Upon comparisons, we should compare two different objects with a similar background. Therefore, here, the authors have to show occidental records for their comparisons.

In the previous response, the authors have claimed, “We fully agree with the referee. The Azuma Kagami, and more generally all East Asian records, reflect a distinct cultural background. Therefore, one should be even more careful when using these sources to assess the impacts of volcanic eruptions on climate. For this reason, we feel that taking the Azuma Kagami as an example is appropriate.” This statement looks self-contradictory and does not make sense to me, unfortunately.

Reply: We feel that the referee did not fully understand Supplementary Text 2. There is no comparison between Western and Eastern records.

Supplementary Text 2 merely emphasises that working with historical reports of lunar eclipses requires careful interpretation and training. This is because descriptions of lunar eclipses can be misleading and lead the reader to believe that a dark lunar eclipse occurred. The lunar eclipse of November 1258, which occurred about a year after the 1257 CE Samalas eruption is a very good example of this. Indeed, a reader unfamiliar with Japanese historical sources or with the Azuma Kagami and its cultural background might be tempted to think that the author of the text was referring to a dark lunar eclipse, although this is not the case.

We hope that our explanations make more sense to the referee.

Minor Comments

P2 L71: "and their occurrences known precisely from astronomical retro-calculation."
=> This sentence looks unclear. Please better rephrase what the authors have done for astronomical retro-calculations. Looking at P3 L96, the authors have tried a short cut with Espenak and Meeus' database. They should clarify that they have entirely relied the astronomical calculations on Espenak and Meeus (2009) and these database has some issues with the recent ΔT revisions (Stephenson et al., 2016; Morrison et al., 2021).

Reply: We do not understand the referee's comment or why s/he suggests using the word "short cut"?

We did not perform astronomical retro-calculations. We used the calculations made by specialists in the field. This is not a secret. We have made this clear from the beginning in the main text, in the methods section as well as in Dataset S1.

We initially used the retro-calculations made by NASA (Espenak and Meeus, 2009). After the referee expressed doubts about the reliability of the work hosted on the NASA website, we used the calculations of Liu (2021), which are based on recent ΔT revisions (Stephenson et al., 2016; Morrison et al., 2021). We have cited both Espenak and Meeus (2009) and Liu (2021).

P3 L90-92: Again, the authors should remove the mentions to the East Asian records. The East Asian records have not played significant roles in their manuscript. If they still insist, they should just mention their comparison for the European records with the Japanese report in 1229.

Reply: This is mentioned a little later in the manuscript, in lines 113-114 and in lines 461-463.

P3 L92: Here, the authors have to explicitly clarify that they have tried a short cut with Kanda (19365) to avoid comprehensive investigations for original historical records. This sentence will mislead the readers, as if the authors have examined a number of Japanese historical records by themselves.

Reply: This sentence simply describes where records of eclipses are to be found; it does not mislead the reader into thinking we have pored over each record ourselves.

We have not "tried a short cut with Kanda to avoid comprehensive investigations." Kanda compiled the records and his work has stood the test of time. There is no need to return to the original sources. It is not a 'short cut' but commonplace citation of data compiled by another scholar. Again, Kanda has been used by several scholars, including for instance Stephenson (1997) in his book *Historical Eclipses and the Earth's Rotation*.

The following text from the methods section of our manuscript clearly shows that we have used Kanda:

“In Japan, the astronomical records are scattered in a variety of works ranging from privately and officially compiled histories, to diaries of courtiers and temple records²³. We thus mainly assessed the lunar eclipse observations compiled in the benchmark work, the *Nihon Tenmon Shiryô* (日本天文史料)⁵³, by Japanese astronomer S. Kanda.”

Additional References [those without DOI]

Foley, N. B. 1989, A statistical study of the solar eclipses recorded in Chinese and Korean history during the pre-telescopic era, M.Sc. thesis, Durham, University of Durham.

Sôma, M., Tanikawa, K. 2015, Reliability of Eclipse Records in Japanese Ancient Periods, Proceedings for Intensive Workshop on Ancient and Medieval Eclipse Data (2014 Nov 27–28), 2014, 5. [<https://www2.nao.ac.jp/~mitsurusoma/WS2014/soma.pdf>]

References

- Breuker, R., Koh, G., Lewis, J.B., 2012. The Tradition of Historical Writing in Korea, in: Foot, S., Robinson, C.F. (Eds.), *The Oxford History of Historical Writing: Volume 2: 400-1400*. Oxford University Press, pp. 119–137. <https://doi.org/10.1093/oso/9780199236428.003.0007>
- Espenak, F., Meeus, J., 2009. Five Millennium Catalog of Lunar Eclipses: -1999 to +3000 (No. TP-2009-214173). NASA.
- Kanda, S., 1935. *Nihon Tenmon Shiryô* (日本天文史料, Japanese Historical Records of Astronomy), Koseisha Koseikaku Co., Ltd. ed. Tokyo.
- Liu, Y.T., 2021. Eight Millenia of Eclipses (4000 BCE – 4000 CE) [WWW Document]. URL <http://ytliu.epizy.com/eclipse> (accessed 2.1.22).
- Morrison, L.V., Stephenson, F.R., Hohenkerk, C.Y., 2022. Accuracy of eclipse records in the Anglo-Saxon Chronicle. *Journal for the History of Astronomy* 53, 209–216. <https://doi.org/10.1177/00218286221097111>
- Morrison, L.V., Stephenson, F.R., Hohenkerk, C.Y., Zawilski, M., 2021. Addendum 2020 to ‘Measurement of the Earth’s rotation: 720 BC to AD 2015.’ *Proc. R. Soc. A.* 477, 20200776. <https://doi.org/10.1098/rspa.2020.0776>
- Schaefer, B.E., 1990. Lunar Visibility and the Crucifixion. *Quarterly Journal of the Royal Astronomical Society* 31, 53–67.
- Steele, J.M., 2000. *Observations and Predictions of Eclipse Times by Early Astronomers*. Springer Netherlands, Dordrecht.
- Stephenson, F.R., 2002. Eclipse Records in Early Korean History: The Koryo-sa, in: Ansari, S.M.R. (Ed.), *History of Oriental Astronomy, Astrophysics and Space Science Library*. Springer Netherlands, Dordrecht, pp. 237–243. https://doi.org/10.1007/978-94-015-9862-0_19
- Stephenson, F.R., 1998. Eclipse Records in Early Korean History: The Koryo-sa, in: Andersen, J. (Ed.), *Highlights of Astronomy*. Springer Netherlands, Dordrecht, pp. 710–711. https://doi.org/10.1007/978-94-011-4778-1_32
- Stephenson, F.R., 1997. *Historical Eclipses And Earth’s Rotation*. Cambridge University Press, Cambridge.
- Stephenson, F.R., Morrison, L.V., Hohenkerk, C.Y., 2018. The Provenance of Early Chinese Records of Large Solar Eclipses and the Determination of the Earth’s Rotation. *Journal for the History of Astronomy* 49, 425–471. <https://doi.org/10.1177/0021828618789850>
- Stephenson, F.R., Morrison, L.V., Hohenkerk, C.Y., 2016. Measurement of the Earth’s rotation: 720 BC to AD 2015. *Proc. R. Soc. A.* 472, 20160404. <https://doi.org/10.1098/rspa.2016.0404>

- Stephenson, F.R., Said S. Said, 1997. Records of Lunar Eclipses in Medieval Arabic Chronicles. *Bulletin of the School of Oriental and African Studies, University of London* 60, 1–34.
- Uchikawa, Y., Cowley, L., Hayakawa, H., Willis, D.M., Stephenson, F.R., 2020. Provenance of the cross sign of 806 in the *Anglo-Saxon Chronicle* : a possible lunar halo over continental Europe? *Hist. Geo Space. Sci.* 11, 81–92. <https://doi.org/10.5194/hgss-11-81-2020>
- Xu, Z., Pankenier, D.W., Jiang, Y., 2000. *East Asian archaeoastronomy: historical records of astronomical observations of China, Japan and Korea*, Earth Space Institute book series. Published on behalf of the Earth Space Institute by Gordon and Breach Science Publishers ; Cambria Press, Amsterdam, Netherlands : Amherst, N.Y.